# PADetBench: Towards Benchmarking Physical Attacks against Object Detection

## Abstract

Physical attacks against object detection have gained significant attention due to their practical implications. However, conducting physical experiments is time-consuming and labor-intensive, and controlling physical dynamics and cross-domain transformations in the real world is challenging, leading to inconsistent evaluations and hindering the development of robust models. To address these issues, we explore realistic simulations to rigorously benchmark physical attacks under controlled conditions. This approach ensures fairness and resolves the problem of capturing stricly aligned adversarial images, which is challenging in the real world. Our benchmark includes 23 physical attacks, 48 object detectors, comprehensive physical dynamics, and evaluation metrics. We provide end-to-end pipelines for dataset generation, detection, evaluation, and analysis. The benchmark is flexible and scalable, allowing easy integration of new objects, attacks, models, and vision tasks. Based on this benchmark, we generate comprehensive datasets and perform over 8,000 evaluations, including overall assessments and detailed ablation studies. These experiments provide detailed analyses from detection and attack perspectives, highlight limitations of existing algorithms and offer revealing insights. The code and datasets will be publicly available.

## 1 Introduction

Deep neural networks (DNNs) have achieved remarkable success in various fields such as computer vision (O'Mahony et al., 2020), natural language processing (Otter et al., 2020), and speech recognition (Nassif et al., 2019). However, studies (Szegedy et al., 2013; Goodfellow et al., 2014; Brown et al., 2017; Kurakin et al., 2018; Buckner, 2020) show that DNNs are vulnerable to adversarial attacks, which can be categorized into digital and physical attacks. Digital attacks add imperceptible perturbations to input images post-imaging, while physical attacks modify the physical properties of targets pre-imaging, such as changing textures (Suryanto et al., 2023; Zheng et al., 2024) or adding stickers (Wei et al., 2022; Li et al., 2019). Physical attacks are more practical and dangerous as they can be easily implemented in real-world scenarios, raising significant concerns in safety-critical applications like autonomous driving (Wang et al., 2023b; Cao et al., 2023), security surveillance (Nguyen et al., 2023; Wang et al., 2019b), and remote sensing (Wang et al., 2024b; Lian et al., 2022).

Object detection is a fundamental and pragmatic task in computer vision, widely deployed in various intelligent systems (Zou et al., 2023; Zhao et al., 2019). Consequently, many physical attacks aim to fool object detectors in real-world scenarios, and the physical adversarial robustness of object detection models has garnered increasing attention in recent years. However, the absence of regulated and easy-to-follow benchmarks hinders the development of physical attack and physically robust detection methods. The main reasons for the lack of physical attack benchmarks are concluded as follows: *1)* **Time-consuming and expensive**: Evaluating the performance of physical attacks and the adversarial robustness of object detection models requires numerous real-world experiments, which are time-consuming and costly. *2)* **Physical dynamics alignment**: Ensuring comparison fairness necessitates strictly controlled and consistent physical dynamics, which is unachievable in real-world scenarios since it is impossible to capture two identical pictures. *3)* **Cross-domain loss**: Physical attacks often involve creating conspicuous adversarial perturbations that must survive the transformation from the physical to the digital domain and vice versa, while this cross-domain loss is uncontrollable. *4)* **Difficulty in comparison**: With the evolution of physical attacks from 2D to 3D space, it becomes challenging to fairly compare different types of physical attack methods. Due to

these challenges, it is difficult to effectively verify the efficacy of physical attacks and the adversarial robustness of object detection models without thorough evaluation and impartial comparisons. As a result, researchers cannot accurately gauge the progress of physical adversarial attacks and robustness development, which slows down advancements in the field.

In this paper, we propose utilizing realistic simulations to benchmark physical attacks under controlled conditions such as weather, viewing angle, and location. These conditions are challenging to align for impartial comparisons in the real world. Our benchmark includes 23 physical attack methods, 48 object detectors, diverse physical dynamics, evaluation metrics from different perspectives, and comprehensive pipelines for data generation, attack and detection evaluation, and subsequent analysis. Moreover, the benchmark is highly flexible and scalable, allowing for easy integration of new physical attacks, models, and even other vision tasks. Based on the benchmark, we generate comprehensive and strictly aligned datasets and perform over 8,000 evaluations, including both overall assessments and detailed ablation studies for controlled physical dynamics. Through these experiments, we provide detailed analyses from detection and attack perspectives, highlight algorithm limitations, and convey valuable insights. In summary, our contributions are as follows:

- We propose a robust and equitable benchmark for physical attacks against object detection models. This benchmark deeply explores the potential of real-world simulators to consistently evaluate physical attacks under a variety of continuous physical dynamics.
- The benchmark includes 23 physical attacks, 48 object detectors, comprehensive physical dynamics, and rigorous evaluation metrics. We provide end-to-end pipelines for dataset generation, detection, evaluation, and analysis, ensuring a thorough evaluation process.
- The benchmark is designed to be highly flexible and scalable, facilitating the easy integration of new physical attacks, models, and even other vision tasks. This adaptability enhances the utility of our framework for ongoing research and development in the field.
- Based on our benchmark, we generate comprehensive datasets and perform over 8,000 evaluations, including overall assessments and detailed ablation studies. These experiments highlight the limitations of existing algorithms and illuminate informative insights.

## 2 RELATED WORK

### 2.1 OBJECT DETECTION

Object detection is a fundamental task in computer vision, aiming to identify and localize objects within images or videos. It can be formulated as a mapping function $f : \mathcal{X} \to \mathcal{Y}$, where $\mathcal{X}$ is the input space and $\mathcal{Y}$ is the output space (e.g., bounding boxes and class labels). Deep learning has significantly advanced object detection. R-CNN (Girshick et al., 2014) and its successors (Girshick, 2015; Ren et al., 2016; Lu et al., 2019; Pang et al., 2019; Wu et al., 2020a; Zhang et al., 2020a; Sun et al., 2021) improved detection speed and accuracy with region proposal networks and shared convolution computations. SSD (Liu et al., 2016) and YOLO series (Redmon et al., 2016; Redmon & Farhadi, 2017; 2018; Bochkovskiy et al., 2020; Jocher et al., 2022; Li et al., 2022a; Wang et al., 2023a; Jocher et al., 2023; Wang & Liao, 2024; Wang et al., 2024a) further accelerated detection by eliminating region proposals, enabling real-time applications. Recently, transformer-based architectures like DETR (Carion et al., 2020), DAB-DETR (Liu et al., 2022), ViTDet (Li et al., 2022b), DINO (Zhang et al., 2022b), and Co-DETR (Zong et al., 2023) have pushed performance boundaries using attention mechanisms. Despite these advancements, object detection in adversarial environments remains challenging, requiring ongoing research.

### 2.2 PHYSICAL ATTACK

Adversarial attacks typically add imperceptible perturbations $\boldsymbol{\delta}$ to the clean input $\boldsymbol{x}$ in the digital domain, fooling DNNs into incorrect predictions. This is formulated as: $\min_{\boldsymbol{\delta}} \mathcal{L}(f(\boldsymbol{x}+\boldsymbol{\delta}), \boldsymbol{y})$ s.t. $\boldsymbol{\delta} \in \mathcal{X}$, where $\mathcal{L}$ is the attack loss and $\boldsymbol{y}$ is the ground-truth. In contrast, physical attacks often manipulate the physical properties of objects to deceive detection models, formulated as: $\min_{\boldsymbol{\delta}} \mathcal{L}(f(\boldsymbol{x} + \mathcal{T}_{P2D}(\mathcal{T}_{D2P}(\boldsymbol{\delta}))), \boldsymbol{y})$ s.t. $\boldsymbol{\delta} \in \mathcal{X}$, where $\mathcal{T}_{D2P}$ and $\mathcal{T}_{P2D}$ are transformations between digital and physical domains. Kurakin et al. (2018) first showed that machine learning systems are vulnerable to adversarial examples in physical contexts. They demonstrated this with adversarial images captured via a cell phone camera, significantly degrading vision system performance. Brown et al. (2017) introduced adversarial patches, which localize perturbations to specific image regions without imperceptibility constraints. These patches are practical and effective in the real world, easily

printed and attached to objects to fool detectors (Song et al., 2018; Thys et al., 2019; Wu et al., 2020b; Zolfi et al., 2021; Zhu et al., 2021; Wang et al., 2022b; Zhu et al., 2022; Hu et al., 2022; Zhang et al., 2022c; Shapira et al., 2022; Huang et al., 2023; Guesmi et al., 2024). To avoid suspicion, natural-style adversarial patches have been proposed (Huang et al., 2020; Hu et al., 2021; Guesmi et al., 2023). Beyond patches, physical perturbations include light (Hu et al., 2023a; Wu et al., 2024), viewpoint (Dong et al., 2022), and 3D objects (Liu et al., 2023a). Extending adversarial perturbations to 3D space (Zhang et al., 2018; Wang et al., 2022a; Suryanto et al., 2022; 2023; Zhou et al., 2024) has proven more effective and applicable in real-world scenarios. The variety in perturbations and settings complicates fair comparisons of physical attack methods.

## 2.3 ROBUSTNESS BENCHMARK

Benchmarking adversarial attacks is crucial for evaluating and improving the robustness of DNN-based models. Croce et al. (2020) established a standardized benchmark for adversarial robustness, accurately reflecting model robustness within a reasonable computational budget. Wu et al. (2022) created a comprehensive benchmark for backdoor attacks in image classification models. Michaelis et al. (2019) provided a benchmark to assess object detection models under deteriorating image quality, such as distortions or adverse weather conditions. Zheng et al. (2023) benchmarked adversarial robustness of image classifiers in black-box settings. Dong et al. (2023) evaluated the robustness of 3D object detection to common corruptions in LiDAR and camera data. Li et al. (2023) focused on benchmarking the visual naturalness of physical adversarial perturbations. Hingun et al. (2023) constructed a large-scale benchmark for evaluating adversarial patches with a traffic sign dataset. CARLA (Dosovitskiy et al., 2017), a realistic autonomous driving simulator, has been used in physical adversarial robustness research. Nesti et al. (2022) presented CARLA-GEAR, a dataset generator for evaluating adversarial robustness of vision models. Zhang et al. (2023b) proposed a pipeline for instance-level data generation using CARLA, creating the DCI dataset and conducting experiments with three detectors and three physical attacks. Despite these efforts, a comprehensive and rigorous benchmark for physical attacks against object detection models is still lacking. This work aims to fill that gap with easy-to-follow instructions and a codebase.

## 3 PADETBENCH

The benchmark encompasses four integral facets: datasets generation, physical attacks, object detection, and comprehensive evaluation & analysis procedures, as shown in Fig. 1. From a technical standpoint, we have engineered each constituent of the benchmark as modular, end-to-end pipelines within the codebase, ensuring straightforward adoption and replication.

### 3.1 DATASETS GENERATION

It is common to use COCO (Lin et al., 2014), PASCAL VOC (Everingham & Winn, 2012), KITTI (Geiger et al., 2012), etc., as benchmark datasets for object detection. However, these datasets are ill-suited for assessing physical attacks since they are static and lack the flexibility required to create manipulated, real-world adversarial scenarios. Physical attacks typically entail altering the physical attributes of objects before capturing their images. To fairly and accurately evaluate and compare such attacks, experiments necessitate applying perturbations in real-world conditions with controlled physical dynamics, which are excessively time-consuming, labor-intensive, and theoretically infeasible. Simulated environments, like CARLA (Dosovitskiy et al., 2017), present a viable solution to these obstacles by enabling the straightforward manipulation of physical dynamics through configurable parameters.

This work contributes an end-to-end pipeline for dataset generation within our codebase, significantly streamlining the dataset generation process and enhancing research productivity. Our pipeline prioritizes user-friendliness, enabling researchers to swiftly generate datasets embodying diverse physical conditions through a concise series of steps. These conditions encompass variations in weather, viewing angles, and distances, along with the capacity to impose physical perturbations on objects. Comprehensively, our pipeline supports over 10 distinct environments ranging from downtowns to small towns and rural landscapes, coupled with a library of more than 40 vehicles and 40 pedestrian models, all customizable concerning their hues and surface textures. It further integrates continuous manipulation of physical dynamics such as fluctuating weather patterns, precise sun positioning, and flexible camera placements concerning both location and orientation (refer to A.2

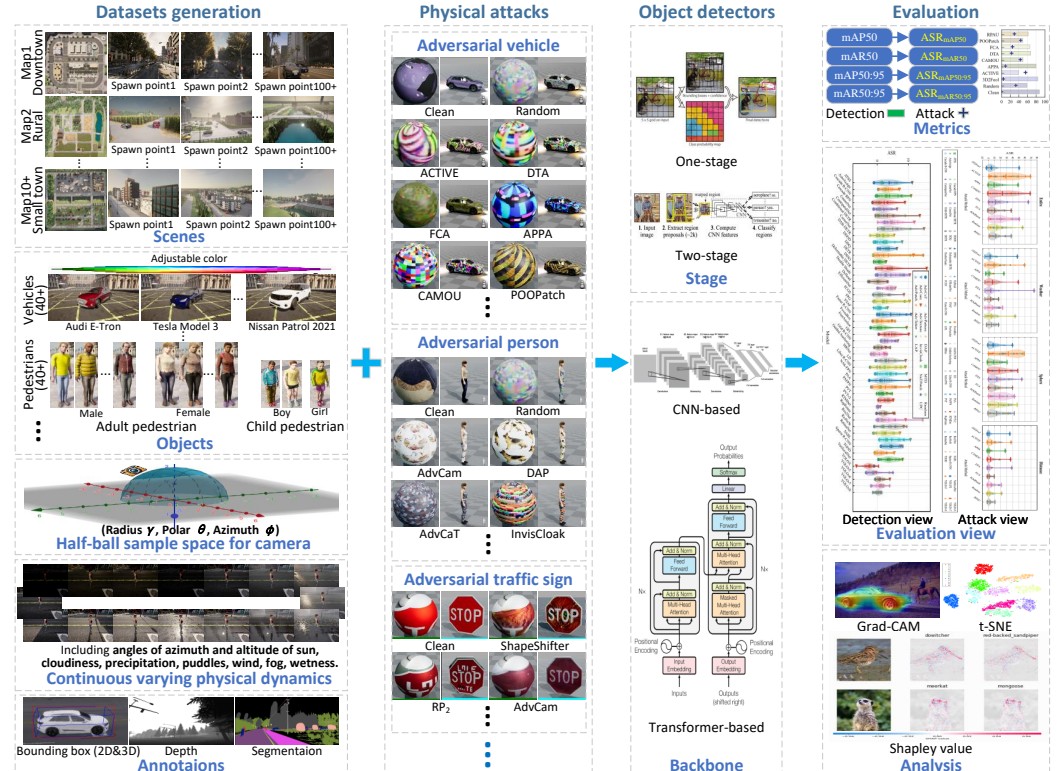

Figure 1: Overview of the benchmark, which consists of four main components: dataset generation, physical attacks, object detection, and evaluation. The end-to-end pipelines for each component are built into the codebase, making them easy to follow and reproduce. Please zoom in for details.

and A.3 for details). To ensure accessibility, we accompany the pipeline with step-by-step guidelines for personalizing object perturbations and seamlessly integrating these modifications within CARLA's (Dosovitskiy et al., 2017) simulation framework.

Our benchmark comprises three categories of datasets: a clean dataset serving as a control group, a dataset with random noise perturbations, and several datasets featuring adversarial perturbations generated through various attack methodologies. To ensure fair comparisons, scene compositions and camera perspectives are meticulously synchronized and regulated across all datasets, achievable effortlessly through our provided pipeline.

Moreover, our pipeline facilitates the automatic generation of supplementary annotations, including 2D and 3D bounding boxes, depth maps, and instance segmentation maps. Consequently, our benchmark extends its utility beyond 2D object detection, also catering to tasks like 3D object detection, instance segmentation, depth estimation, and more, thereby enhancing the scope of research and application in computer vision.

## 3.2 PHYSICAL ATTACKS

Physical attacks are usually tailed for specific object, and the commonly targeted objects are vehicles, persons, and traffic signs as evidenced by Wei et al. (2024). Consequently, we adopt typical objects from these categories as examples to illustrate the proposed benchmark. Specifically, we select 23 representative physical attack methods, which can be categorized into three types according to their target objects: vehicle, person, and traffic sign, as shown in Table 1. The corresponding physical perturbations of these methods are imported into Unreal Engine 4 for CARLA (CarlaUE4) (Dosovitskiy et al., 2017), as shown in the physical attacks part of Fig. 1, to generate the physical adversarial datasets. We adhere to two principles similar to (Wu et al., 2022) when selecting physical attacks. First, the methods are representative or advanced in the research field, which can serve as baseline and state-of-the-art (SOTA) methods for comparison, respectively. Second, physical attacks

are easily conducted and with reproducible performance, which can be conveniently followed and reproduced by other researchers. Since our benchmark evaluates physical attacks based on their crafted perturbations, novel physical attack methods can be easily integrated into the benchmark by following the provided pipeline. We will continue to update the physical attacks in the benchmark to keep pace with the latest research progress.

Table 1: Categorization of physical attack methods based on their target objects.

| Target objects | Physical attacks |
|---|---|
| Vehilcle | FCA (Wang et al., 2022a), DTA (Suryanto et al., 2022), ACTIVE (Suryanto et al., 2023), 3D²Fool (Zheng et al., 2024), POOPatch (Cheng et al., 2022), RPAU (Liu et al., 2023b), CAMOU (Zhang et al., 2018) |
| Person | DAP (Guesmi et al., 2024), AdvPattern (Wang et al., 2019b), UPC (Huang et al., 2020), NatPatch (Hu et al., 2021), MTD (Ding et al., 2021), AdvCaT (Hu et al., 2023b), AdvTexture (Hu et al., 2022), AdvTshirt(Xu et al., 2020), AdvPatch (Thys et al., 2019), LAP (Tan et al., 2021), InvisCloak (Wu et al., 2020b), AdvCam (Duan et al., 2020) |
| Traffic sign | AdvCam (Duan et al., 2020), RP₂ Eykholt et al. (2018), ShapeShifter(Chen et al., 2019b) |

### 3.3 OBJECT DETECTORS

We choose 48 object detectors in the same principles as choosing physical attack methods, covering mainstream object detectors, such as YOLO series (Jocher et al., 2022; Li et al., 2022a; Wang et al., 2023a; Jocher et al., 2023; Ge et al., 2021) (One-stage) and R-CNN series (Girshick et al., 2014; Girshick, 2015; Ren et al., 2016; Cai & Vasconcelos, 2018; Sun et al., 2021), which are based on CNN. Except for canonical detectors, we also include transformer-based detectors, such as DETR (Carion et al., 2020), Conditional DETR (Meng et al., 2021), Deformable DETR (Zhu et al., 2020b), DAB-DETR (Liu et al., 2022), and DINO (Zhang et al., 2022b). All the selected detectors are listed in Table 2 according to their characteristics. Our benchmark provides the end-to-end pipeline for object detection evaluation based on MMDetection (Chen et al., 2019a). Consequently, it is convenient to integrate new detectors into the benchmark, and the benchmark can also be easily extended to evaluate other vision tasks, such as 3D object detection, instance segmentation, and depth estimation.

### 3.4 EVALUATION AND ANALYSIS

**Evaluation metrics**. To rigorously assess the efficacy of physical attacks on object detection systems, we furnish baseline datasets: clean datasets (without perturbations) and those infused with randomized noise (incorporating arbitrary disturbances in $\ell_\infty$-bounded space). This dual-baseline approach sets the stage for a thorough and fair examination. Quantifying performance entails employing evaluation metrics that consider the performance of both object detection and adversarial attack. These metrics comprise several widely adopted indicators, including mean average precision (mAP), mean average recall (mAR), and attack successful rate (ASR). mAP and mAR are calculated as the mean value of average precisions and recalls at $n$ recall and precision levels over $C$ classes, respectively, i.e., $\text{mAP} = \frac{1}{C}\sum_{c=1}^{C}(\frac{1}{n}\sum_{i=1}^{n} P_i)$ and $\text{mAR} = \frac{1}{C}\sum_{c=1}^{C}(\frac{1}{n}\sum_{i=1}^{n} R_i)$. Precision rate and recall rate are calculated as $P = \frac{\text{TP}}{\text{TP+FP}}$ and $R = \frac{\text{TP}}{\text{TP+FN}}$, respectively, where TP, FP, and FN denote the true positive, false positive, and false negative counts of the detector, respectively. On the other hand, ASR quantifies the effectiveness of the adversarial perturbations, calculated as $\text{ASR} = 1 - \frac{\text{M}_{\text{attack}}}{\text{M}_{\text{clean}}}$, where $\text{M}_{\text{attack}}$ and $\text{M}_{\text{clean}}$ denote the value of adopted metric on the attack and clean datasets, respectively. ASR provides a direct measure of the extent to which the attacks undermine the detector's performance.

**Advocation of mAR for physical attacks**. Adversarial attacks aim to induce mispredictions, i.e., to maximize error rate, which is the mathematical expectation of incorrect predictions written as:

$$\text{err} = \mathbb{E}_{y \in Y}[1_{\hat{y} \neq y}] = \frac{|Y - Y \cap \hat{Y}|}{|Y|} \tag{1}$$

where $1_{\hat{y}=y}$ is 1 for a correct prediction and 0 otherwise, and $Y$ and $\hat{Y}$ represent the ground truths and predicted results of all objects, respectively. According to the calculation of performance metrics

Table 2: Categorization of object detection. Note that the categorization is based on the selected version of the methods, and the category may vary with different versions, such as the backbone of a detector being either CNN or Transformer. Refer to A.4 for the corresponding config files.

| Backbone | Category | Detectors |
|---|---|---|
| CNN | One-stage | ATSS(Zhang et al., 2020b), AutoAssign(Zhu et al., 2020a), GFL(Li et al., 2020), CenterNet(Zhou et al., 2019), CornerNet(Law & Deng, 2018), PAA(Kim & Lee, 2020), DDOD(Chen et al., 2021), DyHead(Wu et al., 2020a), EfficientNet(Tan & Le, 2019), FCOS(Tian et al., 1904), FoveaBox(Kong et al., 2020), FreeAnchor(Zhang et al., 2019), LD(Zheng et al., 2022), CentripetalNet(Dong et al., 2020), FSAF(Zhu et al., 2019), RTMDet(Lyu et al., 2022), TOOD(Feng et al., 2021), VarifocalNet(Zhang et al., 2021), YOLOX(Ge et al., 2021), YOLOv5(Jocher et al., 2022), YOLOv6(Li et al., 2022a), YOLOv7(Wang et al., 2023a), RetinaNet(Lin et al., 2017), YOLOv8(Jocher et al., 2023) |
| | Two-stage | Faster R-CNN(Ren et al., 2016), Cascade R-CNN(Cai & Vasconcelos, 2019), Cascade RPN(Vu et al., 2019), Double Heads(Wu et al., 2020a), FPG(Chen et al., 2020), Libra R-CNN(Pang et al., 2019), PAFPN(Liu et al., 2018), HRNet(Sun et al., 2019), ResNeSt(Zhang et al., 2022a), Res2Net(Gao et al., 2019), SABL(Wang et al., 2020), Guided Anchoring(Wang et al., 2019a), Sparse R-CNN(Sun et al., 2021), RepPoints(Yang et al., 2019), Grid R-CNN(Lu et al., 2019) |
| Transformer | - | DETR(Carion et al., 2020), PVT(Wang et al., 2021), PVTv2(Wang et al., 2021), DDQ(Zhang et al., 2023a), DAB-DETR(Liu et al., 2022), DINO(Zhang et al., 2022b), Deformable DETR(Zhu et al., 2020b), Conditional DETR(Meng et al., 2021) |

for detection, we can rewrite the error rate as:

$$\text{err} = \frac{|Y - Y \cap \hat{Y}|}{|Y|} = \mathbb{E}\Big[\frac{\text{FN}}{\text{TP} + \text{FN}}\Big] = 1 - \text{mAR}. \tag{2}$$

Therefore, mAR is a more direct and intuitive metric for evaluating the effectiveness of physical attacks on object detection models. We use mAR as the primary metric in the main manuscript, while mAP is also provided for reference.

**Evaluation perspectives**. Specifically, we use mAP50, i.e., the confidence threshold of 0.5, to evaluate the overall performance of object detection, which is widely adopted in the object detection community. mAR50 is adopted to signify the proportion of correctly identified instances relative to the actual total in the dataset, offering an intuitive gauge of how physical attacks degrade the detection capability of a given adversarial target. However, mAR50 and mAP50 cannot fully reflect the performance of object detection models, especially when the confidence score of a adversarial object is significantly dropped but still higher than the threshold. To address this issue, we also use mAR50:95 and mAP50:95, which are calculated as the mean value over the range of 0.5 to 0.95 of the confidence threshold, to provide a more comprehensive evaluation of the object detection models. In the perspective of physical attacks, we use ASR over the detection metrics mAP50, mAR50, mAP50:95, and mAR50:95 to evaluate the effectiveness of physical attacks on object detection models, ensuring a comprehensive and impartial assessment. Moreover, we also visualize the distribution of evaluation performance using violin plots, which can provide a more intuitive understanding of the performance of object detection models and physical attacks, respectively.

**Analysis tools**. Furthermore, we enhance our codebase by incorporating several ready-to-use explainability visualization tools, facilitating deeper insights into model behavior. These include Grad-CAM (Selvaraju et al., 2017) for visualizing the regions of input data that contribute most to the model's prediction, Shapley value (Lundberg & Lee, 2017) to quantify the individual feature contributions, and t-SNE (van der Maaten & Hinton, 2008) for reducing dimensionality and visualizing high-dimensional data in a more interpretable manner. These additions empower users to conduct comprehensive analyses beyond mere performance evaluation.

## 4 EXPERIMENTS

### 4.1 EXPERIMENTAL SETUP

**Datasets**. *1)* **Overall experiments**. We generate overall datasets with 3 objects, 10 weather conditions, 2 altitude angles, 8 azimuth angles, 5 radius values, 3 spawn points, and 23 physical perturbations, i.e., 7200 samples ($3 \times 10 \times 2 \times 8 \times 5 \times 3 = 7200$) for each attack method, in which the physical

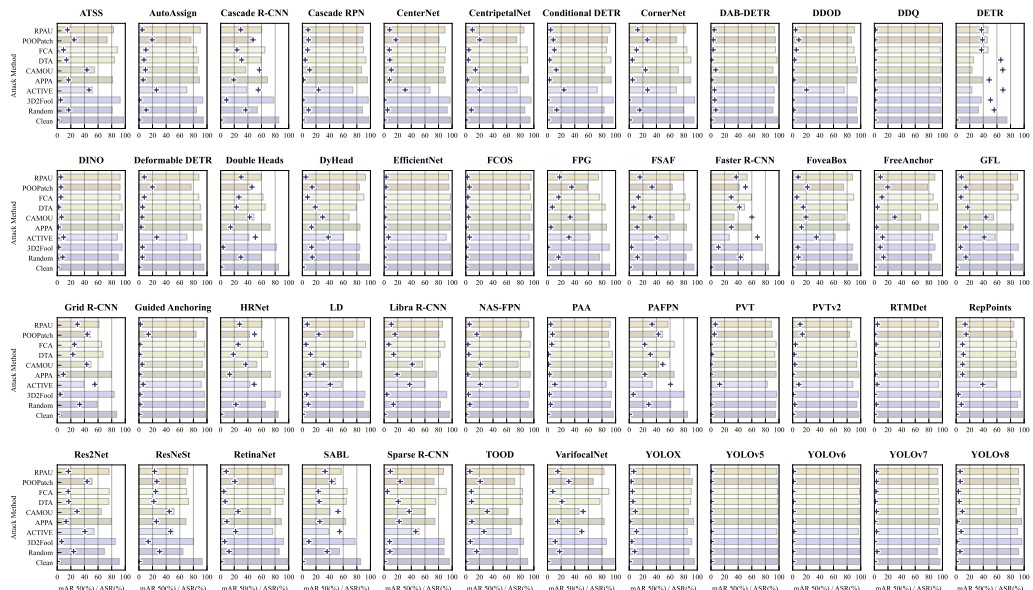

Figure 2: **Overall** results of **vehicle** detection. Each subplot corresponds to a specific detector, illustrating its mAR50 (%) under various attack techniques and control group (Clean) via bar graphs, with + markers denoting the associated ASR (%) values. Zooming in is advised.

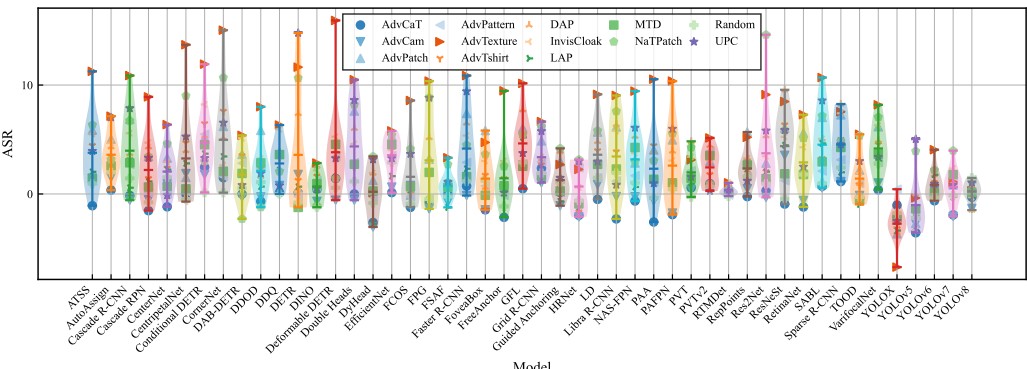

Figure 3: **Overall** results of **person** detection by 48 detectors, reported in **ASR(%)**. Each detector is evaluated against 13 attack methods (marked by different markers and colors, see legend). The violin plot shows the maximum, minimum, and distribution of ASR, where thickness represents the density of attack methods with corresponding ASR. ASR is measured by mAR50.

dynamics are strictly aligned and controlled for impartial comparison (detailed in A.2). Please note that these parameters are adjustable in the pipeline, and the datasets can be easily generated with different settings as needed. *2)* **Ablation Studies**. We conduct in-depth examinations to explore the individual impact of core physical dynamics: weather conditions, venue, camera distance, azimuth angle, altitude angle within a hemispherical space. Accomplishing this involves generating focused sub-benchmarks, each consisting of 100 samples.

**Physical attacks**. We generate 24 datasets for comprehensive evaluation, including 20 physically noised datasets that correspond to 20 physical attacks, an extra 2 clean datasets and 2 randomly noised datasets for comparison of vehicle detection and person detection, respectively. To evaluate the attack transferability, we also adopt perturbations optimized for aerial detection (Lian et al., 2022) and depth estimation (Zheng et al., 2024; Cheng et al., 2022) in the experiments. Furthermore, we generate 4

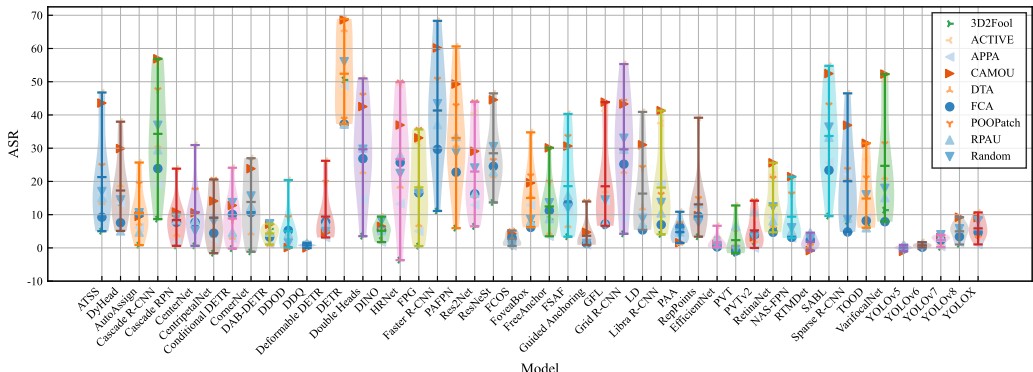

Figure 4: **Overall** results of **vehicle** detection by 48 detectors, reported in **ASR(%)**. Each detector is evaluated against 9 attack methods (marked by different markers and colors, see legend). The violin plot shows the maximum, minimum, and distribution of ASR, where thickness represents the density of attack methods with corresponding ASR. ASR is measured by mAR50.

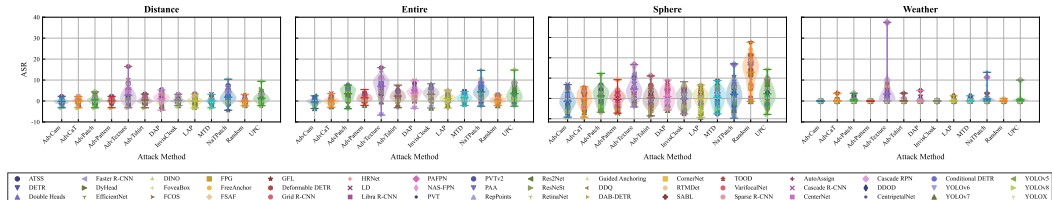

Figure 5: Results of **person** detection from 13 **attack methods** in **ASR (%)**. Each method is evaluated against 48 detectors (marked by different markers and colors, see legend). The violin plot shows the maximum, minimum, and distribution of ASR, where thickness represents the density of detectors with corresponding ASR. ASR is measured by mAR50.

extra datasets concerning traffic sign detection to show the easy extension of the benchmark to other objects (refer to A.3 for more details). The involved physical attacks are detailed in Table 1.

**Object detectors**. We evaluate 48 object detectors covering mainstream types, such as one and two-stage detectors, and transformer-based detectors, as shown in Table 2, by integrating MMDetection (Chen et al., 2019a) into our evaluation pipeline.

Therefore, we conduct a total of 8256 ($24 \times 48 \times (1 + 6) + 4 \times 48$) groups of the experiment, which are conducted with $16 \times$ NVIDIA Geforce 4090.

## 4.2 OVERALL EXPERIMENTS AND ANALYSIS

We present the comprehensive results of vehicle detection against physical attacks in Fig. 2. Additionally, Fig. 3 and Fig. 4 show visualized analyses of the experimental results from detection perspectives, and Fig. 5 and Fig. 6 present the results from attack perspectives. More experimental results and corresponding detailed numerical results are listed in B. From these evaluation, several key observations emerge:

**Detection perspective**. *1)* Vehicle detection performance is significantly impacted by physical attacks, with the average recall rates of detectors decreasing up to 50%, as shown in Fig. 4. However, pedestrian detection performance is less affected regarding various attacks, with the average recall rates of detectors decreasing by less than 20%, as shown in Fig. 3. The potential reason is that the stronger physical perturbations are optimized with consideration of 3D space and accommodate more complex physical dynamics, while physical attacks aiming to fool person detectors are commonly performed with optimized 2D patches, which work well in particular physical dynamics, as detailed in the ablation experiments B.2.2, which empirically demonstrate the pressing need and necessity of a comprehensive and rigorous benchmark for physical attacks. *2)* The performance of different detectors varies significantly, with some detectors exhibiting superior robustness against physical

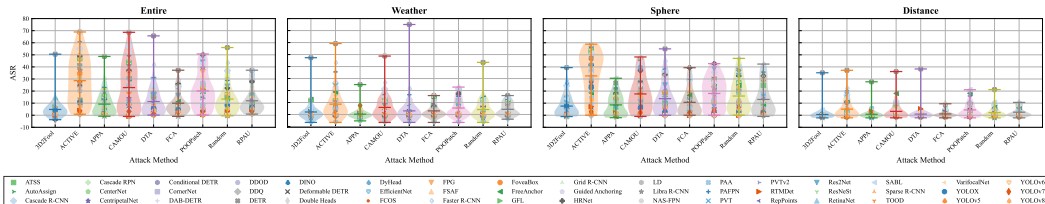

Figure 6: Results of **vehicle** detection from 9 **attack methods** in **ASR (%)**. Each method is evaluated against 48 detectors (marked by different markers and colors, see legend). The violin plot shows the maximum, minimum, and distribution of ASR, where thickness represents the density of detectors with corresponding ASR. ASR is measured by mAR50.

attacks, such as EfficientNet, the YOLO series, and RTMDet among one-stage detectors. Additionally, DDQ demonstrates notable adversarial robustness among transformer-based detectors. While other detectors show varying lower levels of robustness, state-of-the-art detection performance does not necessarily correlate with adversarial robustness. Consequently, the benchmark also serves as an indicator of robustness.

**Attack perspective**. *1)* For vehicle detection, different physical attacks exhibit varying levels of effectiveness, with some attacks achieving ASR values exceeding 70% like ACTIVE, and others failing to surpass 20%. Most of the physical attacks hard to fool the latest SOTA detectors, such as EfficientNet, YOLO series, and RTMDet. This phenomenon is caused by the victim models of the attack method lagging behind the development of the detection method, which also motivates us to fill this gap. *2)* For person detection, the ASR values of physical attacks are generally lower than those for vehicle detection, with the majority of attacks achieving ASR values below 20%. The relatively strongest attack method is AdvTexture, which elaborates on a 2D patch but with tricks for 3D space. This also demonstrates the gap between 2D perturbations and 3D physical space, highlighting the challenges in effectively transferring adversarial attacks from controlled 2D environments to more complex 3D scenarios. Moreover, it underscores the necessity for developing more sophisticated attack strategies that can account for the intricacies of 3D physical dynamics.

### 4.3 Ablation experiments and analysis

Except for the overall experiments, we also conduct ablation experiments to investigate the impact of physical world factors. We show the results of 3 physical dynamics, including weather, distance, and camera viewing angle, in Fig. 5 and Fig. 6, respectively. More experiments on other dynamics are provided in B.3 and B.5. From these evaluation, several key observations emerge: *1)* Physical attack performance can be easily swayed by physical dynamics. This phenomenon is consistent with existing works (Dong et al., 2022; Zhong et al., 2022) and emphasizes the importance of strictly aligning physical dynamics when evaluating physical attacks, which are often underestimated by previous works. *2)* We also observe a gap between the ablation attack performance of our benchmark and the reported performance in the original papers (refer to B.2.1 for more details). Two reasons may contribute to this gap: the first is the adopted SOTA detectors in our benchmark, which are more robust than the victim models in the original papers, and the second is that our benchmark provides more comprehensive and strict evaluation datasets and physical dynamics, which are more challenging for the attack methods. These observations empirically demonstrate the pressing need and necessity of a comprehensive and rigorous benchmark for physical attacks. Please refer to B for more experiments, detailed analysis and discussion.

## 5 Discussion

### 5.1 Where are we?

**Lack of alignment and comprehensiveness in physical dynamics**. Existing works are either limited in comprehensiveness or do not strictly align and control physical dynamics, as illustrated in A.2. As evidenced by previous works (Zhong et al., 2022; Dong et al., 2022), physical dynamics can be exploited to fool DNNs, underscoring the necessity of aligning these dynamics. Consequently, researchers cannot accurately gauge the actual progress of this research domain without a comprehensive and rigorously aligned study, which slows down advancements in the field.

**Discrete and naive physical adaptation**. While theoretically, well-studied digital attacks should benefit physical attacks, the reality often falls short. This discrepancy arises because the theoretical gains cannot survive cross-domain transformations ($\mathcal{T}_{P2D}(\mathcal{T}_{D2P}(\boldsymbol{\delta}))$) as mentioned in 2.2. Existing works use discrete and naive augmentations to model physical dynamics, failing to capture the characteristics of continuous and complex physical scenarios. This explains the gap observed in our ablation experiments (B.2.1), highlighting the need for a comprehensive and rigorous benchmark.

## 5.2 WHERE TO GO?

**Comprehensive and physically aligned benchmark**. A comprehensive and physically aligned benchmark is essential for evaluating physical attacks on object detection models. It ensures rigorous and unbiased assessments, highlighting the strengths and weaknesses of various attacks and detectors, and providing valuable insights for future research. Such a benchmark can drive the development of more robust and resilient object detection models, ultimately enhancing the security and reliability of AI systems in real-world applications.

**Rigorous and differentiable modeling of cross-domain transformations**. Accurate modeling of cross-domain transformations is essential for both physical attacks and defenses. While existing works have attempted to use differentiable neural renderers to automatically generate adversarial examples, they often have limited modeling capabilities and fall short in aligning physical factors between physical perturbations and clean images. With the advent of large foundation models, exploring how to model physical dynamics more rigorously and differentiably using large-scale data and foundation models is a promising direction.

## 6 CONCLUSION

In conclusion, we develop a comprehensive simulation-based benchmark to rigorously evaluate physical attacks under controlled conditions. This benchmark includes 23 physical attacks, 48 object detectors, and detailed physical dynamics, supported by end-to-end pipelines. The benchmark is flexible and scalable, allowing easy integration of new attacks, models, and vision tasks. Through extensive evaluations involving over 8,000 tests, we highlight algorithm limitations and provide valuable insights. We believe this benchmark will significantly advance research in physical adversarial attacks, fostering the development of more robust and reliable models.

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

**Supplemental material Contents**:

## A  ADDITIONAL CONTENT OF THE BENCHMARK

### A.1  MINI-TEST

We kindly invite the reviewers and readers to participate in a mini-test to discriminate the real-world images and the simulated images as shown in Fig. 7, the answer is revealed in its caption.

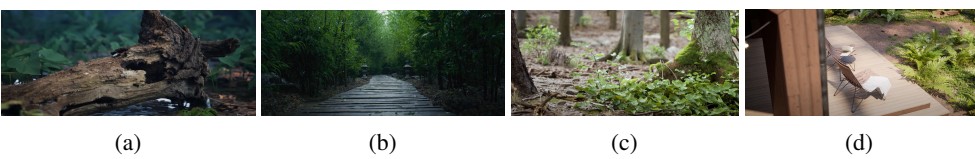

|           (a)           |           (b)           |           (c)           |           (d)           |

Figure 7: Which are simulated images? Surprisingly, they were all generated by Unreal Engine, a popular game engine. The visual quality of the simulated images is so high that it is hard to find any deficiencies. This mini-test demonstrates the potential of the simulated environment in the research field.

Table 3: The selected detectors and their corresponding config files.

| Town | Description |
|------|-------------|
| Town1 | A small, simple town with a river and several bridges. |
| Town2 | A small simple town with a mixture of residential and commercial buildings. |
| Town3 | A larger, urban map with a roundabout and large junctions. |
| Town4 | A small town embedded in the mountains with a special "figure of 8" infinite highway. |
| Town5 | Squared-grid town with cross junctions and a bridge. It has multiple lanes per direction. Useful to perform lane changes. |
| Town6 | Long many lane highways with many highway entrances and exits. It also has a Michigan left. |
| Town7 | A rural environment with narrow roads, corn, barns and hardly any traffic lights. |
| Town8 | Secret "unseen" town used for the Leaderboard challenge. |
| Town9 | Secret "unseen" town used for the Leaderboard challenge. |
| Town10 | A downtown urban environment with skyscrapers, residential buildings and an ocean promenade. |
| Town11 | A Large Map that is undecorated. Serves as a proof of concept for the Large Maps feature. |
| Town12 | A Large Map with numerous different regions, including high-rise, residential and rural environments. |

Full list of the optional maps, where Town8 and Town9 are unseen for competition. Please refer to CARLA (Dosovitskiy et al., 2017) documentary for more details.

### A.2  PHYSICAL DYNAMICS ALIGNMENT

We provide a detailed illustration of the physical dynamics alignment in Fig. 8 and Fig. 9. Specifically, it is observed from Fig. 8 that the imaging settings and lighting conditions are not strictly aligned in the comparison experiments, such as the different view angles and shadows, which have been demonstrated to have a significant impact on fooling deep neural networks (Zhong et al., 2022; Dong et al., 2022). To address this issue, we align the physical dynamics in the benchmark, as shown in Fig. 9, where the physical dynamics are strictly controlled and aligned, ensuring a fair and impartial comparison. Moreover, we also provide a detailed illustration of the physical dynamics in Fig. 10, which includes the weather conditions, camera settings, and lighting conditions. The lighting conditions varying similar to the real-world as shown in Fig. 11, such as the sun positions of 24 hours, the intensity of the light, and the shadow, which are strictly controlled and aligned in the benchmark.

### A.3  THE ADAPTABILITY OF THE BENCHMARK

We provide a detailed illustration of the scene diversity of the benchmark in Table 3 and Fig. 12, where the optional maps are listed with their descriptions. In addition, we display the extendable vehicles, pedestrians, and traffic signs in Fig. 13, Fig. 14, and Fig. 15, respectively, which can be easily extended to evaluate other objects in the benchmark. The users are also allowed to export any

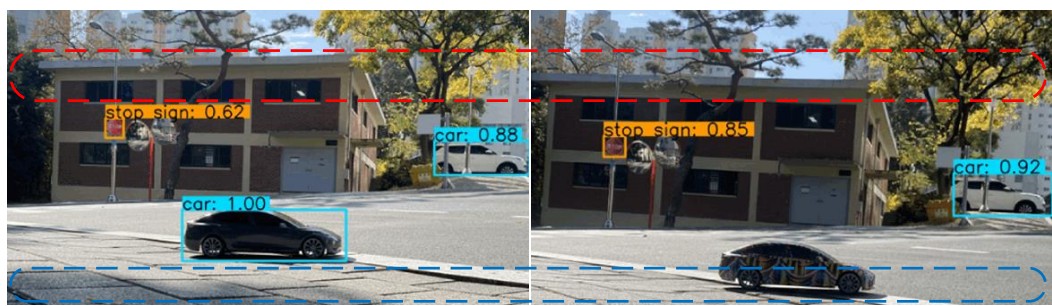

Figure 8: Illustration of the physical dynamic discrepancies. It is observed that the imaging settings and lighting conditions are not stricly aligned in the comparison experiments, such as the different view angles (red dash-line box) and shadows (blue dash-line box), which have been demonstrated to have a significant impact on fooling deep neural networks (Zhong et al., 2022; Dong et al., 2022).

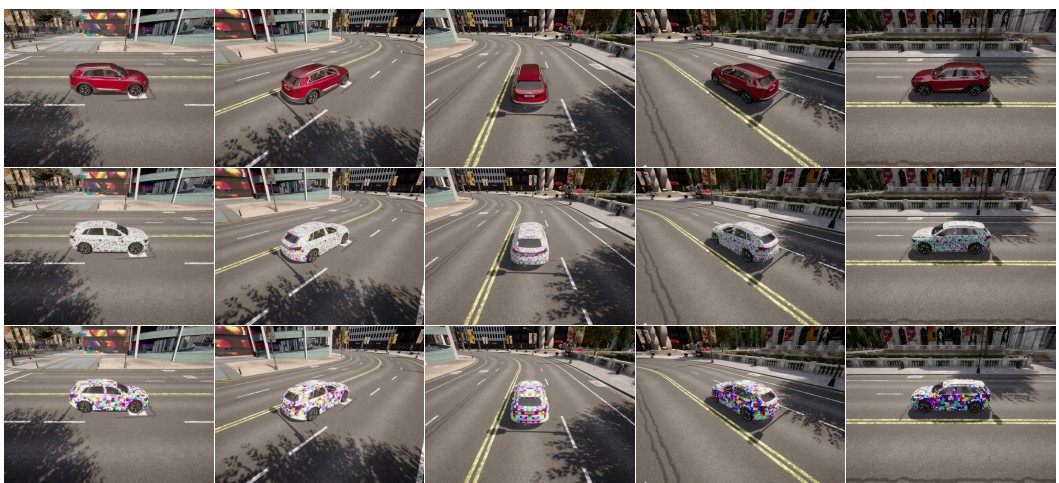

Figure 9: Illustration of the aligned physical dynamics. It is observed that the physical dynamics are strictly controlled and aligned, ensuring a fair and impartial comparison.

Figure 10: Illustration of the physical dynamics.

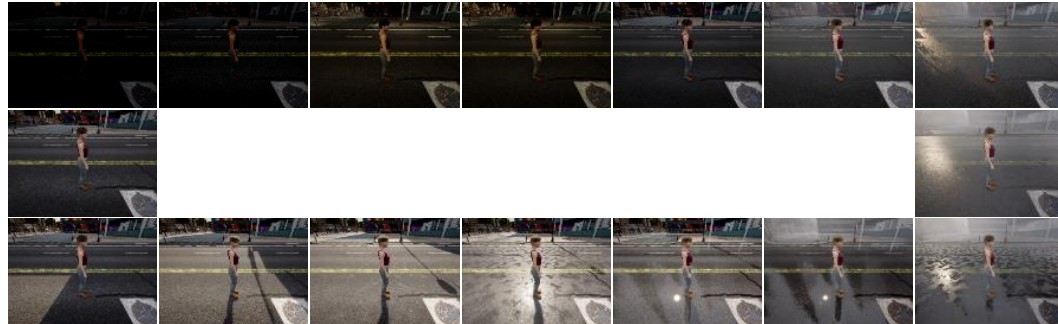

Figure 11: Illustration of the lighting conditions varying with sun positions similar to real-world laws.

customized scenes and objects to the benchmark as needed, which can be easily integrated into the benchmark.

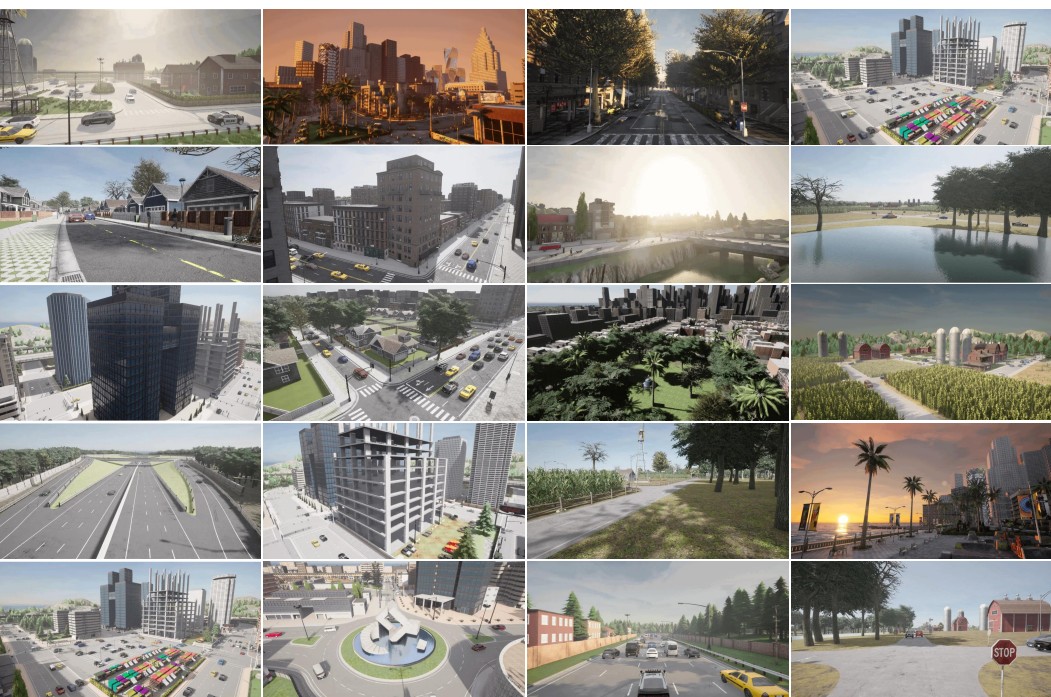

Figure 12: Illustration of the extentable scenes of the benchmark.

## A.4 CORRESPONDING CONFIG FILES OF THE SELECTED DETECTORS

The corresponding config files of the selected detectors are listed in Table 4. Specifically, 1-25 and 26-40 are CNN-based One-stage and Two-stage object detectors, respectively. 41-48 are Transformer-based object detectors. The corresponding config files of the detectors are available in our codebase or MMDetection (Chen et al., 2019a) toolbox.

## A.5 EXPLANATION ABOUT THE SELECTED OBJECTS

According to a survey (Wei et al., 2024) published in TPAMI 2024, most physical attacks against object detection are optimized for specific target categories, such as vehicles, persons, and a few for traffic signs. In line with this, we have chosen vehicles and pedestrians as the representative target categories, to evaluate the robustness of object detectors against physical attacks. .

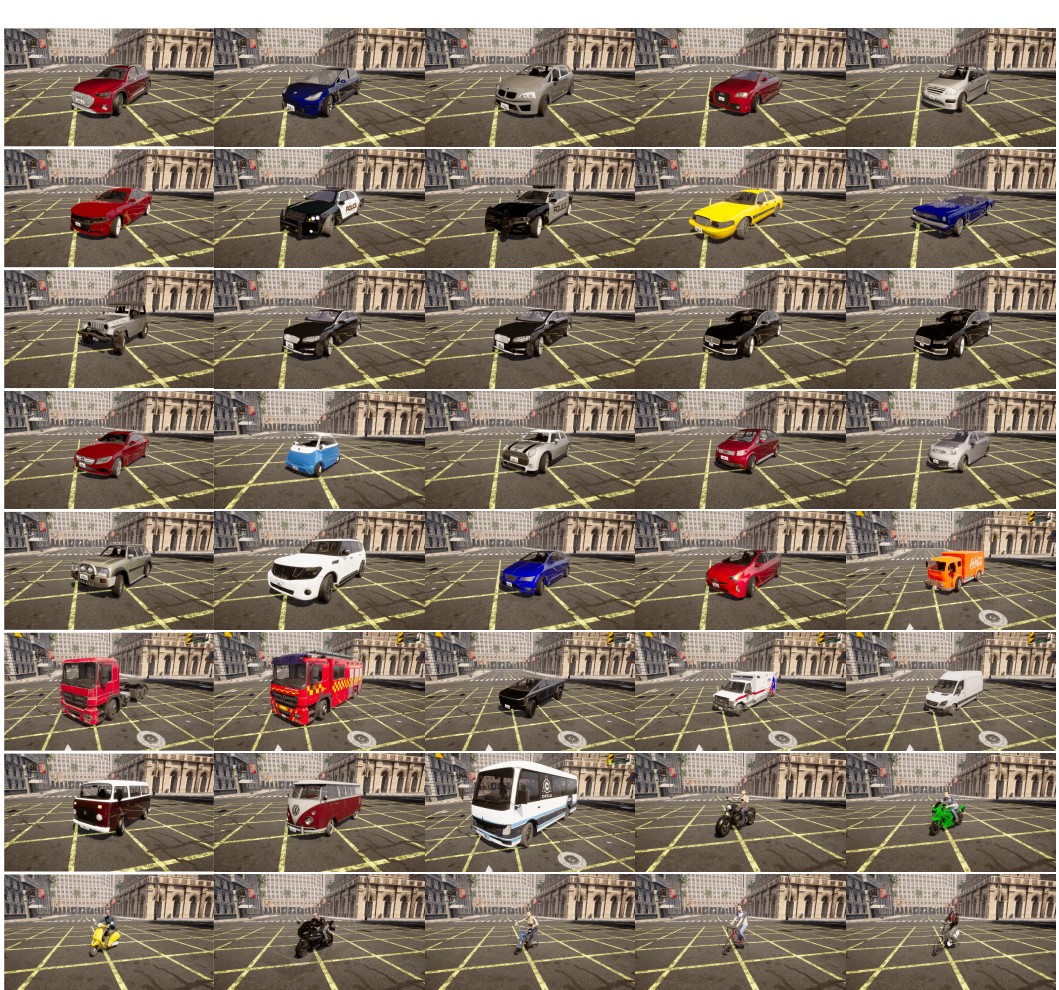

Figure 13: Illustration of the extentable vehicles of the benchmark.

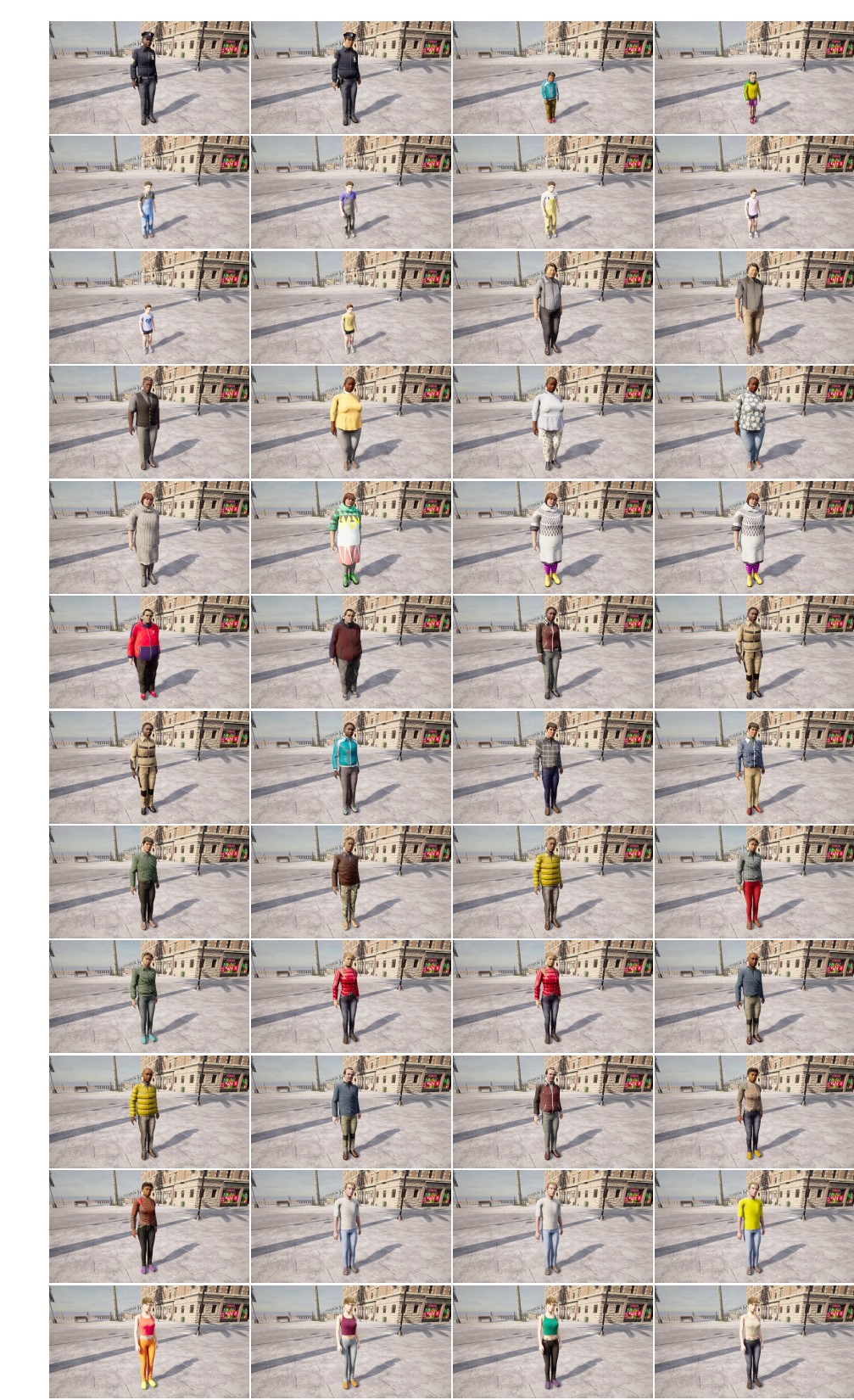

Figure 14: Illustration of the extentable walkers of the benchmark.

Table 4: The selected detectors and their corresponding config files. 1-25 and 26-40 are CNN-based One-stage and Two-stage object detectors, respectively. 41-48 are Transformer-based object detectors. The corresponding config files of the detectors are available in our codebase or MMDetection (Chen et al., 2019a) toolbox.

| Number | Config Files | Detectors |
|---|---|---|
| 1 | atss_r50_fpn_1x_coco | ATSS(Zhang et al., 2020b) |
| 2 | autoassign_r50-caffe_fpn_1x_coco | AutoAssign(Zhu et al., 2020a) |
| 3 | centernet-update_r50-caffe_fpn_ms-1x_coco | CenterNet(Zhou et al., 2019) |
| 4 | centripetalnet_hourglass104_16xb6-crop511-210e-mstest_coco | CentripetalNet(Dong et al., 2020) |
| 5 | cornernet_hourglass104_10xb5-crop511-210e-mstest_coco | CornerNet(Law & Deng, 2018) |
| 6 | ddod_r50_fpn_1x_coco | DDOD(Chen et al., 2021) |
| 7 | atss_r50_fpn_dyhead_1x_coco | DyHead(Wu et al., 2020a) |
| 8 | retinanet_effb3_fpn_8xb4-crop896-1x_coco | EfficientNet(Tan & Le, 2019) |
| 9 | fcos_x101-64x4d_fpn_gn-head_ms-640-800-2x_coco | FCOS(Tian et al., 1904) |
| 10 | fovea_r50_fpn_4xb4-1x_coco | FoveaBox(Kong et al., 2020) |
| 11 | freeanchor_r50_fpn_1x_coco | FreeAnchor(Zhang et al., 2019) |
| 12 | fsaf_r50_fpn_1x_coco | FSAF(Zhu et al., 2019) |
| 13 | gfl_r50_fpn_1x_coco | GFL(Li et al., 2020) |
| 14 | ld_r50-gflv1-r101_fpn_1x_coco | LD(Zheng et al., 2022) |
| 15 | retinanet_r50_nasfpn_crop640-50e_coco | NAS-FPN(Ghiasi et al., 2019) |
| 16 | paa_r50_fpn_1x_coco | PAA(Kim & Lee, 2020) |
| 17 | retinanet_r50_fpn_1x_coco | RetinaNet(Lin et al., 2017) |
| 18 | rtmdet_s_8xb32-300e_coco | RTMDet(Lyu et al., 2022) |
| 19 | tood_r50_fpn_1x_coco | TOOD(Feng et al., 2021) |
| 20 | vfnet_r50_fpn_1x_coco | VarifocalNet(Zhang et al., 2021) |
| 21 | yolov5_l-p6-v62_syncbn_fast_8xb16-300e_coco | YOLOv5(Jocher et al., 2022) |
| 22 | yolov6_l_syncbn_fast_8xb32-300e_coco | YOLOv6(Li et al., 2022a) |
| 23 | yolov7_l_syncbn_fast_8x16b-300e_coco | YOLOv7(Wang et al., 2023a) |
| 24 | yolov8_l_syncbn_fast_8xb16-500e_coco | YOLOv8(Jocher et al., 2023) |
| 25 | yolox_l_fast_8xb8-300e_coco | YOLOX(Ge et al., 2021) |
| 26 | faster-rcnn_r50_fpn_1x_coco | Faster R-CNN(Ren et al., 2016) |
| 27 | cascade-rcnn_r50_fpn_1x_coco | Cascade R-CNN(Cai & Vasconcelos, 2019) |
| 28 | cascade-rpn_faster-rcnn_r50-caffe_fpn_1x_coco | Cascade RPN(Vu et al., 2019) |
| 29 | dh-faster-rcnn_r50_fpn_1x_coco | Double Heads(Wu et al., 2020a) |
| 30 | faster-rcnn_r50_fpg_crop640-50e_coco | FPG(Chen et al., 2020) |
| 31 | grid-rcnn_r50_fpn_gn-head_2x_coco | Grid R-CNN(Lu et al., 2019) |
| 32 | ga-faster-rcnn_x101-32x4d_fpn_1x_coco | Guided Anchoring(Wang et al., 2019a) |
| 33 | faster-rcnn_hrnetv2p-w18-1x_coco | HRNet(Sun et al., 2019) |
| 34 | libra-retinanet_r50_fpn_1x_coco | Libra R-CNN(Pang et al., 2019) |
| 35 | faster-rcnn_r50_pafpn_1x_coco | PAFPN(Liu et al., 2018) |
| 36 | reppoints-moment_r50_fpn_1x_coco | RepPoints(Yang et al., 2019) |
| 37 | faster-rcnn_res2net-101_fpn_2x_coco | Res2Net(Gao et al., 2019) |
| 38 | faster-rcnn_s50_fpn_syncbn-backbone+head_ms-range-1x_coco | ResNeSt(Zhang et al., 2022a) |
| 39 | sabl-faster-rcnn_r50_fpn_1x_coco | SABL(Wang et al., 2020) |
| 40 | sparse-rcnn_r50_fpn_1x_coco | Sparse R-CNN(Sun et al., 2021) |
| 41 | detr_r50_8xb2-150e_coco | DETR(Carion et al., 2020) |
| 42 | conditional-detr_r50_8xb2-50e_coco | Conditional DETR(Meng et al., 2021) |
| 43 | ddq-detr-4scale_r50_8xb2-12e_coco | DDQ(Zhang et al., 2023a) |
| 44 | dab-detr_r50_8xb2-50e_coco | DAB-DETR(Liu et al., 2022) |
| 45 | deformable-detr_r50_16xb2-50e_coco | Deformable DETR(Zhu et al., 2020b) |
| 46 | dino-4scale_r50_8xb2-12e_coco | DINO(Zhang et al., 2022b) |
| 47 | retinanet_pvt-t_fpn_1x_coco | PVT(Wang et al., 2021) |
| 48 | retinanet_pvtv2-b0_fpn_1x_coco | PVTv2(Wang et al., 2021) |

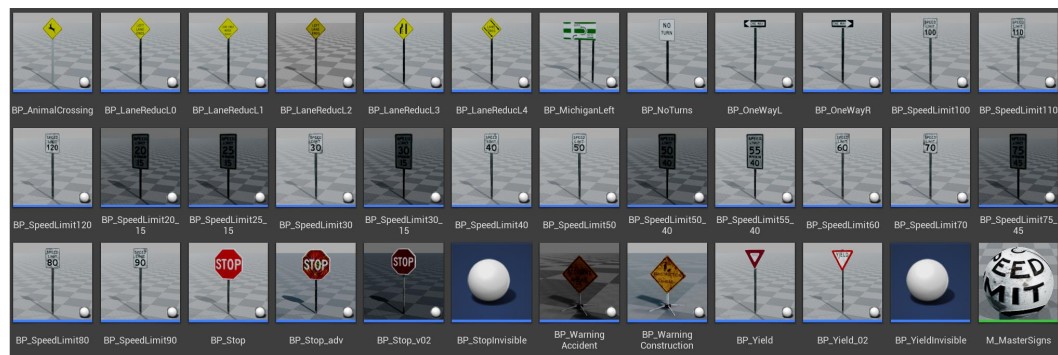

Figure 15: Illustration of the extentable traffic signs of the benchmark.

In order to ensure the validity of our benchmark for different types of objects, we have demonstrated that our benchmark can be easily extended to other target categories, as shown by the experiments conducted on traffic sign in Table 19. The benchmark is designed to evaluate the robustness of object detectors against physical attacks in various aligned scenarios for ensuring fairness. It can be extended to other target categories with minimal modifications.

We have thoroughly reviewed over forty physical attack methods, and we found that most of these methods conducted experiments under unaligned conditions and without fair comparisons. This lack of clarity hinders the accurate assessment of the progress of physical adversarial attacks and the development of physical adversarial robustness. Therefore, we are motivated to establish a comprehensive and rigorous benchmark for physical attacks to address these limitations and provide a solid foundation for future research.

## A.6    THE UTILITY OF THE BENCHMARK

In this section, we sumarize our motivation and provide the potential applications of the benchmark.

### A.6.1    UTILITIES OF THE BENCHMARK

**Standardization and Fair Evaluation**. The primary utility of PADetBench lies in its ability to standardize the evaluation of physical attacks against object detection models. By ensuring that all evaluations are conducted under the same physical dynamics, PADetBench eliminates inconsistencies found in real-world experiments, making it a fair and rigorous benchmark.

**Comprehensive Coverage**: PADetBench includes 23 physical attack methods and evaluates 48 state-of-the-art object detectors, providing a comprehensive coverage that enables researchers to compare and contrast various models and attack strategies.

### A.6.2    POTENTIAL APPLICATIONS OF THE BENCHMARK

**Research and Development**: Researchers developing robust object detection models or physical attack strategies need a benchmark to evaluate and compare their approaches.

**Security Assessments**: Security teams need to assess the robustness of deployed object detection systems in critical infrastructure.

**Regulatory Compliance**: Regulatory bodies require evidence of robustness and security for autonomous systems.

**Product Testing**: Companies developing autonomous vehicles or security systems need to test their products under various physical attack scenarios.

**Educational Purposes**: Educators and students need resources to understand the vulnerabilities of object detection models.

### A.7 LIMITATIONS AND POTENTIAL IMPACTS

**Limitations**

For now, PADetBench primarily focuses on evaluating the robustness of object detection models against physical attacks. In the future, we plan to extend the benchmark to include other vision tasks, such as instance segmentation, 3D object detection, and depth estimation. This expansion will provide a more comprehensive evaluation framework that covers a broader range of computer vision applications.

**Potential Impacts**

*1) Positive Impacts*: The in-depth understanding gained through PADetBench will contribute significantly to the development of more robust object detection models. By identifying vulnerabilities and limitations, researchers and practitioners can design improved algorithms that are better equipped to handle physical adversarial attacks. This enhanced robustness is crucial for real-world applications where reliability and accuracy are paramount.

*2) Negative Impacts*: While the benchmark provides valuable insights, there is a risk that it could be misused to conduct physical attacks in real-life scenarios. Such misuse could threaten the security of critical applications involving intelligent visual perception systems. Therefore, it is essential to promote responsible use of the benchmark and to emphasize the importance of ethical considerations in research and development.

## B ADDITIONAL CONTENT OF THE EXPERIMENTS

### B.1 GENETATED DATA FOR ABLATION STUDIES

We provide the generated data samples for the ablation studies in Fig. 16.

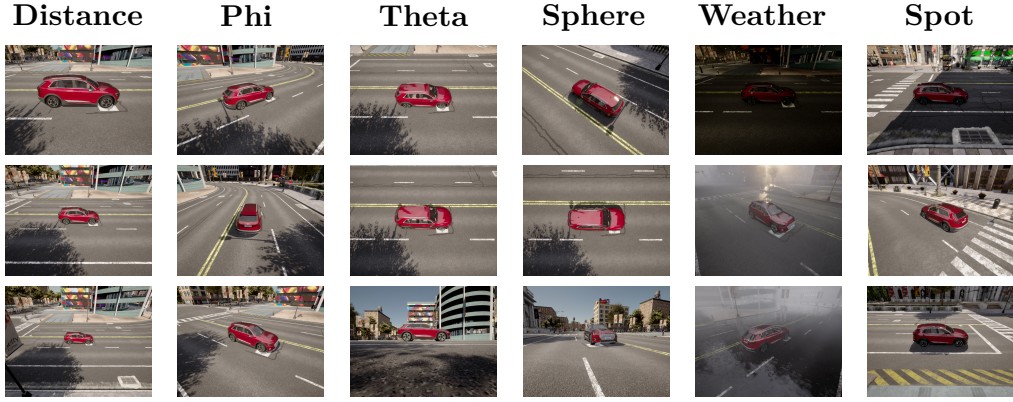

Figure 16: The randomly selected samples for the ablation studies of six different dynamics.

### B.2 A DETAILED ILLUSTRATION OF THE PERFORMANCE GAP

#### B.2.1 PERFORMANCE GAP BETWEEN THE BENCHMARK AND THE ORIGINAL PAPERS

In this section, we provide an explanation for the performance gap between the reported attack performance in the original papers and the results in our benchmark. Our benchmark encompasses a wide range of physical dynamics, whereas previous validation settings are often limited to a few specific scenarios. The comprehensive physical dynamics in our benchmark reveal the shortcomings of existing object detectors and physical attacks and this is the main motivation of our work. Therefore, our benchmark might not be captured by previous validation settings, leading to the discrepancy between our results and the reviewer's individual experiences.

In comparison, we removed various physical dynamics including weather (rain, snow, fog), lighting (nighttime), and distance (far positions), and reproduced the results of several attack methods on YOLOv7 as reported in ACTIVE (Suryanto et al., 2023), which are listed in Table 22. It is worth noting that these reported results are also included in our benchmark with particular evaluation settings.

Contrary to the simplified settings of these reproduced experiments, more comprehensive physical dynamics incorporated into our benchmark significantly highlight the ineffectiveness of existing physical attacks. These aspects may not have been adequately captured by previous validation settings. As illustrated in Tabel 22, when we exclude various dynamics, the effectiveness of physical attacks notably increases, thereby reducing the performance of object detectors. Therefore, our benchmark strive to encompass and align these physical dynamics for comprehensive and equitable comparisons.

### B.2.2 PERFORMANCE GAP BETWEEN ATTACKS AGAINST VEHICLE AND PERSON DETECTION

For the gap between attacks against vehicle and person detection, one reason is that these attacks are optimized to fool object detectors in particular target detection during training process. Consequently, we follow the attack purpose of the original works in this benchmark to attack specified target category accordingly for fairness, which partially accounts for the phenomenon that pedestrian detection performance is less affected regarding various attacks in comparison with car detection.

Another potential reason is that the stronger physical perturbations are optimized with consideration of 3D space and accommodate more complex physical dynamics, while physical attacks aiming to fool person detectors are commonly performed with optimized 2D patches, which work well in particular physical dynamics, as evidenced by the ablation experiments in B.2.1, which empirically demonstrate the pressing need and necessity of a comprehensive and rigorous benchmark for physical attacks.

### B.3 SUPPLEMENTED EXPERIMENTS ANALYSIS AND DISCUSSION

### B.3.1 DETECTION PERSPECTIVE

Vehicle Detection Perspective:

Physical attacks on vehicle detection systems pose a substantial challenge due to the specialized nature of the perturbations crafted to deceive these models. These attacks can lead to a drastic decline in average recall rates, reaching as low as 50%. This high level of vulnerability is largely attributed to the complex dynamics in the 3D environment where vehicles operate. Physical attacks on vehicle detectors exploit this three-dimensional context, introducing perturbations that consider real-world factors such as lighting, perspective, occlusion, and motion, making them more effective in disrupting the model's performance.

On the other hand, pedestrians, operating in a somewhat simpler 2D plane, seem to be less affected by similar adversarial attacks, with a decrease in average recall rates of less than 20%. Adversarial examples targeting pedestrian detection typically involve 2D patches, which might be more straightforward to apply in specific scenarios but may not account for the full range of real-world complexities. As a result, there is an urgent demand to establish a comprehensive and stringent benchmark to systematically evaluate the resilience of these models against physical attacks, facilitating research and development towards more secure systems.

Pedestrian Detection Perspective:

In contrast to vehicle detection, pedestrian detectors exhibit a certain level of inherent robustness, potentially due to the simpler constraints imposed on the recognition process. Nevertheless, as seen across various detectors, the extent of this robustness varies widely. Models like EfficientNet, YOLO series, RTMDet (one-stage detectors), and DDQ (transformer-based detectors) demonstrate commendable resistance to physical attacks. The superior performance of DDQ could be linked to the attention mechanisms inherent to transformer architectures, which are capable of capturing global spatial dependencies, thus mitigating the impact of adversarial perturbations.

However, it is evident from the benchmark results that not all state-of-the-art (SOTA) detectors offer comparable adversarial robustness. Many detectors exhibit varying degrees of vulnerability,

indicating that peak accuracy in standard detection tasks does not automatically guarantee resilience against adversarial threats. Consequently, this benchmarking framework not only identifies areas of weakness for refinement but also contributes to a better understanding of the interplay between detection performance and adversarial robustness in real-world deployments.

In conclusion, understanding and mitigating the effects of physical attacks in both vehicle and pedestrian detection domains can greatly benefit deep learning and computer vision research. By developing more robust models resistant to such attacks, we can enhance the safety and reliability of autonomous systems that rely on accurate object detection, ultimately fostering advancements in the fields of automotive technology, smart city infrastructure, and robotics. Furthermore, this benchmark would encourage researchers to explore defensive techniques and novel architectures that better withstand both digital and physical adversarial threats, pushing the boundaries of deep learning and computer vision capabilities.

### B.3.2 ATTACK PERSPECTIVE

From the attacker's viewpoint, the effectiveness of physical attacks on deep learning-based vehicle detection systems is highly variant. Certain methodologies, such as ACTIVE, achieve astonishingly high success rates in defeating the detectors, with ASR values surpassing 70%. However, the majority of current attacks struggle to maintain comparable performance, often failing to reach even 20% ASR. This discrepancy can be partly attributed to the rapid advancements in detection algorithms, with the latest state-of-the-art models like EfficientNet, YOLO series, and RTMDet demonstrating increased resilience against known attacks. This disparity in the evolutionary pace between attackers and defenders underscores the importance of continuous research and innovation in adversarial attacks to keep pace with the evolving landscape of detection techniques.

Moreover, this evolving dynamic underscores a critical need for a more dynamic and collaborative ecosystem in deep learning and computer vision research. By closing the gap between attack methods and detector capabilities, the field will likely see increased robustness and security measures, ultimately benefiting automotive safety and other real-world applications relying on these systems.

On the other hand, when evaluating person detection, the outcome of physical attacks exhibits a different pattern, with ASR values typically remaining below 20%, and often even below 0%, which indicates that the attack method is less effective than random guessing and the eye-catching perturbation may arouse more attention than the object itself. Additionally, the variable transferability of these attack methodologies across different detectors leads to a wide disparity in ASR values. In certain instances, this manifests as negative ASR figures, indicative of a backfiring effect where the detectors become more adept at identifying targets in the presence of attempted attacks.

The significantly lower effectiveness of these attacks on pedestrian detection models highlights the comparative advantages of their 2D nature against primarily 2D adversarial perturbations. Nevertheless, the AdvTexture method, despite being a 2D approach, manages to incorporate 3D considerations, achieving higher ASRs compared to other attacks. This underscores the pivotal role of incorporating 3D awareness into attack strategies to exploit the vulnerabilities of pedestrian detectors more effectively.

These contrasting observations highlight the need for more sophisticated attack methods in the domain of pedestrian detection. By advancing the understanding of how 2D techniques can be adapted or combined with 3D concepts, attackers can create more potent adversarial samples, driving defender-side innovation to fortify models further. Such advancements will ultimately contribute to the progression of the field by promoting the design of more secure and reliable computer vision systems, particularly relevant in surveillance, autonomous navigation, and smart city infrastructures.

In summary, the diverse outcomes of physical attacks on both vehicle and person detection emphasize the importance of ongoing research and competition between attack and defense approaches. As the attacks become more intricate and align with the complex nature of real-world scenarios, deep learning and computer vision models will adapt, increasing their resilience and overall functionality. This continuous push-and-pull between adversaries and protectors fosters the evolution of robust, secure, and accurate object-detection technologies essential for numerous applications, including automotive safety, surveillance, and urban automation.

Table 5: **Overall** experimental results of **vehicle** detection in the metric of **mAP50(%)**.

| | Clean | Random | ACTIVE | DTA | FCA | APPA | POOPatch | 3D2Fool | CAMOU | RPAU |
|---|---|---|---|---|---|---|---|---|---|---|
| ATSS | 0.83 | 0.477 | 0.231 | 0.545 | 0.606 | 0.502 | 0.434 | 0.678 | 0.235 | 0.532 |
| AutoAssign | 0.786 | 0.574 | 0.37 | 0.559 | 0.589 | 0.609 | 0.415 | 0.722 | 0.487 | 0.648 |
| CenterNet | 0.839 | 0.558 | 0.297 | 0.552 | 0.57 | 0.58 | 0.426 | 0.742 | 0.412 | 0.521 |
| CentripetalNet | 0.78 | 0.685 | 0.558 | 0.725 | 0.687 | 0.725 | 0.527 | 0.801 | 0.501 | 0.648 |
| CornerNet | 0.748 | 0.586 | 0.438 | 0.652 | 0.593 | 0.653 | 0.458 | 0.779 | 0.429 | 0.582 |
| DDOD | 0.838 | 0.708 | 0.433 | 0.695 | 0.686 | 0.745 | 0.548 | 0.785 | 0.694 | 0.646 |
| DyHead | 0.876 | 0.611 | 0.385 | 0.566 | 0.725 | 0.614 | 0.574 | 0.671 | 0.402 | 0.73 |
| EfficientNet | 0.881 | 0.711 | 0.506 | 0.687 | 0.721 | 0.764 | 0.638 | 0.763 | 0.71 | 0.665 |
| FCOS | 0.933 | 0.804 | 0.676 | 0.838 | 0.795 | 0.855 | 0.658 | 0.894 | 0.76 | 0.824 |
| FoveaBox | 0.814 | 0.597 | 0.294 | 0.514 | 0.645 | 0.548 | 0.467 | 0.649 | 0.469 | 0.618 |
| FreeAnchor | 0.81 | 0.51 | 0.381 | 0.611 | 0.563 | 0.643 | 0.431 | 0.638 | 0.336 | 0.582 |
| FSAF | 0.788 | 0.51 | 0.233 | 0.566 | 0.559 | 0.537 | 0.432 | 0.661 | 0.364 | 0.529 |
| GFL | 0.852 | 0.509 | 0.202 | 0.456 | 0.626 | 0.485 | 0.485 | 0.63 | 0.201 | 0.602 |
| LD | 0.825 | 0.554 | 0.305 | 0.563 | 0.658 | 0.548 | 0.463 | 0.664 | 0.29 | 0.591 |
| NAS-FPN | 0.87 | 0.623 | 0.473 | 0.662 | 0.673 | 0.695 | 0.5 | 0.764 | 0.382 | 0.655 |
| PAA | 0.808 | 0.582 | 0.474 | 0.621 | 0.605 | 0.619 | 0.501 | 0.685 | 0.567 | 0.64 |
| RetinaNet | 0.85 | 0.511 | 0.349 | 0.565 | 0.653 | 0.568 | 0.479 | 0.684 | 0.43 | 0.584 |
| RTMDet | 0.875 | 0.717 | 0.625 | 0.771 | 0.733 | 0.753 | 0.736 | 0.821 | 0.676 | 0.72 |
| TOOD | 0.781 | 0.495 | 0.353 | 0.522 | 0.584 | 0.557 | 0.462 | 0.615 | 0.37 | 0.572 |
| VarifocalNet | 0.874 | 0.472 | 0.205 | 0.424 | 0.628 | 0.529 | 0.419 | 0.573 | 0.208 | 0.538 |
| YOLOv5 | 0.886 | 0.76 | 0.744 | 0.762 | 0.807 | 0.821 | 0.788 | 0.857 | 0.812 | 0.75 |
| YOLOv6 | 0.907 | 0.824 | 0.747 | 0.834 | 0.833 | 0.887 | 0.776 | 0.896 | 0.851 | 0.784 |
| YOLOv7 | 0.906 | 0.774 | 0.762 | 0.829 | 0.822 | 0.866 | 0.786 | 0.903 | 0.774 | 0.803 |
| YOLOv8 | 0.929 | 0.791 | 0.761 | 0.812 | 0.838 | 0.873 | 0.793 | 0.917 | 0.74 | 0.803 |
| YOLOX | 0.908 | 0.766 | 0.683 | 0.783 | 0.817 | 0.851 | 0.794 | 0.849 | 0.702 | 0.782 |
| Faster R-CNN | 0.772 | 0.375 | 0.141 | 0.338 | 0.509 | 0.46 | 0.369 | 0.612 | 0.268 | 0.438 |
| Cascade R-CNN | 0.802 | 0.483 | 0.297 | 0.488 | 0.574 | 0.607 | 0.407 | 0.673 | 0.334 | 0.532 |
| Cascade RPN | 0.805 | 0.53 | 0.291 | 0.452 | 0.588 | 0.55 | 0.441 | 0.662 | 0.452 | 0.548 |
| Double Heads | 0.797 | 0.521 | 0.295 | 0.539 | 0.537 | 0.621 | 0.404 | 0.713 | 0.445 | 0.516 |
| FPG | 0.846 | 0.678 | 0.486 | 0.714 | 0.671 | 0.749 | 0.503 | 0.806 | 0.47 | 0.65 |
| Grid R-CNN | 0.795 | 0.472 | 0.244 | 0.513 | 0.529 | 0.65 | 0.399 | 0.699 | 0.392 | 0.494 |
| Guided Anchoring | 0.904 | 0.723 | 0.555 | 0.747 | 0.748 | 0.781 | 0.524 | 0.828 | 0.738 | 0.717 |
| HRNet | 0.76 | 0.547 | 0.29 | 0.52 | 0.512 | 0.571 | 0.336 | 0.738 | 0.462 | 0.511 |
| Libra R-CNN | 0.78 | 0.49 | 0.294 | 0.527 | 0.563 | 0.492 | 0.486 | 0.664 | 0.334 | 0.54 |
| PAFPN | 0.8 | 0.497 | 0.206 | 0.463 | 0.562 | 0.54 | 0.413 | 0.682 | 0.383 | 0.457 |
| RepPoints | 0.847 | 0.576 | 0.222 | 0.525 | 0.547 | 0.557 | 0.427 | 0.712 | 0.565 | 0.523 |
| Res2Net | 0.874 | 0.64 | 0.494 | 0.698 | 0.72 | 0.74 | 0.482 | 0.797 | 0.601 | 0.716 |
| ResNeSt | 0.837 | 0.502 | 0.352 | 0.535 | 0.587 | 0.499 | 0.538 | 0.493 | 0.407 | 0.555 |
| SABL | 0.796 | 0.46 | 0.262 | 0.501 | 0.563 | 0.535 | 0.423 | 0.648 | 0.359 | 0.484 |
| Sparse R-CNN | 0.774 | 0.469 | 0.257 | 0.398 | 0.604 | 0.418 | 0.44 | 0.518 | 0.316 | 0.532 |
| DETR | 0.636 | 0.114 | 0.048 | 0.033 | 0.351 | 0.17 | 0.333 | 0.198 | 0.025 | 0.339 |
| Conditional DETR | 0.793 | 0.554 | 0.408 | 0.575 | 0.644 | 0.617 | 0.525 | 0.7 | 0.457 | 0.671 |
| DDQ | 0.809 | 0.55 | 0.457 | 0.531 | 0.649 | 0.676 | 0.631 | 0.626 | 0.453 | 0.629 |
| DAB-DETR | 0.838 | 0.391 | 0.194 | 0.308 | 0.616 | 0.473 | 0.488 | 0.576 | 0.163 | 0.526 |
| Deformable DETR | 0.827 | 0.642 | 0.371 | 0.528 | 0.641 | 0.67 | 0.46 | 0.662 | 0.525 | 0.626 |
| DINO | 0.78 | 0.351 | 0.236 | 0.322 | 0.543 | 0.56 | 0.423 | 0.526 | 0.217 | 0.49 |
| PVT | 0.828 | 0.719 | 0.355 | 0.648 | 0.711 | 0.802 | 0.547 | 0.853 | 0.592 | 0.52 |
| PVTv2 | 0.845 | 0.666 | 0.494 | 0.763 | 0.621 | 0.844 | 0.425 | 0.803 | 0.704 | 0.476 |

## B.4 ADDITIONAL OVERALL EXPERIMENTAL RESULTS

Due to space constraints, we provide additional overall experimental results in this part, as shown in Table 5, 6, 7, 8, 9, 10, 11, and 12. In addition, the visualized evaluation results are shown in Fig. 17, 18, 19, 20, 21, 22, and 23.

Table 6: **Overall** experimental results of **vehicle** detection in the metric of **mAP50:95(%)**.

| | Clean | Random | ACTIVE | DTA | FCA | APPA | POOPatch | 3D2Fool | CAMOU | RPAU |
|---|---|---|---|---|---|---|---|---|---|---|
| ATSS | 0.238 | 0.156 | 0.077 | 0.182 | 0.183 | 0.164 | 0.133 | 0.212 | 0.087 | 0.166 |
| AutoAssign | 0.238 | 0.183 | 0.126 | 0.189 | 0.182 | 0.199 | 0.139 | 0.236 | 0.164 | 0.212 |
| CenterNet | 0.238 | 0.167 | 0.093 | 0.178 | 0.165 | 0.182 | 0.14 | 0.23 | 0.126 | 0.156 |
| CentripetalNet | 0.228 | 0.215 | 0.164 | 0.225 | 0.206 | 0.22 | 0.161 | 0.245 | 0.155 | 0.194 |
| CornerNet | 0.218 | 0.184 | 0.132 | 0.198 | 0.175 | 0.199 | 0.142 | 0.237 | 0.129 | 0.173 |
| DDOD | 0.242 | 0.21 | 0.127 | 0.221 | 0.202 | 0.234 | 0.174 | 0.242 | 0.211 | 0.193 |
| DyHead | 0.26 | 0.196 | 0.121 | 0.18 | 0.221 | 0.191 | 0.18 | 0.21 | 0.126 | 0.227 |
| EfficientNet | 0.252 | 0.225 | 0.168 | 0.217 | 0.219 | 0.239 | 0.206 | 0.243 | 0.232 | 0.21 |
| FCOS | 0.277 | 0.251 | 0.211 | 0.266 | 0.246 | 0.272 | 0.214 | 0.285 | 0.236 | 0.256 |
| FoveaBox | 0.235 | 0.184 | 0.089 | 0.165 | 0.191 | 0.17 | 0.145 | 0.196 | 0.149 | 0.193 |
| FreeAnchor | 0.241 | 0.159 | 0.111 | 0.19 | 0.179 | 0.205 | 0.138 | 0.205 | 0.098 | 0.184 |
| FSAF | 0.231 | 0.171 | 0.077 | 0.187 | 0.181 | 0.181 | 0.141 | 0.213 | 0.129 | 0.179 |
| GFL | 0.244 | 0.169 | 0.064 | 0.152 | 0.192 | 0.163 | 0.157 | 0.201 | 0.069 | 0.192 |
| LD | 0.235 | 0.176 | 0.095 | 0.181 | 0.205 | 0.176 | 0.146 | 0.208 | 0.094 | 0.187 |
| NAS-FPN | 0.256 | 0.199 | 0.146 | 0.209 | 0.213 | 0.215 | 0.16 | 0.251 | 0.124 | 0.208 |
| PAA | 0.237 | 0.178 | 0.139 | 0.189 | 0.178 | 0.195 | 0.157 | 0.208 | 0.173 | 0.194 |
| RetinaNet | 0.249 | 0.169 | 0.108 | 0.194 | 0.209 | 0.189 | 0.166 | 0.227 | 0.155 | 0.191 |
| RTMDet | 0.254 | 0.227 | 0.185 | 0.235 | 0.224 | 0.232 | 0.239 | 0.241 | 0.209 | 0.221 |
| TOOD | 0.231 | 0.155 | 0.109 | 0.159 | 0.173 | 0.176 | 0.14 | 0.183 | 0.118 | 0.169 |
| VarifocalNet | 0.248 | 0.144 | 0.063 | 0.127 | 0.188 | 0.164 | 0.132 | 0.175 | 0.062 | 0.162 |
| YOLOv5 | 0.259 | 0.227 | 0.223 | 0.23 | 0.237 | 0.249 | 0.244 | 0.253 | 0.246 | 0.221 |
| YOLOv6 | 0.256 | 0.245 | 0.218 | 0.251 | 0.242 | 0.262 | 0.238 | 0.263 | 0.25 | 0.229 |
| YOLOv7 | 0.265 | 0.241 | 0.225 | 0.252 | 0.246 | 0.262 | 0.241 | 0.272 | 0.227 | 0.243 |
| YOLOv8 | 0.276 | 0.246 | 0.239 | 0.254 | 0.252 | 0.269 | 0.256 | 0.283 | 0.236 | 0.246 |
| YOLOX | 0.263 | 0.233 | 0.212 | 0.236 | 0.237 | 0.253 | 0.248 | 0.251 | 0.217 | 0.233 |
| Faster R-CNN | 0.212 | 0.117 | 0.042 | 0.11 | 0.159 | 0.145 | 0.111 | 0.193 | 0.087 | 0.137 |
| Cascade R-CNN | 0.232 | 0.152 | 0.088 | 0.15 | 0.182 | 0.19 | 0.135 | 0.214 | 0.115 | 0.169 |
| Cascade RPN | 0.229 | 0.157 | 0.083 | 0.132 | 0.175 | 0.157 | 0.137 | 0.201 | 0.122 | 0.166 |
| Double Heads | 0.238 | 0.168 | 0.09 | 0.179 | 0.171 | 0.205 | 0.143 | 0.231 | 0.15 | 0.164 |
| FPG | 0.247 | 0.222 | 0.149 | 0.224 | 0.21 | 0.233 | 0.161 | 0.265 | 0.152 | 0.216 |
| Grid R-CNN | 0.231 | 0.154 | 0.078 | 0.173 | 0.17 | 0.208 | 0.135 | 0.225 | 0.127 | 0.164 |
| Guided Anchoring | 0.269 | 0.239 | 0.176 | 0.244 | 0.235 | 0.243 | 0.174 | 0.265 | 0.244 | 0.229 |
| HRNet | 0.219 | 0.171 | 0.091 | 0.164 | 0.159 | 0.173 | 0.114 | 0.236 | 0.148 | 0.165 |
| Libra R-CNN | 0.234 | 0.161 | 0.094 | 0.18 | 0.179 | 0.169 | 0.159 | 0.222 | 0.115 | 0.175 |
| PAFPN | 0.219 | 0.147 | 0.057 | 0.145 | 0.17 | 0.168 | 0.122 | 0.205 | 0.129 | 0.139 |
| RepPoints | 0.251 | 0.18 | 0.068 | 0.167 | 0.172 | 0.185 | 0.141 | 0.234 | 0.189 | 0.166 |
| Res2Net | 0.25 | 0.195 | 0.154 | 0.217 | 0.212 | 0.22 | 0.162 | 0.244 | 0.192 | 0.22 |
| ResNeSt | 0.23 | 0.156 | 0.101 | 0.158 | 0.174 | 0.154 | 0.169 | 0.142 | 0.126 | 0.162 |
| SABL | 0.233 | 0.146 | 0.08 | 0.159 | 0.173 | 0.177 | 0.139 | 0.199 | 0.123 | 0.155 |
| Sparse R-CNN | 0.238 | 0.159 | 0.095 | 0.137 | 0.195 | 0.145 | 0.154 | 0.172 | 0.128 | 0.168 |
| DETR | 0.186 | 0.047 | 0.017 | 0.01 | 0.113 | 0.059 | 0.111 | 0.065 | 0.009 | 0.105 |
| Conditional DETR | 0.236 | 0.183 | 0.129 | 0.183 | 0.199 | 0.205 | 0.172 | 0.211 | 0.154 | 0.212 |
| DDQ | 0.232 | 0.164 | 0.133 | 0.152 | 0.185 | 0.2 | 0.189 | 0.182 | 0.136 | 0.187 |
| DAB-DETR | 0.239 | 0.115 | 0.057 | 0.09 | 0.178 | 0.145 | 0.143 | 0.157 | 0.05 | 0.143 |
| Deformable DETR | 0.229 | 0.187 | 0.106 | 0.147 | 0.177 | 0.197 | 0.136 | 0.188 | 0.158 | 0.171 |
| DINO | 0.232 | 0.11 | 0.075 | 0.104 | 0.172 | 0.18 | 0.139 | 0.161 | 0.078 | 0.156 |
| PVT | 0.229 | 0.229 | 0.106 | 0.213 | 0.224 | 0.254 | 0.177 | 0.26 | 0.199 | 0.163 |
| PVTv2 | 0.24 | 0.214 | 0.154 | 0.252 | 0.195 | 0.275 | 0.147 | 0.244 | 0.237 | 0.149 |

## B.5 ADDITIONAL ABLATION EXPERIMENTS

### B.5.1 ABLATION STUDY ON PHYSICAL DYNAMICS

Due to space constraints, we provide additional ablation experimental results in this part, as shown in Table 13, 14, 15, 16, 17, and 18. In addition, the visualized evaluation results are shown in Fig. 24, 25, and 26.

Table 7: **Overall** experimental results of **vehicle** detection in the metric of **mAR50(%)**.

| | Clean | Random | ACTIVE | DTA | FCA | APPA | POOPatch | 3D2Fool | CAMOU | RPAU |
|---|---|---|---|---|---|---|---|---|---|---|
| ATSS | 0.973 | 0.808 | 0.518 | 0.84 | 0.883 | 0.811 | 0.732 | 0.924 | 0.549 | 0.826 |
| AutoAssign | 0.946 | 0.846 | 0.703 | 0.876 | 0.849 | 0.888 | 0.763 | 0.938 | 0.856 | 0.901 |
| CenterNet | 0.976 | 0.926 | 0.674 | 0.893 | 0.901 | 0.897 | 0.806 | 0.97 | 0.873 | 0.895 |
| CentripetalNet | 0.943 | 0.867 | 0.753 | 0.908 | 0.901 | 0.918 | 0.749 | 0.958 | 0.81 | 0.853 |
| CornerNet | 0.948 | 0.8 | 0.692 | 0.905 | 0.847 | 0.902 | 0.697 | 0.959 | 0.722 | 0.829 |
| DDOD | 0.95 | 0.936 | 0.756 | 0.923 | 0.9 | 0.934 | 0.863 | 0.948 | 0.949 | 0.909 |
| DyHead | 0.977 | 0.836 | 0.606 | 0.792 | 0.903 | 0.845 | 0.838 | 0.843 | 0.685 | 0.927 |
| EfficientNet | 0.977 | 0.974 | 0.912 | 0.975 | 0.974 | 0.97 | 0.951 | 0.973 | 0.966 | 0.948 |
| FCOS | 0.981 | 0.974 | 0.93 | 0.968 | 0.951 | 0.975 | 0.931 | 0.972 | 0.939 | 0.959 |
| FoveaBox | 0.955 | 0.873 | 0.623 | 0.806 | 0.896 | 0.832 | 0.748 | 0.881 | 0.768 | 0.878 |
| FreeAnchor | 0.973 | 0.841 | 0.855 | 0.928 | 0.864 | 0.939 | 0.781 | 0.89 | 0.68 | 0.887 |
| FSAF | 0.95 | 0.836 | 0.567 | 0.886 | 0.823 | 0.84 | 0.632 | 0.918 | 0.659 | 0.8 |
| GFL | 0.978 | 0.837 | 0.575 | 0.81 | 0.908 | 0.839 | 0.836 | 0.913 | 0.549 | 0.903 |
| LD | 0.98 | 0.893 | 0.579 | 0.862 | 0.927 | 0.871 | 0.743 | 0.917 | 0.676 | 0.91 |
| NAS-FPN | 0.975 | 0.916 | 0.768 | 0.932 | 0.944 | 0.946 | 0.817 | 0.937 | 0.766 | 0.925 |
| PAA | 0.966 | 0.923 | 0.861 | 0.952 | 0.905 | 0.937 | 0.895 | 0.938 | 0.951 | 0.92 |
| RetinaNet | 0.98 | 0.859 | 0.764 | 0.915 | 0.934 | 0.89 | 0.775 | 0.921 | 0.729 | 0.9 |
| RTMDet | 0.982 | 0.954 | 0.943 | 0.971 | 0.958 | 0.971 | 0.975 | 0.98 | 0.99 | 0.937 |
| TOOD | 0.908 | 0.763 | 0.666 | 0.83 | 0.834 | 0.826 | 0.717 | 0.847 | 0.622 | 0.853 |
| VarifocalNet | 0.977 | 0.802 | 0.486 | 0.77 | 0.9 | 0.832 | 0.671 | 0.866 | 0.466 | 0.829 |
| YOLOv5 | 0.975 | 0.979 | 0.967 | 0.974 | 0.977 | 0.974 | 0.974 | 0.972 | 0.985 | 0.966 |
| YOLOv6 | 0.985 | 0.982 | 0.968 | 0.974 | 0.983 | 0.979 | 0.974 | 0.984 | 0.978 | 0.973 |
| YOLOv7 | 0.962 | 0.924 | 0.931 | 0.948 | 0.939 | 0.958 | 0.932 | 0.96 | 0.932 | 0.931 |
| YOLOv8 | 0.975 | 0.919 | 0.903 | 0.93 | 0.942 | 0.96 | 0.916 | 0.965 | 0.885 | 0.917 |
| YOLOX | 0.955 | 0.877 | 0.853 | 0.902 | 0.91 | 0.945 | 0.926 | 0.918 | 0.868 | 0.891 |
| Faster R-CNN | 0.846 | 0.479 | 0.268 | 0.493 | 0.595 | 0.593 | 0.417 | 0.752 | 0.337 | 0.532 |
| Cascade R-CNN | 0.854 | 0.539 | 0.382 | 0.591 | 0.65 | 0.689 | 0.448 | 0.78 | 0.368 | 0.602 |
| Cascade RPN | 0.973 | 0.884 | 0.741 | 0.933 | 0.898 | 0.957 | 0.885 | 0.967 | 0.868 | 0.891 |
| Double Heads | 0.849 | 0.597 | 0.416 | 0.654 | 0.621 | 0.725 | 0.459 | 0.819 | 0.488 | 0.594 |
| FPG | 0.912 | 0.763 | 0.624 | 0.848 | 0.761 | 0.866 | 0.587 | 0.907 | 0.61 | 0.748 |
| Grid R-CNN | 0.873 | 0.585 | 0.39 | 0.672 | 0.653 | 0.794 | 0.488 | 0.836 | 0.495 | 0.614 |
| Guided Anchoring | 0.975 | 0.962 | 0.914 | 0.966 | 0.962 | 0.966 | 0.839 | 0.961 | 0.929 | 0.956 |
| HRNet | 0.844 | 0.655 | 0.429 | 0.687 | 0.627 | 0.732 | 0.423 | 0.875 | 0.532 | 0.609 |
| Libra R-CNN | 0.959 | 0.828 | 0.599 | 0.824 | 0.892 | 0.775 | 0.805 | 0.92 | 0.563 | 0.858 |
| PAFPN | 0.856 | 0.61 | 0.337 | 0.591 | 0.661 | 0.659 | 0.49 | 0.805 | 0.434 | 0.569 |
| RepPoints | 0.978 | 0.903 | 0.595 | 0.874 | 0.885 | 0.884 | 0.833 | 0.945 | 0.883 | 0.848 |
| Res2Net | 0.911 | 0.692 | 0.541 | 0.757 | 0.763 | 0.793 | 0.511 | 0.852 | 0.646 | 0.761 |
| ResNeSt | 0.929 | 0.646 | 0.497 | 0.727 | 0.701 | 0.691 | 0.685 | 0.802 | 0.515 | 0.716 |
| SABL | 0.856 | 0.545 | 0.387 | 0.646 | 0.656 | 0.636 | 0.488 | 0.774 | 0.407 | 0.571 |
| Sparse R-CNN | 0.959 | 0.877 | 0.513 | 0.759 | 0.913 | 0.746 | 0.733 | 0.882 | 0.605 | 0.871 |
| DETR | 0.746 | 0.328 | 0.232 | 0.256 | 0.468 | 0.383 | 0.457 | 0.369 | 0.234 | 0.468 |
| Conditional DETR | 0.962 | 0.831 | 0.73 | 0.931 | 0.865 | 0.934 | 0.881 | 0.964 | 0.839 | 0.917 |
| DDQ | 0.983 | 0.976 | 0.972 | 0.979 | 0.975 | 0.975 | 0.977 | 0.974 | 0.983 | 0.969 |
| DAB-DETR | 0.98 | 0.909 | 0.928 | 0.97 | 0.948 | 0.968 | 0.946 | 0.924 | 0.91 | 0.934 |
| Deformable DETR | 0.954 | 0.907 | 0.704 | 0.902 | 0.879 | 0.924 | 0.766 | 0.905 | 0.91 | 0.88 |
| DINO | 0.975 | 0.895 | 0.883 | 0.953 | 0.923 | 0.958 | 0.922 | 0.953 | 0.912 | 0.924 |
| PVT | 0.948 | 0.953 | 0.827 | 0.936 | 0.957 | 0.956 | 0.901 | 0.963 | 0.954 | 0.886 |
| PVTv2 | 0.973 | 0.942 | 0.884 | 0.952 | 0.934 | 0.973 | 0.835 | 0.967 | 0.941 | 0.867 |

### B.5.2 ABLATION STUDY ON TRAINING DATASET

To further investigate the impact of the training dataset on the physical attacks, we collected ten physical attacks for fooling person detection, and the results are shown in Table 20, where the Median ASR represents the median attack success rate across the 48 detectors. It can be observed that physical attacks trained on the INRIA and COCO datasets achieve comparable performance in general.

Table 8: **Overall** experimental results of **vehicle** detection in the metric of **mAR50:95(%)**.

| | Clean | Random | ACTIVE | DTA | FCA | APPA | POOPatch | 3D2Fool | CAMOU | RPAU |
|---|---|---|---|---|---|---|---|---|---|---|
| ATSS | 0.374 | 0.318 | 0.206 | 0.341 | 0.342 | 0.321 | 0.296 | 0.371 | 0.219 | 0.325 |
| AutoAssign | 0.385 | 0.341 | 0.293 | 0.366 | 0.335 | 0.367 | 0.32 | 0.39 | 0.345 | 0.374 |
| CenterNet | 0.396 | 0.365 | 0.275 | 0.373 | 0.353 | 0.373 | 0.342 | 0.408 | 0.344 | 0.361 |
| CentripetalNet | 0.378 | 0.362 | 0.313 | 0.386 | 0.365 | 0.384 | 0.31 | 0.404 | 0.321 | 0.35 |
| CornerNet | 0.387 | 0.337 | 0.286 | 0.382 | 0.343 | 0.38 | 0.292 | 0.401 | 0.282 | 0.336 |
| DDOD | 0.366 | 0.363 | 0.302 | 0.371 | 0.347 | 0.379 | 0.357 | 0.383 | 0.366 | 0.358 |
| DyHead | 0.378 | 0.338 | 0.254 | 0.32 | 0.35 | 0.338 | 0.342 | 0.336 | 0.255 | 0.364 |
| EfficientNet | 0.387 | 0.397 | 0.377 | 0.395 | 0.393 | 0.394 | 0.399 | 0.402 | 0.406 | 0.38 |
| FCOS | 0.401 | 0.4 | 0.382 | 0.399 | 0.389 | 0.405 | 0.389 | 0.418 | 0.37 | 0.397 |
| FoveaBox | 0.371 | 0.337 | 0.246 | 0.327 | 0.344 | 0.333 | 0.304 | 0.353 | 0.303 | 0.344 |
| FreeAnchor | 0.384 | 0.332 | 0.333 | 0.367 | 0.345 | 0.38 | 0.324 | 0.365 | 0.244 | 0.357 |
| FSAF | 0.37 | 0.337 | 0.229 | 0.362 | 0.33 | 0.342 | 0.271 | 0.377 | 0.257 | 0.325 |
| GFL | 0.375 | 0.337 | 0.238 | 0.336 | 0.354 | 0.341 | 0.348 | 0.373 | 0.226 | 0.357 |
| LD | 0.372 | 0.352 | 0.236 | 0.351 | 0.367 | 0.348 | 0.314 | 0.367 | 0.268 | 0.36 |
| NAS-FPN | 0.38 | 0.367 | 0.307 | 0.373 | 0.378 | 0.375 | 0.34 | 0.384 | 0.296 | 0.373 |
| PAA | 0.399 | 0.371 | 0.337 | 0.385 | 0.357 | 0.39 | 0.376 | 0.388 | 0.383 | 0.369 |
| RetinaNet | 0.392 | 0.343 | 0.298 | 0.371 | 0.378 | 0.367 | 0.34 | 0.39 | 0.301 | 0.364 |
| RTMDet | 0.357 | 0.322 | 0.293 | 0.345 | 0.319 | 0.351 | 0.34 | 0.362 | 0.304 | 0.311 |
| TOOD | 0.345 | 0.293 | 0.261 | 0.332 | 0.312 | 0.329 | 0.283 | 0.327 | 0.236 | 0.323 |
| VarifocalNet | 0.376 | 0.31 | 0.188 | 0.304 | 0.352 | 0.329 | 0.277 | 0.338 | 0.173 | 0.321 |
| YOLOv5 | 0.364 | 0.364 | 0.366 | 0.371 | 0.358 | 0.378 | 0.377 | 0.373 | 0.354 | 0.355 |
| YOLOv6 | 0.357 | 0.361 | 0.343 | 0.363 | 0.352 | 0.37 | 0.359 | 0.37 | 0.351 | 0.343 |
| YOLOv7 | 0.366 | 0.356 | 0.339 | 0.36 | 0.355 | 0.371 | 0.358 | 0.376 | 0.338 | 0.352 |
| YOLOv8 | 0.376 | 0.358 | 0.354 | 0.369 | 0.357 | 0.376 | 0.368 | 0.385 | 0.336 | 0.354 |
| YOLOX | 0.362 | 0.33 | 0.325 | 0.345 | 0.338 | 0.368 | 0.367 | 0.358 | 0.309 | 0.332 |
| Faster R-CNN | 0.303 | 0.187 | 0.099 | 0.194 | 0.228 | 0.236 | 0.163 | 0.302 | 0.134 | 0.203 |
| Cascade R-CNN | 0.316 | 0.213 | 0.147 | 0.232 | 0.255 | 0.272 | 0.185 | 0.312 | 0.145 | 0.239 |
| Cascade RPN | 0.375 | 0.333 | 0.276 | 0.351 | 0.341 | 0.358 | 0.353 | 0.376 | 0.307 | 0.341 |
| Double Heads | 0.32 | 0.239 | 0.165 | 0.27 | 0.249 | 0.299 | 0.2 | 0.333 | 0.193 | 0.238 |
| FPG | 0.347 | 0.31 | 0.249 | 0.35 | 0.302 | 0.352 | 0.239 | 0.379 | 0.249 | 0.306 |
| Grid R-CNN | 0.326 | 0.23 | 0.155 | 0.273 | 0.259 | 0.312 | 0.205 | 0.33 | 0.188 | 0.244 |
| Guided Anchoring | 0.38 | 0.396 | 0.373 | 0.39 | 0.385 | 0.384 | 0.35 | 0.399 | 0.363 | 0.379 |
| HRNet | 0.303 | 0.239 | 0.159 | 0.252 | 0.228 | 0.264 | 0.167 | 0.333 | 0.188 | 0.226 |
| Libra R-CNN | 0.383 | 0.326 | 0.233 | 0.334 | 0.357 | 0.315 | 0.333 | 0.385 | 0.223 | 0.337 |
| PAFPN | 0.304 | 0.226 | 0.127 | 0.228 | 0.249 | 0.254 | 0.185 | 0.304 | 0.167 | 0.214 |
| RepPoints | 0.387 | 0.356 | 0.235 | 0.355 | 0.349 | 0.372 | 0.358 | 0.393 | 0.364 | 0.336 |
| Res2Net | 0.341 | 0.261 | 0.208 | 0.302 | 0.284 | 0.305 | 0.21 | 0.338 | 0.237 | 0.292 |
| ResNeSt | 0.341 | 0.257 | 0.189 | 0.274 | 0.261 | 0.27 | 0.268 | 0.302 | 0.201 | 0.271 |
| SABL | 0.318 | 0.213 | 0.151 | 0.26 | 0.253 | 0.257 | 0.204 | 0.303 | 0.16 | 0.226 |
| Sparse R-CNN | 0.395 | 0.367 | 0.219 | 0.324 | 0.38 | 0.32 | 0.312 | 0.381 | 0.249 | 0.361 |
| DETR | 0.334 | 0.144 | 0.105 | 0.106 | 0.203 | 0.171 | 0.204 | 0.158 | 0.085 | 0.203 |
| Conditional DETR | 0.387 | 0.341 | 0.32 | 0.404 | 0.342 | 0.406 | 0.367 | 0.391 | 0.328 | 0.374 |
| DDQ | 0.393 | 0.389 | 0.388 | 0.394 | 0.39 | 0.389 | 0.403 | 0.391 | 0.374 | 0.389 |
| DAB-DETR | 0.404 | 0.366 | 0.384 | 0.406 | 0.385 | 0.413 | 0.404 | 0.376 | 0.36 | 0.383 |
| Deformable DETR | 0.378 | 0.352 | 0.277 | 0.351 | 0.333 | 0.372 | 0.3 | 0.351 | 0.339 | 0.34 |
| DINO | 0.388 | 0.353 | 0.353 | 0.39 | 0.364 | 0.388 | 0.372 | 0.384 | 0.346 | 0.37 |
| PVT | 0.349 | 0.371 | 0.319 | 0.365 | 0.37 | 0.379 | 0.365 | 0.382 | 0.387 | 0.34 |
| PVTv2 | 0.372 | 0.372 | 0.359 | 0.386 | 0.365 | 0.399 | 0.354 | 0.39 | 0.384 | 0.343 |

### B.5.3 ABLATION STUDY ON 2D AND 3D PERTURBATIONS

Physical attacks that evaluate 2D adversarial patches from a frontal perspective have a significant limitation, as they do not account for the effects of multiple viewing angles in a 3D environment. Our study aims to bridge this gap by developing a comprehensive benchmark for assessing physical attacks from various angles and incorporating a broader range of physical dynamics. During our investigation, we noted a substantial drop in performance (detection rate: $\frac{n_{detected}}{n_{total}}$) when adversarial patches were only applied to the frontal view of objects. To ensure a fair comparison and enhance

Table 9: **Overall** experimental results of **person** detection in the metric of **mAP50(%)**.

| | Clean | Random | AdvCam | UPC | NatPatch | MTD | LAP | InvisCloak | DAP | AdvTshirt | AdvTexture | AdvPatch | AdvPattern | AdvCaT |
|---|---|---|---|---|---|---|---|---|---|---|---|---|---|---|
| ATSS | 0.54 | 0.517 | 0.498 | 0.428 | 0.419 | 0.473 | 0.522 | 0.468 | 0.495 | 0.458 | 0.385 | 0.454 | 0.492 | 0.514 |
| AutoAssign | 0.491 | 0.466 | 0.454 | 0.314 | 0.36 | 0.423 | 0.456 | 0.43 | 0.403 | 0.41 | 0.346 | 0.427 | 0.453 | 0.484 |
| CenterNet | 0.524 | 0.476 | 0.477 | 0.408 | 0.39 | 0.469 | 0.483 | 0.436 | 0.437 | 0.45 | 0.372 | 0.43 | 0.474 | 0.524 |
| CentripetalNet | 0.526 | 0.53 | 0.524 | 0.48 | 0.349 | 0.524 | 0.508 | 0.51 | 0.471 | 0.473 | 0.405 | 0.472 | 0.515 | 0.526 |
| CornerNet | 0.517 | 0.51 | 0.505 | 0.403 | 0.295 | 0.488 | 0.494 | 0.449 | 0.444 | 0.42 | 0.345 | 0.414 | 0.48 | 0.506 |
| DDOD | 0.481 | 0.48 | 0.448 | 0.359 | 0.416 | 0.421 | 0.47 | 0.445 | 0.453 | 0.433 | 0.329 | 0.409 | 0.442 | 0.45 |
| DyHead | 0.474 | 0.483 | 0.485 | 0.406 | 0.402 | 0.464 | 0.501 | 0.454 | 0.473 | 0.433 | 0.4 | 0.433 | 0.467 | 0.474 |
| EfficientNet | 0.457 | 0.431 | 0.442 | 0.398 | 0.418 | 0.406 | 0.431 | 0.399 | 0.407 | 0.403 | 0.394 | 0.396 | 0.422 | 0.433 |
| FCOS | 0.45 | 0.438 | 0.429 | 0.364 | 0.383 | 0.41 | 0.433 | 0.407 | 0.404 | 0.407 | 0.356 | 0.409 | 0.425 | 0.448 |
| FoveaBox | 0.543 | 0.53 | 0.536 | 0.473 | 0.475 | 0.482 | 0.54 | 0.481 | 0.523 | 0.483 | 0.374 | 0.458 | 0.498 | 0.533 |
| FreeAnchor | 0.537 | 0.522 | 0.493 | 0.414 | 0.396 | 0.431 | 0.492 | 0.443 | 0.446 | 0.411 | 0.331 | 0.447 | 0.48 | 0.513 |
| FSAF | 0.554 | 0.551 | 0.529 | 0.444 | 0.439 | 0.485 | 0.538 | 0.496 | 0.515 | 0.457 | 0.379 | 0.479 | 0.512 | 0.527 |
| GFL | 0.57 | 0.541 | 0.532 | 0.431 | 0.453 | 0.495 | 0.509 | 0.478 | 0.495 | 0.48 | 0.398 | 0.459 | 0.52 | 0.547 |
| LD | 0.57 | 0.54 | 0.524 | 0.401 | 0.397 | 0.486 | 0.517 | 0.484 | 0.489 | 0.477 | 0.385 | 0.484 | 0.519 | 0.535 |
| NAS-FPN | 0.442 | 0.436 | 0.433 | 0.365 | 0.391 | 0.4 | 0.451 | 0.399 | 0.394 | 0.398 | 0.32 | 0.399 | 0.413 | 0.435 |
| PAA | 0.464 | 0.464 | 0.457 | 0.402 | 0.389 | 0.43 | 0.451 | 0.432 | 0.447 | 0.402 | 0.322 | 0.409 | 0.441 | 0.463 |
| RetinaNet | 0.522 | 0.543 | 0.497 | 0.438 | 0.425 | 0.459 | 0.505 | 0.483 | 0.478 | 0.454 | 0.384 | 0.475 | 0.489 | 0.528 |
| RTMDet | 0.533 | 0.482 | 0.515 | 0.52 | 0.466 | 0.46 | 0.495 | 0.437 | 0.472 | 0.47 | 0.459 | 0.449 | 0.466 | 0.5 |
| TOOD | 0.474 | 0.5 | 0.475 | 0.384 | 0.376 | 0.453 | 0.503 | 0.453 | 0.486 | 0.441 | 0.371 | 0.435 | 0.457 | 0.475 |
| VarifocalNet | 0.492 | 0.505 | 0.481 | 0.387 | 0.395 | 0.443 | 0.504 | 0.444 | 0.469 | 0.445 | 0.368 | 0.436 | 0.47 | 0.506 |
| YOLOv5 | 0.481 | 0.46 | 0.472 | 0.448 | 0.403 | 0.453 | 0.484 | 0.454 | 0.485 | 0.418 | 0.35 | 0.42 | 0.452 | 0.459 |
| YOLOv6 | 0.467 | 0.445 | 0.461 | 0.456 | 0.435 | 0.438 | 0.444 | 0.446 | 0.459 | 0.437 | 0.423 | 0.432 | 0.444 | 0.449 |
| YOLOv7 | 0.463 | 0.438 | 0.48 | 0.446 | 0.343 | 0.401 | 0.438 | 0.403 | 0.457 | 0.402 | 0.372 | 0.366 | 0.429 | 0.437 |
| YOLOv8 | 0.434 | 0.421 | 0.431 | 0.432 | 0.402 | 0.415 | 0.421 | 0.416 | 0.429 | 0.416 | 0.405 | 0.409 | 0.415 | 0.417 |
| YOLOX | 0.448 | 0.436 | 0.457 | 0.457 | 0.382 | 0.432 | 0.46 | 0.425 | 0.46 | 0.433 | 0.393 | 0.412 | 0.426 | 0.437 |
| Faster R-CNN | 0.541 | 0.547 | 0.497 | 0.416 | 0.425 | 0.456 | 0.532 | 0.468 | 0.456 | 0.448 | 0.341 | 0.432 | 0.512 | 0.534 |
| Cascade R-CNN | 0.559 | 0.551 | 0.539 | 0.431 | 0.445 | 0.488 | 0.551 | 0.463 | 0.508 | 0.479 | 0.355 | 0.454 | 0.523 | 0.55 |
| Cascade RPN | 0.538 | 0.537 | 0.528 | 0.389 | 0.407 | 0.482 | 0.508 | 0.472 | 0.483 | 0.461 | 0.335 | 0.445 | 0.508 | 0.532 |
| Double Heads | 0.552 | 0.526 | 0.531 | 0.419 | 0.408 | 0.46 | 0.533 | 0.442 | 0.489 | 0.454 | 0.359 | 0.44 | 0.496 | 0.527 |
| FPG | 0.462 | 0.473 | 0.451 | 0.413 | 0.395 | 0.415 | 0.466 | 0.408 | 0.424 | 0.405 | 0.317 | 0.409 | 0.444 | 0.45 |
| Grid R-CNN | 0.512 | 0.502 | 0.492 | 0.397 | 0.404 | 0.449 | 0.517 | 0.462 | 0.471 | 0.43 | 0.363 | 0.424 | 0.481 | 0.488 |
| Guided Anchoring | 0.497 | 0.537 | 0.504 | 0.427 | 0.396 | 0.47 | 0.525 | 0.479 | 0.454 | 0.449 | 0.375 | 0.452 | 0.494 | 0.502 |
| HRNet | 0.498 | 0.489 | 0.495 | 0.457 | 0.404 | 0.453 | 0.489 | 0.47 | 0.442 | 0.472 | 0.419 | 0.437 | 0.443 | 0.497 |
| Libra R-CNN | 0.535 | 0.517 | 0.479 | 0.452 | 0.404 | 0.446 | 0.468 | 0.433 | 0.447 | 0.431 | 0.374 | 0.421 | 0.442 | 0.494 |
| PAFPN | 0.539 | 0.534 | 0.529 | 0.429 | 0.438 | 0.468 | 0.522 | 0.477 | 0.47 | 0.458 | 0.349 | 0.447 | 0.516 | 0.559 |
| RepPoints | 0.572 | 0.559 | 0.53 | 0.434 | 0.475 | 0.478 | 0.535 | 0.47 | 0.504 | 0.455 | 0.38 | 0.451 | 0.525 | 0.56 |
| Res2Net | 0.449 | 0.437 | 0.435 | 0.403 | 0.301 | 0.402 | 0.458 | 0.406 | 0.407 | 0.386 | 0.324 | 0.387 | 0.428 | 0.451 |
| ResNeSt | 0.443 | 0.455 | 0.409 | 0.396 | 0.396 | 0.405 | 0.432 | 0.385 | 0.364 | 0.374 | 0.358 | 0.374 | 0.416 | 0.451 |
| SABL | 0.563 | 0.559 | 0.525 | 0.418 | 0.471 | 0.491 | 0.534 | 0.503 | 0.498 | 0.496 | 0.382 | 0.484 | 0.534 | 0.552 |
| Sparse R-CNN | 0.492 | 0.481 | 0.477 | 0.38 | 0.347 | 0.434 | 0.484 | 0.386 | 0.396 | 0.407 | 0.352 | 0.389 | 0.455 | 0.493 |
| DETR | 0.553 | 0.497 | 0.481 | 0.337 | 0.318 | 0.467 | 0.49 | 0.466 | 0.432 | 0.473 | 0.343 | 0.444 | 0.475 | 0.51 |
| Conditional DETR | 0.535 | 0.497 | 0.453 | 0.378 | 0.351 | 0.424 | 0.449 | 0.401 | 0.44 | 0.416 | 0.294 | 0.412 | 0.422 | 0.485 |
| DDQ | 0.449 | 0.454 | 0.452 | 0.377 | 0.388 | 0.424 | 0.451 | 0.426 | 0.408 | 0.422 | 0.34 | 0.421 | 0.439 | 0.451 |
| DAB-DETR | 0.441 | 0.429 | 0.428 | 0.344 | 0.36 | 0.371 | 0.407 | 0.357 | 0.373 | 0.374 | 0.276 | 0.336 | 0.392 | 0.44 |
| Deformable DETR | 0.475 | 0.462 | 0.475 | 0.345 | 0.389 | 0.421 | 0.46 | 0.442 | 0.418 | 0.424 | 0.286 | 0.416 | 0.471 | 0.45 |
| DINO | 0.419 | 0.421 | 0.416 | 0.337 | 0.283 | 0.385 | 0.415 | 0.402 | 0.375 | 0.391 | 0.316 | 0.378 | 0.394 | 0.423 |
| PVT | 0.474 | 0.465 | 0.418 | 0.378 | 0.368 | 0.383 | 0.405 | 0.359 | 0.369 | 0.393 | 0.381 | 0.391 | 0.41 | 0.426 |
| PVTv2 | 0.51 | 0.431 | 0.41 | 0.403 | 0.4 | 0.395 | 0.414 | 0.389 | 0.41 | 0.384 | 0.347 | 0.382 | 0.392 | 0.436 |

the efficacy of the attacks, we expanded the application of these patches to cover the entirety of the object's surface. Additional experiments were conducted to assess the impact of adversarial patches on frontal views using several object detection algorithms. The results are summarized in Table 21. The 'Entire Surface' column highlights cases where the adversarial patch was applied across the entire surface of an object. The values in parentheses indicate the relative decrease in performance compared to full-surface patching.

## C  USER FEEDBACK

To ensure ease of use, we have addressed potential barriers by user feedback, such as CARLA deployment and customizing adversarial objects, by providing a comprehensive Docker installation guide for CARLA and a tutorial on customizing adversarial objects in our documentation. These resources enable users to install CARLA and customize objects in just a few minutes. We also conducted usability testing with five researchers from a well-known University and got feedback from them in the form of a survey questionnaire as shown in Table 24. The users consistently found the benchmark easy to use and provided positive feedback on its usability.

Table 10: **Overall** experimental results of **person** detection in the metric of **mAP50:95(%)**.

| | Clean | Random | AdvCam | UPC | NatPatch | MTD | LAP | InvisCloak | DAP | AdvTshirt | AdvTexture | AdvPatch | AdvPattern | AdvCaT |
|---|---|---|---|---|---|---|---|---|---|---|---|---|---|---|
| ATSS | 0.157 | 0.148 | 0.147 | 0.121 | 0.113 | 0.127 | 0.155 | 0.138 | 0.143 | 0.128 | 0.098 | 0.132 | 0.138 | 0.151 |
| AutoAssign | 0.149 | 0.137 | 0.13 | 0.082 | 0.098 | 0.12 | 0.136 | 0.126 | 0.113 | 0.114 | 0.089 | 0.123 | 0.131 | 0.143 |
| CenterNet | 0.161 | 0.145 | 0.147 | 0.12 | 0.111 | 0.141 | 0.151 | 0.132 | 0.132 | 0.135 | 0.102 | 0.124 | 0.142 | 0.166 |
| CentripetalNet | 0.143 | 0.145 | 0.144 | 0.142 | 0.083 | 0.145 | 0.139 | 0.142 | 0.131 | 0.128 | 0.109 | 0.127 | 0.138 | 0.143 |
| CornerNet | 0.142 | 0.141 | 0.138 | 0.12 | 0.07 | 0.133 | 0.141 | 0.128 | 0.125 | 0.115 | 0.091 | 0.111 | 0.132 | 0.14 |
| DDOD | 0.142 | 0.141 | 0.129 | 0.102 | 0.119 | 0.115 | 0.139 | 0.133 | 0.131 | 0.129 | 0.085 | 0.114 | 0.126 | 0.128 |
| DyHead | 0.135 | 0.137 | 0.144 | 0.113 | 0.109 | 0.127 | 0.149 | 0.132 | 0.141 | 0.12 | 0.106 | 0.122 | 0.131 | 0.134 |
| EfficientNet | 0.123 | 0.115 | 0.119 | 0.11 | 0.113 | 0.106 | 0.116 | 0.105 | 0.11 | 0.106 | 0.104 | 0.104 | 0.109 | 0.115 |
| FCOS | 0.126 | 0.117 | 0.117 | 0.1 | 0.102 | 0.108 | 0.118 | 0.109 | 0.106 | 0.107 | 0.094 | 0.106 | 0.109 | 0.123 |
| FoveaBox | 0.173 | 0.163 | 0.166 | 0.135 | 0.133 | 0.141 | 0.162 | 0.141 | 0.155 | 0.136 | 0.098 | 0.131 | 0.15 | 0.161 |
| FreeAnchor | 0.165 | 0.155 | 0.147 | 0.122 | 0.106 | 0.121 | 0.149 | 0.134 | 0.132 | 0.108 | 0.083 | 0.127 | 0.14 | 0.152 |
| FSAF | 0.175 | 0.174 | 0.166 | 0.135 | 0.13 | 0.142 | 0.169 | 0.155 | 0.158 | 0.131 | 0.097 | 0.144 | 0.159 | 0.167 |
| GFL | 0.177 | 0.164 | 0.162 | 0.123 | 0.13 | 0.138 | 0.16 | 0.143 | 0.157 | 0.144 | 0.108 | 0.133 | 0.153 | 0.164 |
| LD | 0.179 | 0.164 | 0.162 | 0.114 | 0.109 | 0.143 | 0.162 | 0.147 | 0.148 | 0.142 | 0.101 | 0.145 | 0.157 | 0.162 |
| NAS-FPN | 0.127 | 0.119 | 0.122 | 0.107 | 0.11 | 0.106 | 0.133 | 0.112 | 0.111 | 0.106 | 0.076 | 0.11 | 0.111 | 0.122 |
| PAA | 0.132 | 0.132 | 0.13 | 0.112 | 0.103 | 0.116 | 0.135 | 0.128 | 0.129 | 0.108 | 0.081 | 0.12 | 0.122 | 0.129 |
| RetinaNet | 0.155 | 0.167 | 0.151 | 0.135 | 0.123 | 0.13 | 0.158 | 0.152 | 0.142 | 0.134 | 0.106 | 0.144 | 0.144 | 0.154 |
| RTMDet | 0.152 | 0.133 | 0.148 | 0.158 | 0.129 | 0.12 | 0.14 | 0.112 | 0.137 | 0.129 | 0.126 | 0.117 | 0.121 | 0.138 |
| TOOD | 0.131 | 0.145 | 0.139 | 0.108 | 0.099 | 0.121 | 0.154 | 0.13 | 0.143 | 0.122 | 0.093 | 0.122 | 0.123 | 0.132 |
| VarifocalNet | 0.141 | 0.152 | 0.145 | 0.11 | 0.111 | 0.127 | 0.157 | 0.133 | 0.144 | 0.125 | 0.095 | 0.132 | 0.137 | 0.154 |
| YOLOv5 | 0.135 | 0.124 | 0.132 | 0.129 | 0.104 | 0.122 | 0.136 | 0.124 | 0.142 | 0.108 | 0.082 | 0.106 | 0.122 | 0.124 |
| YOLOv6 | 0.13 | 0.121 | 0.129 | 0.139 | 0.119 | 0.118 | 0.126 | 0.123 | 0.128 | 0.118 | 0.115 | 0.116 | 0.119 | 0.123 |
| YOLOv7 | 0.125 | 0.115 | 0.129 | 0.117 | 0.083 | 0.101 | 0.115 | 0.105 | 0.129 | 0.101 | 0.091 | 0.091 | 0.111 | 0.115 |
| YOLOv8 | 0.12 | 0.113 | 0.12 | 0.128 | 0.111 | 0.111 | 0.115 | 0.111 | 0.119 | 0.109 | 0.106 | 0.106 | 0.111 | 0.112 |
| YOLOX | 0.13 | 0.122 | 0.129 | 0.138 | 0.106 | 0.117 | 0.129 | 0.12 | 0.137 | 0.121 | 0.103 | 0.112 | 0.117 | 0.123 |
| Faster R-CNN | 0.159 | 0.166 | 0.146 | 0.114 | 0.11 | 0.119 | 0.164 | 0.136 | 0.133 | 0.122 | 0.083 | 0.121 | 0.152 | 0.158 |
| Cascade R-CNN | 0.171 | 0.166 | 0.165 | 0.12 | 0.122 | 0.141 | 0.17 | 0.136 | 0.15 | 0.136 | 0.09 | 0.131 | 0.154 | 0.165 |
| Cascade RPN | 0.163 | 0.161 | 0.153 | 0.105 | 0.109 | 0.138 | 0.153 | 0.14 | 0.142 | 0.13 | 0.082 | 0.129 | 0.151 | 0.162 |
| Double Heads | 0.169 | 0.161 | 0.164 | 0.12 | 0.11 | 0.134 | 0.165 | 0.128 | 0.144 | 0.128 | 0.085 | 0.126 | 0.151 | 0.16 |
| FPG | 0.133 | 0.136 | 0.132 | 0.121 | 0.109 | 0.112 | 0.141 | 0.115 | 0.124 | 0.105 | 0.073 | 0.112 | 0.123 | 0.126 |
| Grid R-CNN | 0.151 | 0.145 | 0.145 | 0.109 | 0.11 | 0.123 | 0.15 | 0.13 | 0.139 | 0.116 | 0.091 | 0.114 | 0.137 | 0.144 |
| Guided Anchoring | 0.142 | 0.162 | 0.153 | 0.119 | 0.107 | 0.137 | 0.163 | 0.136 | 0.132 | 0.123 | 0.094 | 0.128 | 0.146 | 0.151 |
| HRNet | 0.147 | 0.139 | 0.146 | 0.132 | 0.105 | 0.117 | 0.141 | 0.133 | 0.122 | 0.132 | 0.109 | 0.117 | 0.122 | 0.144 |
| Libra R-CNN | 0.155 | 0.145 | 0.136 | 0.133 | 0.119 | 0.119 | 0.134 | 0.123 | 0.126 | 0.121 | 0.102 | 0.115 | 0.12 | 0.135 |
| PAFPN | 0.163 | 0.157 | 0.158 | 0.118 | 0.12 | 0.124 | 0.156 | 0.137 | 0.137 | 0.125 | 0.086 | 0.121 | 0.149 | 0.165 |
| RepPoints | 0.178 | 0.177 | 0.166 | 0.124 | 0.14 | 0.131 | 0.167 | 0.135 | 0.146 | 0.132 | 0.099 | 0.129 | 0.154 | 0.174 |
| Res2Net | 0.123 | 0.115 | 0.122 | 0.112 | 0.075 | 0.106 | 0.132 | 0.113 | 0.115 | 0.1 | 0.086 | 0.099 | 0.111 | 0.12 |
| ResNeSt | 0.125 | 0.132 | 0.118 | 0.11 | 0.11 | 0.109 | 0.125 | 0.108 | 0.102 | 0.101 | 0.091 | 0.1 | 0.118 | 0.127 |
| SABL | 0.177 | 0.169 | 0.166 | 0.118 | 0.133 | 0.145 | 0.166 | 0.151 | 0.147 | 0.143 | 0.096 | 0.141 | 0.161 | 0.172 |
| Sparse R-CNN | 0.135 | 0.134 | 0.138 | 0.104 | 0.092 | 0.114 | 0.141 | 0.106 | 0.111 | 0.109 | 0.095 | 0.103 | 0.122 | 0.131 |
| DETR | 0.171 | 0.151 | 0.151 | 0.101 | 0.088 | 0.142 | 0.152 | 0.146 | 0.126 | 0.138 | 0.095 | 0.128 | 0.142 | 0.151 |
| Conditional DETR | 0.157 | 0.138 | 0.128 | 0.1 | 0.09 | 0.114 | 0.129 | 0.112 | 0.123 | 0.108 | 0.068 | 0.105 | 0.111 | 0.135 |
| DDQ | 0.119 | 0.119 | 0.124 | 0.099 | 0.099 | 0.112 | 0.124 | 0.116 | 0.113 | 0.11 | 0.085 | 0.113 | 0.117 | 0.12 |
| DAB-DETR | 0.118 | 0.111 | 0.117 | 0.092 | 0.094 | 0.099 | 0.111 | 0.095 | 0.101 | 0.098 | 0.071 | 0.087 | 0.103 | 0.117 |
| Deformable DETR | 0.136 | 0.129 | 0.142 | 0.093 | 0.107 | 0.118 | 0.134 | 0.128 | 0.12 | 0.12 | 0.073 | 0.114 | 0.137 | 0.121 |
| DINO | 0.112 | 0.11 | 0.112 | 0.086 | 0.068 | 0.097 | 0.113 | 0.106 | 0.098 | 0.101 | 0.083 | 0.097 | 0.1 | 0.112 |
| PVT | 0.14 | 0.134 | 0.117 | 0.107 | 0.099 | 0.108 | 0.115 | 0.099 | 0.102 | 0.111 | 0.106 | 0.109 | 0.113 | 0.12 |
| PVTv2 | 0.156 | 0.132 | 0.119 | 0.119 | 0.112 | 0.108 | 0.126 | 0.118 | 0.122 | 0.11 | 0.097 | 0.108 | 0.111 | 0.129 |

Table 11: **Overall** experimental results of **person** detection in the metric of **mAR50(%)**.

| | Clean | Random | AdvCam | UPC | NatPatch | MTD | LAP | InvisCloak | DAP | AdvTshirt | AdvTexture | AdvPatch | AdvPattern | AdvCaT |
|---|---|---|---|---|---|---|---|---|---|---|---|---|---|---|
| ATSS | 0.835 | 0.827 | 0.823 | 0.802 | 0.782 | 0.823 | 0.818 | 0.789 | 0.788 | 0.798 | 0.741 | 0.786 | 0.823 | 0.844 |
| AutoAssign | 0.854 | 0.837 | 0.841 | 0.794 | 0.827 | 0.832 | 0.826 | 0.814 | 0.817 | 0.827 | 0.793 | 0.811 | 0.832 | 0.851 |
| CenterNet | 0.848 | 0.835 | 0.84 | 0.849 | 0.809 | 0.842 | 0.829 | 0.823 | 0.822 | 0.828 | 0.794 | 0.82 | 0.848 | 0.858 |
| CentripetalNet | 0.854 | 0.858 | 0.838 | 0.809 | 0.777 | 0.85 | 0.83 | 0.86 | 0.82 | 0.814 | 0.737 | 0.841 | 0.852 | 0.854 |
| CornerNet | 0.871 | 0.87 | 0.853 | 0.814 | 0.778 | 0.853 | 0.841 | 0.848 | 0.826 | 0.805 | 0.74 | 0.817 | 0.857 | 0.858 |
| DDOD | 0.737 | 0.729 | 0.739 | 0.723 | 0.746 | 0.716 | 0.732 | 0.713 | 0.725 | 0.72 | 0.678 | 0.694 | 0.72 | 0.742 |
| DyHead | 0.725 | 0.731 | 0.747 | 0.702 | 0.7 | 0.723 | 0.745 | 0.711 | 0.711 | 0.725 | 0.724 | 0.716 | 0.727 | 0.744 |
| EfficientNet | 0.794 | 0.771 | 0.789 | 0.768 | 0.751 | 0.758 | 0.781 | 0.762 | 0.754 | 0.762 | 0.748 | 0.752 | 0.766 | 0.793 |
| FCOS | 0.897 | 0.905 | 0.89 | 0.864 | 0.86 | 0.891 | 0.894 | 0.893 | 0.858 | 0.897 | 0.82 | 0.898 | 0.899 | 0.908 |
| FoveaBox | 0.824 | 0.814 | 0.836 | 0.836 | 0.794 | 0.825 | 0.813 | 0.799 | 0.819 | 0.81 | 0.785 | 0.776 | 0.818 | 0.836 |
| FreeAnchor | 0.792 | 0.801 | 0.788 | 0.791 | 0.772 | 0.786 | 0.789 | 0.768 | 0.769 | 0.781 | 0.717 | 0.784 | 0.793 | 0.809 |
| FSAF | 0.812 | 0.814 | 0.818 | 0.803 | 0.79 | 0.807 | 0.822 | 0.808 | 0.808 | 0.803 | 0.785 | 0.803 | 0.805 | 0.812 |
| GFL | 0.827 | 0.803 | 0.805 | 0.796 | 0.782 | 0.806 | 0.784 | 0.781 | 0.763 | 0.785 | 0.743 | 0.777 | 0.804 | 0.823 |
| LD | 0.809 | 0.797 | 0.802 | 0.785 | 0.762 | 0.787 | 0.786 | 0.764 | 0.77 | 0.777 | 0.735 | 0.763 | 0.797 | 0.813 |
| NAS-FPN | 0.773 | 0.765 | 0.775 | 0.726 | 0.75 | 0.74 | 0.768 | 0.744 | 0.732 | 0.741 | 0.7 | 0.751 | 0.76 | 0.778 |
| PAA | 0.816 | 0.819 | 0.821 | 0.808 | 0.791 | 0.805 | 0.805 | 0.792 | 0.784 | 0.78 | 0.73 | 0.783 | 0.807 | 0.837 |
| RetinaNet | 0.838 | 0.823 | 0.843 | 0.817 | 0.795 | 0.823 | 0.822 | 0.802 | 0.796 | 0.818 | 0.777 | 0.791 | 0.823 | 0.848 |
| RTMDet | 0.976 | 0.978 | 0.974 | 0.973 | 0.972 | 0.975 | 0.974 | 0.966 | 0.971 | 0.97 | 0.966 | 0.976 | 0.977 | 0.976 |
| TOOD | 0.73 | 0.732 | 0.731 | 0.708 | 0.691 | 0.734 | 0.737 | 0.716 | 0.723 | 0.724 | 0.69 | 0.714 | 0.723 | 0.733 |
| VarifocalNet | 0.794 | 0.772 | 0.786 | 0.767 | 0.738 | 0.765 | 0.77 | 0.744 | 0.755 | 0.758 | 0.729 | 0.745 | 0.771 | 0.791 |
| YOLOv5 | 0.814 | 0.832 | 0.836 | 0.773 | 0.782 | 0.825 | 0.831 | 0.841 | 0.814 | 0.829 | 0.817 | 0.836 | 0.831 | 0.843 |
| YOLOv6 | 0.96 | 0.963 | 0.957 | 0.951 | 0.939 | 0.959 | 0.943 | 0.964 | 0.943 | 0.95 | 0.921 | 0.956 | 0.963 | 0.966 |
| YOLOv7 | 0.729 | 0.73 | 0.744 | 0.723 | 0.7 | 0.716 | 0.731 | 0.712 | 0.721 | 0.731 | 0.721 | 0.723 | 0.722 | 0.743 |
| YOLOv8 | 0.677 | 0.667 | 0.686 | 0.67 | 0.669 | 0.676 | 0.676 | 0.679 | 0.687 | 0.674 | 0.675 | 0.674 | 0.668 | 0.679 |
| YOLOX | 0.685 | 0.697 | 0.702 | 0.702 | 0.702 | 0.7 | 0.708 | 0.706 | 0.709 | 0.707 | 0.731 | 0.71 | 0.682 | 0.692 |
| Faster R-CNN | 0.69 | 0.682 | 0.691 | 0.625 | 0.659 | 0.679 | 0.674 | 0.653 | 0.645 | 0.679 | 0.615 | 0.639 | 0.67 | 0.685 |
| Cascade R-CNN | 0.699 | 0.689 | 0.703 | 0.644 | 0.652 | 0.679 | 0.695 | 0.66 | 0.673 | 0.677 | 0.623 | 0.651 | 0.68 | 0.7 |
| Cascade RPN | 0.785 | 0.791 | 0.789 | 0.759 | 0.758 | 0.78 | 0.776 | 0.748 | 0.751 | 0.774 | 0.715 | 0.758 | 0.784 | 0.797 |
| Double Heads | 0.696 | 0.688 | 0.7 | 0.636 | 0.639 | 0.677 | 0.678 | 0.664 | 0.654 | 0.679 | 0.623 | 0.643 | 0.676 | 0.696 |
| FPG | 0.656 | 0.659 | 0.665 | 0.598 | 0.598 | 0.643 | 0.659 | 0.636 | 0.622 | 0.642 | 0.588 | 0.635 | 0.659 | 0.661 |
| Grid R-CNN | 0.677 | 0.664 | 0.67 | 0.638 | 0.635 | 0.666 | 0.669 | 0.654 | 0.655 | 0.659 | 0.632 | 0.644 | 0.657 | 0.661 |
| Guided Anchoring | 0.74 | 0.732 | 0.748 | 0.729 | 0.709 | 0.738 | 0.745 | 0.733 | 0.711 | 0.736 | 0.72 | 0.718 | 0.739 | 0.739 |
| HRNet | 0.665 | 0.669 | 0.679 | 0.645 | 0.644 | 0.671 | 0.673 | 0.661 | 0.646 | 0.676 | 0.65 | 0.645 | 0.649 | 0.678 |
| Libra R-CNN | 0.829 | 0.813 | 0.831 | 0.822 | 0.766 | 0.81 | 0.811 | 0.781 | 0.776 | 0.797 | 0.754 | 0.778 | 0.819 | 0.848 |
| PAFPN | 0.685 | 0.682 | 0.697 | 0.644 | 0.647 | 0.678 | 0.679 | 0.664 | 0.658 | 0.682 | 0.614 | 0.651 | 0.679 | 0.698 |
| RepPoints | 0.826 | 0.813 | 0.822 | 0.816 | 0.813 | 0.804 | 0.807 | 0.799 | 0.808 | 0.801 | 0.783 | 0.779 | 0.813 | 0.828 |
| Res2Net | 0.636 | 0.63 | 0.638 | 0.599 | 0.543 | 0.627 | 0.623 | 0.614 | 0.602 | 0.623 | 0.578 | 0.618 | 0.63 | 0.634 |
| ResNeSt | 0.648 | 0.651 | 0.625 | 0.61 | 0.61 | 0.636 | 0.619 | 0.614 | 0.586 | 0.607 | 0.593 | 0.609 | 0.64 | 0.654 |
| SABL | 0.711 | 0.704 | 0.707 | 0.65 | 0.66 | 0.69 | 0.678 | 0.675 | 0.675 | 0.688 | 0.635 | 0.661 | 0.696 | 0.706 |
| Sparse R-CNN | 0.702 | 0.685 | 0.694 | 0.67 | 0.668 | 0.672 | 0.688 | 0.65 | 0.644 | 0.667 | 0.649 | 0.651 | 0.68 | 0.694 |
| DETR | 0.893 | 0.9 | 0.877 | 0.761 | 0.798 | 0.904 | 0.887 | 0.891 | 0.827 | 0.882 | 0.789 | 0.889 | 0.895 | 0.892 |
| Conditional DETR | 0.73 | 0.729 | 0.717 | 0.706 | 0.708 | 0.697 | 0.703 | 0.67 | 0.691 | 0.683 | 0.643 | 0.692 | 0.69 | 0.712 |
| DDQ | 0.774 | 0.774 | 0.768 | 0.768 | 0.758 | 0.746 | 0.757 | 0.728 | 0.744 | 0.748 | 0.725 | 0.745 | 0.746 | 0.773 |
| DAB-DETR | 0.743 | 0.739 | 0.745 | 0.737 | 0.76 | 0.729 | 0.742 | 0.705 | 0.73 | 0.717 | 0.703 | 0.716 | 0.722 | 0.743 |
| Deformable DETR | 0.703 | 0.699 | 0.707 | 0.68 | 0.693 | 0.671 | 0.692 | 0.666 | 0.675 | 0.673 | 0.591 | 0.67 | 0.679 | 0.693 |
| DINO | 0.739 | 0.748 | 0.744 | 0.735 | 0.747 | 0.732 | 0.747 | 0.723 | 0.72 | 0.727 | 0.718 | 0.734 | 0.73 | 0.736 |
| PVT | 0.703 | 0.705 | 0.7 | 0.692 | 0.673 | 0.693 | 0.696 | 0.679 | 0.669 | 0.692 | 0.681 | 0.691 | 0.688 | 0.699 |
| PVTv2 | 0.738 | 0.733 | 0.714 | 0.736 | 0.731 | 0.712 | 0.733 | 0.715 | 0.705 | 0.725 | 0.7 | 0.716 | 0.709 | 0.731 |

Table 12: **Overall** experimental results of **person** detection in the metric of **mAR50:95(%)**.

| | Clean | Random | AdvCam | UPC | NatPatch | MTD | LAP | InvisCloak | DAP | AdvTshirt | AdvTexture | AdvPatch | AdvPattern | AdvCaT |
|---|---|---|---|---|---|---|---|---|---|---|---|---|---|---|
| ATSS | 0.326 | 0.313 | 0.318 | 0.301 | 0.298 | 0.304 | 0.306 | 0.291 | 0.302 | 0.297 | 0.269 | 0.286 | 0.307 | 0.322 |
| AutoAssign | 0.317 | 0.309 | 0.309 | 0.288 | 0.306 | 0.299 | 0.305 | 0.297 | 0.3 | 0.301 | 0.282 | 0.297 | 0.306 | 0.314 |
| CenterNet | 0.351 | 0.344 | 0.342 | 0.341 | 0.315 | 0.342 | 0.332 | 0.326 | 0.325 | 0.332 | 0.309 | 0.329 | 0.346 | 0.352 |
| CentripetalNet | 0.332 | 0.337 | 0.326 | 0.32 | 0.293 | 0.333 | 0.319 | 0.338 | 0.311 | 0.307 | 0.277 | 0.321 | 0.331 | 0.332 |
| CornerNet | 0.331 | 0.333 | 0.325 | 0.306 | 0.281 | 0.322 | 0.314 | 0.314 | 0.306 | 0.297 | 0.27 | 0.298 | 0.321 | 0.326 |
| DDOD | 0.273 | 0.267 | 0.275 | 0.262 | 0.28 | 0.256 | 0.267 | 0.256 | 0.264 | 0.262 | 0.245 | 0.251 | 0.262 | 0.273 |
| DyHead | 0.283 | 0.284 | 0.294 | 0.267 | 0.273 | 0.275 | 0.293 | 0.273 | 0.279 | 0.279 | 0.27 | 0.274 | 0.279 | 0.291 |
| EfficientNet | 0.29 | 0.283 | 0.295 | 0.283 | 0.284 | 0.277 | 0.291 | 0.278 | 0.284 | 0.28 | 0.274 | 0.272 | 0.276 | 0.291 |
| FCOS | 0.339 | 0.341 | 0.341 | 0.323 | 0.331 | 0.331 | 0.339 | 0.337 | 0.322 | 0.337 | 0.296 | 0.335 | 0.333 | 0.344 |
| FoveaBox | 0.304 | 0.298 | 0.316 | 0.309 | 0.29 | 0.295 | 0.293 | 0.285 | 0.301 | 0.292 | 0.276 | 0.279 | 0.294 | 0.309 |
| FreeAnchor | 0.289 | 0.289 | 0.29 | 0.283 | 0.278 | 0.28 | 0.284 | 0.276 | 0.278 | 0.277 | 0.249 | 0.28 | 0.284 | 0.292 |
| FSAF | 0.307 | 0.305 | 0.31 | 0.298 | 0.304 | 0.299 | 0.307 | 0.299 | 0.307 | 0.294 | 0.279 | 0.297 | 0.301 | 0.305 |
| GFL | 0.311 | 0.298 | 0.304 | 0.29 | 0.29 | 0.292 | 0.29 | 0.283 | 0.285 | 0.284 | 0.267 | 0.281 | 0.295 | 0.308 |
| LD | 0.302 | 0.293 | 0.298 | 0.283 | 0.28 | 0.285 | 0.291 | 0.278 | 0.286 | 0.283 | 0.263 | 0.276 | 0.29 | 0.297 |
| NAS-FPN | 0.294 | 0.28 | 0.293 | 0.265 | 0.278 | 0.27 | 0.285 | 0.271 | 0.272 | 0.271 | 0.242 | 0.272 | 0.278 | 0.288 |
| PAA | 0.321 | 0.322 | 0.323 | 0.308 | 0.303 | 0.312 | 0.32 | 0.312 | 0.313 | 0.303 | 0.269 | 0.304 | 0.318 | 0.33 |
| RetinaNet | 0.317 | 0.32 | 0.327 | 0.314 | 0.305 | 0.312 | 0.316 | 0.309 | 0.307 | 0.312 | 0.295 | 0.304 | 0.312 | 0.327 |
| RTMDet | 0.246 | 0.242 | 0.248 | 0.235 | 0.233 | 0.236 | 0.244 | 0.229 | 0.237 | 0.238 | 0.233 | 0.233 | 0.234 | 0.238 |
| TOOD | 0.277 | 0.276 | 0.278 | 0.262 | 0.258 | 0.271 | 0.277 | 0.266 | 0.274 | 0.269 | 0.249 | 0.265 | 0.268 | 0.278 |
| VarifocalNet | 0.295 | 0.287 | 0.296 | 0.281 | 0.277 | 0.28 | 0.286 | 0.271 | 0.28 | 0.277 | 0.262 | 0.274 | 0.284 | 0.294 |
| YOLOv5 | 0.241 | 0.239 | 0.243 | 0.235 | 0.222 | 0.237 | 0.242 | 0.238 | 0.241 | 0.229 | 0.179 | 0.223 | 0.238 | 0.235 |
| YOLOv6 | 0.241 | 0.237 | 0.24 | 0.232 | 0.233 | 0.231 | 0.236 | 0.232 | 0.238 | 0.231 | 0.228 | 0.228 | 0.232 | 0.233 |
| YOLOv7 | 0.242 | 0.236 | 0.242 | 0.229 | 0.207 | 0.221 | 0.237 | 0.218 | 0.235 | 0.226 | 0.215 | 0.207 | 0.232 | 0.233 |
| YOLOv8 | 0.244 | 0.241 | 0.244 | 0.235 | 0.235 | 0.237 | 0.241 | 0.238 | 0.242 | 0.237 | 0.232 | 0.233 | 0.239 | 0.238 |
| YOLOX | 0.242 | 0.241 | 0.243 | 0.233 | 0.217 | 0.237 | 0.241 | 0.237 | 0.239 | 0.236 | 0.227 | 0.234 | 0.239 | 0.237 |
| Faster R-CNN | 0.256 | 0.255 | 0.261 | 0.228 | 0.242 | 0.246 | 0.253 | 0.24 | 0.244 | 0.25 | 0.212 | 0.233 | 0.247 | 0.255 |
| Cascade R-CNN | 0.262 | 0.255 | 0.267 | 0.238 | 0.246 | 0.248 | 0.261 | 0.244 | 0.256 | 0.249 | 0.224 | 0.24 | 0.249 | 0.258 |
| Cascade RPN | 0.279 | 0.278 | 0.28 | 0.26 | 0.266 | 0.27 | 0.274 | 0.261 | 0.266 | 0.268 | 0.241 | 0.262 | 0.273 | 0.281 |
| Double Heads | 0.26 | 0.255 | 0.262 | 0.233 | 0.232 | 0.243 | 0.253 | 0.24 | 0.245 | 0.244 | 0.209 | 0.233 | 0.249 | 0.258 |
| FPG | 0.248 | 0.243 | 0.251 | 0.217 | 0.211 | 0.233 | 0.247 | 0.235 | 0.235 | 0.231 | 0.196 | 0.229 | 0.242 | 0.241 |
| Grid R-CNN | 0.25 | 0.244 | 0.251 | 0.235 | 0.24 | 0.243 | 0.25 | 0.241 | 0.248 | 0.246 | 0.234 | 0.238 | 0.24 | 0.244 |
| Guided Anchoring | 0.271 | 0.266 | 0.271 | 0.254 | 0.252 | 0.262 | 0.272 | 0.259 | 0.257 | 0.265 | 0.244 | 0.253 | 0.264 | 0.268 |
| HRNet | 0.253 | 0.248 | 0.256 | 0.243 | 0.233 | 0.246 | 0.25 | 0.246 | 0.243 | 0.252 | 0.239 | 0.234 | 0.238 | 0.254 |
| Libra R-CNN | 0.321 | 0.314 | 0.319 | 0.319 | 0.293 | 0.302 | 0.307 | 0.294 | 0.3 | 0.303 | 0.285 | 0.291 | 0.309 | 0.326 |
| PAFPN | 0.257 | 0.253 | 0.263 | 0.232 | 0.243 | 0.245 | 0.255 | 0.243 | 0.251 | 0.249 | 0.212 | 0.235 | 0.25 | 0.259 |
| RepPoints | 0.314 | 0.31 | 0.319 | 0.304 | 0.306 | 0.295 | 0.301 | 0.291 | 0.305 | 0.298 | 0.28 | 0.287 | 0.306 | 0.316 |
| Res2Net | 0.237 | 0.231 | 0.24 | 0.22 | 0.193 | 0.229 | 0.234 | 0.227 | 0.226 | 0.228 | 0.216 | 0.222 | 0.23 | 0.232 |
| ResNeSt | 0.244 | 0.248 | 0.236 | 0.224 | 0.226 | 0.236 | 0.236 | 0.231 | 0.217 | 0.226 | 0.214 | 0.221 | 0.241 | 0.245 |
| SABL | 0.271 | 0.26 | 0.27 | 0.24 | 0.251 | 0.254 | 0.256 | 0.247 | 0.257 | 0.252 | 0.228 | 0.245 | 0.26 | 0.265 |
| Sparse R-CNN | 0.268 | 0.262 | 0.267 | 0.25 | 0.253 | 0.251 | 0.263 | 0.246 | 0.247 | 0.253 | 0.247 | 0.244 | 0.259 | 0.265 |
| DETR | 0.42 | 0.42 | 0.391 | 0.334 | 0.352 | 0.399 | 0.39 | 0.399 | 0.351 | 0.383 | 0.328 | 0.392 | 0.404 | 0.397 |
| Conditional DETR | 0.293 | 0.302 | 0.29 | 0.278 | 0.281 | 0.285 | 0.28 | 0.268 | 0.269 | 0.268 | 0.236 | 0.275 | 0.278 | 0.286 |
| DDQ | 0.322 | 0.32 | 0.32 | 0.303 | 0.304 | 0.301 | 0.313 | 0.29 | 0.29 | 0.305 | 0.284 | 0.299 | 0.3 | 0.322 |
| DAB-DETR | 0.301 | 0.301 | 0.303 | 0.3 | 0.31 | 0.295 | 0.296 | 0.277 | 0.294 | 0.285 | 0.27 | 0.28 | 0.292 | 0.303 |
| Deformable DETR | 0.281 | 0.271 | 0.278 | 0.258 | 0.282 | 0.256 | 0.27 | 0.252 | 0.261 | 0.26 | 0.221 | 0.257 | 0.261 | 0.267 |
| DINO | 0.303 | 0.309 | 0.303 | 0.29 | 0.294 | 0.294 | 0.302 | 0.287 | 0.276 | 0.293 | 0.276 | 0.293 | 0.295 | 0.299 |
| PVT | 0.273 | 0.275 | 0.27 | 0.264 | 0.253 | 0.268 | 0.271 | 0.26 | 0.254 | 0.271 | 0.266 | 0.267 | 0.266 | 0.275 |
| PVTv2 | 0.284 | 0.283 | 0.277 | 0.278 | 0.277 | 0.271 | 0.281 | 0.276 | 0.266 | 0.279 | 0.264 | 0.273 | 0.273 | 0.28 |

Table 13: **Ablation** experimental results (**weather**) of **vehicle** detection in the metric of **mAR50(%)**.

| | Clean | Random | ACTIVE | DTA | FCA | APPA | POOPatch | 3D2Fool | CAMOU | RPAU |
|---|---|---|---|---|---|---|---|---|---|---|
| ATSS | 0.975 | 0.963 | 0.863 | 0.963 | 0.95 | 1.0 | 0.863 | 0.963 | 0.887 | 0.875 |
| AutoAssign | 0.975 | 1.0 | 0.975 | 1.0 | 1.0 | 1.0 | 0.975 | 0.988 | 1.0 | 0.988 |
| CenterNet | 0.95 | 1.0 | 0.963 | 1.0 | 0.975 | 1.0 | 0.887 | 1.0 | 1.0 | 0.925 |
| CentripetalNet | 0.975 | 1.0 | 0.938 | 1.0 | 0.95 | 1.0 | 0.975 | 1.0 | 0.988 | 0.988 |
| CornerNet | 1.0 | 1.0 | 0.95 | 1.0 | 1.0 | 1.0 | 1.0 | 1.0 | 1.0 | 0.95 |
| DDOD | 1.0 | 1.0 | 1.0 | 1.0 | 1.0 | 1.0 | 1.0 | 1.0 | 1.0 | 0.988 |
| DyHead | 0.988 | 1.0 | 1.0 | 1.0 | 1.0 | 1.0 | 1.0 | 0.988 | 0.975 | 1.0 |
| EfficientNet | 0.938 | 1.0 | 1.0 | 1.0 | 1.0 | 0.988 | 1.0 | 1.0 | 1.0 | 0.975 |
| FCOS | 1.0 | 1.0 | 1.0 | 1.0 | 1.0 | 1.0 | 1.0 | 0.988 | 1.0 | 0.975 |
| FoveaBox | 1.0 | 1.0 | 0.95 | 1.0 | 1.0 | 1.0 | 0.988 | 1.0 | 1.0 | 1.0 |
| FreeAnchor | 1.0 | 0.863 | 0.812 | 0.975 | 0.863 | 1.0 | 0.8 | 0.875 | 0.875 | 0.887 |
| FSAF | 0.988 | 0.887 | 0.887 | 0.988 | 0.925 | 1.0 | 0.925 | 0.975 | 0.9 | 0.912 |
| GFL | 0.988 | 0.875 | 0.875 | 1.0 | 0.912 | 1.0 | 0.887 | 0.875 | 0.875 | 0.875 |
| LD | 1.0 | 0.9 | 0.887 | 1.0 | 0.975 | 1.0 | 0.887 | 1.0 | 0.9 | 0.975 |
| NAS-FPN | 1.0 | 0.912 | 1.0 | 1.0 | 0.963 | 1.0 | 0.975 | 1.0 | 0.975 | 0.963 |
| PAA | 0.988 | 1.0 | 0.988 | 1.0 | 0.975 | 1.0 | 0.988 | 1.0 | 1.0 | 1.0 |
| RetinaNet | 0.988 | 0.938 | 0.925 | 0.963 | 0.925 | 1.0 | 0.863 | 0.988 | 0.863 | 0.925 |
| RTMDet | 1.0 | 1.0 | 1.0 | 1.0 | 1.0 | 0.988 | 1.0 | 1.0 | 1.0 | 1.0 |
| TOOD | 1.0 | 1.0 | 0.912 | 1.0 | 1.0 | 1.0 | 0.988 | 1.0 | 1.0 | 1.0 |
| VarifocalNet | 0.988 | 0.938 | 0.85 | 0.887 | 0.95 | 1.0 | 0.925 | 0.85 | 0.875 | 0.887 |
| YOLOv5 | 1.0 | 1.0 | 1.0 | 1.0 | 1.0 | 1.0 | 1.0 | 1.0 | 1.0 | 1.0 |
| YOLOv6 | 1.0 | 1.0 | 1.0 | 1.0 | 1.0 | 0.963 | 1.0 | 1.0 | 1.0 | 1.0 |
| YOLOv7 | 1.0 | 1.0 | 1.0 | 1.0 | 1.0 | 0.925 | 1.0 | 1.0 | 1.0 | 1.0 |
| YOLOv8 | 1.0 | 1.0 | 1.0 | 1.0 | 1.0 | 0.975 | 1.0 | 1.0 | 1.0 | 1.0 |
| YOLOX | 1.0 | 1.0 | 1.0 | 1.0 | 1.0 | 0.988 | 1.0 | 1.0 | 1.0 | 1.0 |
| Faster R-CNN | 0.988 | 0.85 | 0.562 | 0.775 | 0.85 | 0.938 | 0.762 | 0.887 | 0.725 | 0.875 |
| Cascade R-CNN | 1.0 | 0.863 | 0.738 | 0.875 | 0.912 | 0.975 | 0.787 | 0.925 | 0.787 | 0.863 |
| Cascade RPN | 1.0 | 1.0 | 0.988 | 1.0 | 0.988 | 0.975 | 0.988 | 1.0 | 0.988 | 0.988 |
| Double Heads | 0.975 | 0.9 | 0.812 | 0.875 | 0.825 | 1.0 | 0.838 | 0.975 | 0.838 | 0.875 |
| FPG | 0.988 | 0.988 | 0.95 | 1.0 | 0.988 | 1.0 | 0.9 | 0.988 | 0.938 | 0.925 |
| Grid R-CNN | 1.0 | 0.863 | 0.738 | 0.963 | 0.85 | 0.988 | 0.838 | 1.0 | 0.875 | 0.875 |
| Guided Anchoring | 1.0 | 1.0 | 1.0 | 1.0 | 1.0 | 0.988 | 1.0 | 1.0 | 1.0 | 0.975 |
| HRNet | 0.938 | 0.875 | 0.938 | 0.95 | 0.875 | 0.963 | 0.775 | 0.9 | 0.95 | 0.925 |
| Libra R-CNN | 0.988 | 0.938 | 0.938 | 0.963 | 0.963 | 1.0 | 0.975 | 1.0 | 0.9 | 0.975 |
| PAFPN | 0.988 | 0.863 | 0.637 | 0.825 | 0.887 | 0.963 | 0.838 | 0.963 | 0.825 | 0.863 |
| RepPoints | 0.988 | 0.975 | 0.875 | 1.0 | 0.9 | 1.0 | 0.975 | 0.975 | 0.975 | 0.912 |
| Res2Net | 0.975 | 1.0 | 0.925 | 0.988 | 0.988 | 1.0 | 0.912 | 0.938 | 0.988 | 1.0 |
| ResNeSt | 0.975 | 0.863 | 0.825 | 0.988 | 1.0 | 1.0 | 1.0 | 1.0 | 0.863 | 0.963 |
| SABL | 1.0 | 0.875 | 0.775 | 0.938 | 0.887 | 0.988 | 0.787 | 0.95 | 0.85 | 0.863 |
| Sparse R-CNN | 0.975 | 0.912 | 0.85 | 1.0 | 0.988 | 0.988 | 0.95 | 0.95 | 0.838 | 0.9 |
| DETR | 0.95 | 0.537 | 0.388 | 0.237 | 0.8 | 0.713 | 0.812 | 0.5 | 0.487 | 0.8 |
| Conditional DETR | 0.988 | 0.887 | 0.975 | 0.975 | 0.975 | 1.0 | 1.0 | 1.0 | 0.875 | 0.912 |
| DDQ | 0.988 | 1.0 | 1.0 | 1.0 | 1.0 | 1.0 | 1.0 | 1.0 | 1.0 | 1.0 |
| DAB-DETR | 1.0 | 1.0 | 1.0 | 1.0 | 1.0 | 1.0 | 1.0 | 1.0 | 0.975 | 1.0 |
| Deformable DETR | 0.963 | 0.975 | 0.9 | 0.938 | 0.95 | 0.975 | 0.9 | 0.975 | 0.912 | 0.938 |
| DINO | 1.0 | 1.0 | 0.988 | 0.925 | 0.975 | 0.988 | 1.0 | 1.0 | 0.9 | 1.0 |
| PVT | 0.988 | 1.0 | 0.75 | 0.988 | 1.0 | 1.0 | 0.988 | 1.0 | 1.0 | 0.925 |
| PVTv2 | 1.0 | 0.988 | 0.988 | 1.0 | 1.0 | 1.0 | 0.887 | 1.0 | 1.0 | 0.912 |

Table 14: **Ablation** experimental results (**spot**) of **vehicle** detection in the metric of **mAR50(%)**.

| | Clean | Random | ACTIVE | DTA | FCA | APPA | POOPatch | 3D2Fool | CAMOU | RPAU |
|---|---|---|---|---|---|---|---|---|---|---|
| ATSS | 1.0 | 0.979 | 0.885 | 0.99 | 0.969 | 0.99 | 0.844 | 0.969 | 0.875 | 0.875 |
| AutoAssign | 0.958 | 0.99 | 0.99 | 0.99 | 0.99 | 0.99 | 0.969 | 0.99 | 0.99 | 0.99 |
| CenterNet | 0.99 | 0.979 | 0.99 | 0.99 | 0.979 | 0.99 | 0.885 | 0.979 | 0.99 | 0.938 |
| CentripetalNet | 1.0 | 0.99 | 0.979 | 0.99 | 0.99 | 0.99 | 0.979 | 1.0 | 0.99 | 0.979 |
| CornerNet | 0.99 | 0.99 | 0.958 | 0.99 | 0.99 | 0.99 | 0.979 | 1.0 | 0.969 | 0.99 |
| DDOD | 1.0 | 1.0 | 0.99 | 0.99 | 0.99 | 0.99 | 0.99 | 0.99 | 0.99 | 0.99 |
| DyHead | 1.0 | 1.0 | 0.979 | 1.0 | 0.99 | 1.0 | 1.0 | 0.979 | 0.969 | 1.0 |
| EfficientNet | 0.958 | 0.99 | 0.969 | 0.99 | 0.99 | 0.99 | 0.979 | 1.0 | 0.99 | 0.979 |
| FCOS | 1.0 | 0.99 | 0.99 | 0.99 | 0.99 | 0.99 | 0.979 | 0.99 | 0.99 | 0.958 |
| FoveaBox | 0.99 | 0.99 | 0.875 | 0.99 | 0.99 | 0.99 | 0.99 | 0.99 | 0.99 | 0.979 |
| FreeAnchor | 0.99 | 0.865 | 0.833 | 0.979 | 0.885 | 0.99 | 0.792 | 0.938 | 0.885 | 0.865 |
| FSAF | 1.0 | 0.906 | 0.812 | 0.99 | 0.938 | 0.99 | 0.865 | 0.938 | 0.875 | 0.885 |
| GFL | 0.99 | 0.927 | 0.969 | 0.99 | 0.938 | 0.99 | 0.885 | 0.927 | 0.917 | 0.896 |
| LD | 0.99 | 0.948 | 0.896 | 0.99 | 0.958 | 0.99 | 0.823 | 0.979 | 0.896 | 0.99 |
| NAS-FPN | 1.0 | 0.885 | 0.969 | 0.979 | 0.948 | 1.0 | 0.979 | 1.0 | 0.927 | 0.917 |
| PAA | 0.99 | 0.99 | 0.979 | 0.99 | 0.99 | 0.99 | 0.958 | 0.99 | 0.99 | 0.99 |
| RetinaNet | 0.99 | 0.865 | 0.865 | 0.917 | 0.948 | 0.979 | 0.854 | 0.917 | 0.802 | 0.875 |
| RTMDet | 1.0 | 1.0 | 0.99 | 1.0 | 0.99 | 0.979 | 0.99 | 0.99 | 1.0 | 0.99 |
| TOOD | 0.99 | 0.99 | 0.927 | 0.99 | 0.99 | 0.99 | 0.99 | 0.99 | 0.99 | 0.99 |
| VarifocalNet | 1.0 | 0.958 | 0.802 | 0.865 | 0.938 | 0.99 | 0.885 | 0.823 | 0.833 | 0.875 |
| YOLOv5 | 1.0 | 0.99 | 0.99 | 1.0 | 0.99 | 1.0 | 1.0 | 0.99 | 0.99 | 0.969 |
| YOLOv6 | 1.0 | 1.0 | 1.0 | 1.0 | 1.0 | 0.979 | 1.0 | 1.0 | 1.0 | 1.0 |
| YOLOv7 | 1.0 | 1.0 | 1.0 | 0.99 | 0.99 | 0.917 | 1.0 | 1.0 | 0.99 | 0.99 |
| YOLOv8 | 1.0 | 1.0 | 1.0 | 1.0 | 1.0 | 0.958 | 1.0 | 1.0 | 0.99 | 1.0 |
| YOLOX | 1.0 | 0.99 | 1.0 | 0.99 | 0.99 | 0.958 | 0.99 | 1.0 | 0.99 | 0.99 |
| Faster R-CNN | 0.99 | 0.865 | 0.74 | 0.969 | 0.885 | 0.99 | 0.792 | 0.948 | 0.833 | 0.885 |
| Cascade R-CNN | 0.99 | 0.865 | 0.854 | 0.969 | 0.885 | 0.99 | 0.823 | 0.917 | 0.844 | 0.854 |
| Cascade RPN | 0.99 | 0.958 | 0.938 | 0.99 | 0.938 | 0.969 | 0.906 | 0.99 | 0.99 | 0.958 |
| Double Heads | 0.99 | 0.896 | 0.927 | 0.969 | 0.823 | 0.99 | 0.865 | 0.99 | 0.844 | 0.865 |
| FPG | 1.0 | 0.948 | 0.896 | 0.979 | 0.958 | 0.99 | 0.833 | 0.99 | 0.969 | 0.906 |
| Grid R-CNN | 0.979 | 0.906 | 0.885 | 0.979 | 0.917 | 0.979 | 0.875 | 0.99 | 0.917 | 0.896 |
| Guided Anchoring | 1.0 | 0.979 | 0.99 | 0.99 | 0.99 | 0.917 | 0.979 | 0.99 | 0.99 | 0.979 |
| HRNet | 0.969 | 0.906 | 0.958 | 0.958 | 0.854 | 0.969 | 0.812 | 0.958 | 0.969 | 0.927 |
| Libra R-CNN | 1.0 | 0.958 | 0.917 | 0.99 | 0.948 | 0.99 | 0.948 | 0.99 | 0.854 | 0.948 |
| PAFPN | 0.99 | 0.875 | 0.74 | 0.938 | 0.938 | 0.969 | 0.927 | 0.979 | 0.844 | 0.844 |
| RepPoints | 0.99 | 0.979 | 0.823 | 0.99 | 0.885 | 0.99 | 0.948 | 0.979 | 0.927 | 0.896 |
| Res2Net | 0.979 | 0.979 | 0.979 | 0.99 | 0.99 | 0.99 | 0.927 | 0.99 | 0.979 | 0.979 |
| ResNeSt | 0.958 | 0.865 | 0.938 | 0.979 | 0.99 | 0.99 | 0.99 | 0.979 | 0.917 | 0.969 |
| SABL | 0.99 | 0.917 | 0.854 | 0.99 | 0.875 | 0.99 | 0.833 | 0.99 | 0.865 | 0.865 |
| Sparse R-CNN | 0.99 | 0.969 | 0.917 | 0.99 | 0.99 | 0.99 | 0.958 | 0.979 | 0.875 | 0.948 |
| DETR | 0.99 | 0.677 | 0.448 | 0.302 | 0.885 | 0.875 | 0.854 | 0.74 | 0.594 | 0.844 |
| Conditional DETR | 0.99 | 0.958 | 0.938 | 0.979 | 0.979 | 0.948 | 0.969 | 0.99 | 0.906 | 0.896 |
| DDQ | 1.0 | 0.99 | 1.0 | 1.0 | 0.99 | 1.0 | 1.0 | 0.99 | 1.0 | 1.0 |
| DAB-DETR | 1.0 | 0.99 | 0.99 | 0.99 | 0.99 | 0.99 | 0.99 | 0.99 | 0.99 | 0.99 |
| Deformable DETR | 1.0 | 0.948 | 0.875 | 0.917 | 0.917 | 0.969 | 0.917 | 0.979 | 0.979 | 0.938 |
| DINO | 1.0 | 0.99 | 0.948 | 0.938 | 0.979 | 0.979 | 0.979 | 0.99 | 0.927 | 0.99 |
| PVT | 0.938 | 0.979 | 0.625 | 0.979 | 0.969 | 0.99 | 0.927 | 0.99 | 0.99 | 0.812 |
| PVTv2 | 1.0 | 0.979 | 0.99 | 0.99 | 0.979 | 0.99 | 0.802 | 0.99 | 0.99 | 0.875 |

Table 15: **Ablation** experimental results (**distance**) of **vehicle** detection in the metric of **mAR50(%)**.

| | Clean | Random | ACTIVE | DTA | FCA | APPA | POOPatch | 3D2Fool | CAMOU | RPAU |
|---|---|---|---|---|---|---|---|---|---|---|
| ATSS | 0.979 | 0.979 | 0.958 | 0.99 | 0.99 | 0.99 | 0.948 | 0.979 | 0.979 | 0.958 |
| AutoAssign | 0.969 | 0.99 | 0.99 | 0.979 | 0.979 | 0.99 | 0.958 | 0.99 | 0.979 | 0.99 |
| CenterNet | 0.99 | 0.979 | 1.0 | 0.99 | 0.979 | 1.0 | 0.99 | 1.0 | 1.0 | 0.969 |
| CentripetalNet | 0.979 | 0.99 | 0.969 | 0.99 | 0.99 | 0.99 | 0.99 | 0.99 | 0.99 | 0.99 |
| CornerNet | 1.0 | 0.99 | 0.99 | 0.99 | 0.99 | 0.979 | 0.99 | 0.99 | 0.99 | 0.99 |
| DDOD | 0.99 | 0.979 | 0.958 | 0.99 | 0.979 | 0.99 | 0.969 | 0.99 | 0.99 | 0.979 |
| DyHead | 0.99 | 0.99 | 0.99 | 0.99 | 0.99 | 0.979 | 0.99 | 0.99 | 0.979 | 0.99 |
| EfficientNet | 0.969 | 0.99 | 0.99 | 0.979 | 0.979 | 0.917 | 0.99 | 0.99 | 0.99 | 0.979 |
| FCOS | 0.99 | 0.99 | 0.99 | 0.979 | 0.99 | 0.99 | 0.969 | 0.99 | 0.99 | 0.969 |
| FoveaBox | 0.99 | 0.99 | 0.979 | 0.99 | 0.99 | 0.99 | 0.99 | 0.99 | 0.99 | 0.979 |
| FreeAnchor | 0.979 | 0.917 | 0.865 | 0.979 | 0.927 | 0.99 | 0.802 | 0.969 | 0.802 | 0.917 |
| FSAF | 1.0 | 0.917 | 0.885 | 0.99 | 0.958 | 0.979 | 0.896 | 0.99 | 0.917 | 0.906 |
| GFL | 0.979 | 0.969 | 0.948 | 0.99 | 0.969 | 0.979 | 0.958 | 0.99 | 0.896 | 0.969 |
| LD | 0.979 | 0.969 | 0.969 | 0.99 | 0.979 | 0.99 | 0.958 | 0.99 | 0.969 | 0.979 |
| NAS-FPN | 0.99 | 0.969 | 0.979 | 0.979 | 0.979 | 0.99 | 0.99 | 0.99 | 0.948 | 0.958 |
| PAA | 0.99 | 1.0 | 0.979 | 0.99 | 0.99 | 0.979 | 0.969 | 0.99 | 0.99 | 0.99 |
| RetinaNet | 0.979 | 0.917 | 0.917 | 0.969 | 0.979 | 0.979 | 0.875 | 0.979 | 0.875 | 0.927 |
| RTMDet | 0.99 | 0.99 | 1.0 | 0.938 | 0.99 | 0.958 | 1.0 | 1.0 | 0.99 | 0.99 |
| TOOD | 0.99 | 0.99 | 0.948 | 0.979 | 0.979 | 0.99 | 0.99 | 0.99 | 0.99 | 0.979 |
| VarifocalNet | 0.979 | 0.938 | 0.854 | 0.958 | 0.969 | 0.979 | 0.906 | 0.979 | 0.833 | 0.896 |
| YOLOv5 | 0.99 | 1.0 | 1.0 | 0.99 | 1.0 | 0.979 | 0.99 | 1.0 | 0.99 | 1.0 |
| YOLOv6 | 0.99 | 0.99 | 0.99 | 0.969 | 0.99 | 0.99 | 0.99 | 0.99 | 0.979 | 0.99 |
| YOLOv7 | 0.979 | 0.99 | 0.99 | 0.99 | 0.99 | 0.969 | 0.99 | 0.99 | 0.99 | 0.99 |
| YOLOv8 | 0.99 | 0.99 | 0.99 | 0.99 | 0.99 | 0.99 | 0.99 | 0.99 | 0.99 | 0.99 |
| YOLOX | 0.979 | 0.99 | 0.99 | 0.99 | 0.99 | 0.99 | 0.99 | 0.99 | 0.99 | 0.99 |
| Faster R-CNN | 0.979 | 0.885 | 0.76 | 0.99 | 0.958 | 0.979 | 0.802 | 0.958 | 0.875 | 0.896 |
| Cascade R-CNN | 0.99 | 0.906 | 0.823 | 0.979 | 0.948 | 0.99 | 0.792 | 0.958 | 0.875 | 0.896 |
| Cascade RPN | 0.99 | 0.979 | 0.958 | 0.979 | 0.979 | 0.99 | 0.927 | 0.99 | 0.979 | 0.958 |
| Double Heads | 0.99 | 0.948 | 0.885 | 0.99 | 0.896 | 0.99 | 0.875 | 0.969 | 0.927 | 0.885 |
| FPG | 0.979 | 0.99 | 0.979 | 0.979 | 0.979 | 0.979 | 0.906 | 0.99 | 0.958 | 0.958 |
| Grid R-CNN | 0.969 | 0.958 | 0.917 | 0.979 | 0.99 | 0.979 | 0.885 | 0.99 | 0.979 | 0.896 |
| Guided Anchoring | 0.99 | 0.99 | 0.99 | 0.99 | 0.99 | 0.958 | 0.99 | 0.99 | 0.99 | 0.979 |
| HRNet | 0.979 | 0.927 | 0.875 | 0.979 | 0.938 | 0.979 | 0.885 | 0.969 | 0.99 | 0.969 |
| Libra R-CNN | 0.99 | 0.969 | 0.969 | 0.979 | 0.979 | 0.979 | 0.969 | 0.99 | 0.958 | 0.938 |
| PAFPN | 0.979 | 0.875 | 0.781 | 0.969 | 0.938 | 0.979 | 0.823 | 0.958 | 0.885 | 0.885 |
| RepPoints | 0.99 | 0.958 | 0.896 | 0.99 | 0.948 | 0.99 | 0.979 | 1.0 | 1.0 | 0.938 |
| Res2Net | 0.979 | 0.99 | 0.979 | 0.99 | 0.979 | 0.979 | 0.979 | 0.99 | 0.99 | 0.979 |
| ResNeSt | 0.979 | 0.927 | 0.958 | 0.969 | 0.969 | 0.99 | 0.979 | 0.969 | 0.917 | 0.969 |
| SABL | 0.99 | 0.885 | 0.844 | 0.979 | 0.938 | 0.979 | 0.781 | 0.948 | 0.958 | 0.896 |
| Sparse R-CNN | 0.979 | 0.979 | 0.865 | 0.99 | 0.99 | 0.979 | 0.938 | 0.99 | 0.906 | 0.979 |
| DETR | 0.979 | 0.771 | 0.615 | 0.604 | 0.896 | 0.708 | 0.885 | 0.635 | 0.625 | 0.906 |
| Conditional DETR | 0.979 | 0.99 | 0.99 | 0.99 | 0.99 | 0.99 | 0.99 | 0.99 | 0.99 | 0.99 |
| DDQ | 0.979 | 0.99 | 0.99 | 0.99 | 0.979 | 0.979 | 0.99 | 0.99 | 0.99 | 0.99 |
| DAB-DETR | 0.99 | 0.99 | 0.99 | 0.99 | 0.99 | 0.99 | 1.0 | 0.979 | 0.99 | 0.99 |
| Deformable DETR | 0.99 | 0.99 | 0.969 | 0.979 | 0.969 | 0.969 | 0.948 | 1.0 | 0.979 | 0.979 |
| DINO | 0.99 | 0.948 | 0.979 | 0.99 | 0.969 | 0.99 | 0.99 | 0.979 | 0.979 | 0.99 |
| PVT | 0.958 | 0.979 | 0.792 | 0.958 | 0.969 | 0.979 | 0.875 | 0.979 | 0.969 | 0.958 |
| PVTv2 | 0.979 | 0.99 | 0.917 | 1.0 | 1.0 | 0.99 | 0.875 | 0.99 | 0.99 | 0.948 |

Table 16: **Ablation** experimental results ($\phi$) of **vehicle** detection in the metric of **mAR50(%)**.

| | Clean | Random | ACTIVE | DTA | FCA | APPA | POOPatch | 3D2Fool | CAMOU | RPAU |
|---|---|---|---|---|---|---|---|---|---|---|
| ATSS | 1.0 | 0.98 | 0.86 | 0.98 | 0.99 | 1.0 | 0.92 | 1.0 | 0.9 | 0.91 |
| AutoAssign | 1.0 | 1.0 | 1.0 | 1.0 | 1.0 | 1.0 | 1.0 | 1.0 | 1.0 | 0.99 |
| CenterNet | 0.99 | 1.0 | 0.98 | 1.0 | 1.0 | 1.0 | 0.98 | 1.0 | 0.99 | 0.98 |
| CentripetalNet | 1.0 | 1.0 | 1.0 | 1.0 | 1.0 | 1.0 | 1.0 | 1.0 | 0.99 | 1.0 |
| CornerNet | 1.0 | 1.0 | 1.0 | 1.0 | 1.0 | 1.0 | 1.0 | 1.0 | 1.0 | 1.0 |
| DDOD | 1.0 | 1.0 | 1.0 | 1.0 | 1.0 | 1.0 | 1.0 | 1.0 | 1.0 | 1.0 |
| DyHead | 1.0 | 1.0 | 0.99 | 1.0 | 1.0 | 1.0 | 1.0 | 1.0 | 1.0 | 1.0 |
| EfficientNet | 1.0 | 1.0 | 0.98 | 1.0 | 1.0 | 1.0 | 1.0 | 1.0 | 1.0 | 1.0 |
| FCOS | 1.0 | 1.0 | 1.0 | 1.0 | 1.0 | 0.99 | 1.0 | 1.0 | 1.0 | 1.0 |
| FoveaBox | 1.0 | 1.0 | 0.95 | 1.0 | 1.0 | 1.0 | 1.0 | 1.0 | 1.0 | 1.0 |
| FreeAnchor | 1.0 | 0.87 | 0.93 | 0.96 | 0.89 | 1.0 | 0.89 | 1.0 | 0.98 | 0.9 |
| FSAF | 1.0 | 0.98 | 0.94 | 0.99 | 0.96 | 1.0 | 0.96 | 1.0 | 0.95 | 0.91 |
| GFL | 1.0 | 0.95 | 0.93 | 0.99 | 1.0 | 1.0 | 0.95 | 1.0 | 0.86 | 0.9 |
| LD | 0.99 | 1.0 | 1.0 | 1.0 | 1.0 | 1.0 | 0.98 | 1.0 | 0.93 | 1.0 |
| NAS-FPN | 1.0 | 1.0 | 0.98 | 0.99 | 0.98 | 1.0 | 0.98 | 1.0 | 0.99 | 0.97 |
| PAA | 1.0 | 1.0 | 1.0 | 1.0 | 1.0 | 1.0 | 1.0 | 1.0 | 1.0 | 1.0 |
| RetinaNet | 1.0 | 0.97 | 0.98 | 0.99 | 1.0 | 1.0 | 0.96 | 1.0 | 0.91 | 0.89 |
| RTMDet | 1.0 | 1.0 | 1.0 | 0.97 | 1.0 | 0.97 | 1.0 | 1.0 | 1.0 | 1.0 |
| TOOD | 1.0 | 1.0 | 0.96 | 1.0 | 1.0 | 1.0 | 1.0 | 1.0 | 1.0 | 1.0 |
| VarifocalNet | 0.99 | 0.94 | 0.97 | 0.98 | 0.99 | 1.0 | 0.94 | 1.0 | 0.88 | 0.91 |
| YOLOv5 | 1.0 | 1.0 | 1.0 | 1.0 | 1.0 | 1.0 | 1.0 | 1.0 | 1.0 | 1.0 |
| YOLOv6 | 1.0 | 1.0 | 1.0 | 1.0 | 1.0 | 1.0 | 1.0 | 1.0 | 1.0 | 1.0 |
| YOLOv7 | 1.0 | 1.0 | 1.0 | 1.0 | 1.0 | 0.97 | 1.0 | 1.0 | 1.0 | 1.0 |
| YOLOv8 | 1.0 | 1.0 | 1.0 | 1.0 | 1.0 | 0.99 | 1.0 | 1.0 | 1.0 | 1.0 |
| YOLOX | 1.0 | 1.0 | 1.0 | 1.0 | 1.0 | 0.98 | 1.0 | 1.0 | 1.0 | 1.0 |
| Faster R-CNN | 1.0 | 0.86 | 0.77 | 0.95 | 0.91 | 1.0 | 0.83 | 1.0 | 0.83 | 0.85 |
| Cascade R-CNN | 1.0 | 0.89 | 0.91 | 0.99 | 0.97 | 1.0 | 0.93 | 0.99 | 0.88 | 0.9 |
| Cascade RPN | 1.0 | 1.0 | 0.97 | 1.0 | 1.0 | 0.99 | 0.96 | 1.0 | 1.0 | 0.93 |
| Double Heads | 1.0 | 1.0 | 0.95 | 1.0 | 0.97 | 1.0 | 0.95 | 1.0 | 0.92 | 0.91 |
| FPG | 1.0 | 1.0 | 0.96 | 0.99 | 1.0 | 0.99 | 0.97 | 1.0 | 1.0 | 0.99 |
| Grid R-CNN | 0.98 | 0.97 | 0.89 | 1.0 | 0.95 | 0.99 | 0.95 | 1.0 | 0.94 | 0.9 |
| Guided Anchoring | 1.0 | 1.0 | 1.0 | 1.0 | 1.0 | 0.99 | 1.0 | 1.0 | 1.0 | 1.0 |
| HRNet | 1.0 | 0.98 | 1.0 | 0.99 | 0.94 | 0.99 | 0.84 | 1.0 | 0.98 | 0.95 |
| Libra R-CNN | 1.0 | 1.0 | 0.91 | 1.0 | 0.97 | 1.0 | 1.0 | 1.0 | 0.96 | 0.96 |
| PAFPN | 1.0 | 0.93 | 0.83 | 0.97 | 0.99 | 1.0 | 0.95 | 1.0 | 0.9 | 0.9 |
| RepPoints | 1.0 | 1.0 | 0.89 | 1.0 | 0.95 | 1.0 | 1.0 | 1.0 | 0.96 | 0.95 |
| Res2Net | 0.98 | 1.0 | 1.0 | 0.99 | 1.0 | 1.0 | 1.0 | 1.0 | 0.93 | 1.0 |
| ResNeSt | 0.96 | 0.85 | 0.94 | 0.99 | 1.0 | 0.96 | 0.99 | 0.99 | 0.9 | 0.99 |
| SABL | 1.0 | 1.0 | 0.89 | 0.99 | 0.96 | 1.0 | 0.92 | 1.0 | 0.89 | 0.89 |
| Sparse R-CNN | 1.0 | 1.0 | 0.88 | 0.99 | 1.0 | 0.98 | 1.0 | 1.0 | 0.92 | 0.95 |
| DETR | 1.0 | 0.84 | 0.72 | 0.49 | 0.93 | 0.93 | 0.96 | 0.87 | 0.81 | 0.94 |
| Conditional DETR | 1.0 | 1.0 | 1.0 | 1.0 | 1.0 | 1.0 | 1.0 | 1.0 | 1.0 | 1.0 |
| DDQ | 1.0 | 1.0 | 1.0 | 1.0 | 1.0 | 1.0 | 1.0 | 1.0 | 1.0 | 1.0 |
| DAB-DETR | 1.0 | 1.0 | 1.0 | 1.0 | 1.0 | 1.0 | 1.0 | 1.0 | 1.0 | 1.0 |
| Deformable DETR | 0.99 | 1.0 | 0.99 | 1.0 | 0.99 | 1.0 | 0.99 | 1.0 | 0.98 | 1.0 |
| DINO | 1.0 | 1.0 | 0.99 | 1.0 | 1.0 | 1.0 | 1.0 | 1.0 | 0.98 | 1.0 |
| PVT | 1.0 | 0.98 | 0.7 | 0.98 | 0.96 | 1.0 | 0.92 | 1.0 | 0.96 | 0.85 |
| PVTv2 | 1.0 | 1.0 | 1.0 | 1.0 | 1.0 | 1.0 | 0.82 | 1.0 | 1.0 | 0.97 |

Table 17: **Ablation** experimental results ($\theta$) of **vehicle** detection in the metric of **mAR50(%)**.

| | Clean | Random | ACTIVE | DTA | FCA | APPA | POOPatch | 3D2Fool | CAMOU | RPAU |
|---|---|---|---|---|---|---|---|---|---|---|
| ATSS | 0.98 | 0.96 | 0.47 | 0.73 | 0.94 | 0.85 | 0.77 | 1.0 | 0.76 | 0.7 |
| AutoAssign | 1.0 | 0.86 | 0.74 | 0.92 | 0.91 | 0.97 | 0.71 | 0.93 | 0.87 | 0.98 |
| CenterNet | 1.0 | 1.0 | 0.62 | 0.98 | 0.98 | 0.99 | 0.71 | 0.91 | 0.92 | 1.0 |
| CentripetalNet | 1.0 | 1.0 | 0.78 | 1.0 | 0.96 | 1.0 | 0.74 | 1.0 | 0.88 | 0.98 |
| CornerNet | 0.99 | 0.79 | 0.49 | 0.75 | 0.93 | 0.78 | 0.51 | 0.95 | 0.66 | 0.6 |
| DDOD | 0.99 | 0.97 | 0.61 | 0.99 | 0.85 | 0.98 | 1.0 | 1.0 | 0.96 | 0.97 |
| DyHead | 1.0 | 0.81 | 0.52 | 0.53 | 0.98 | 0.81 | 0.61 | 0.55 | 0.62 | 0.93 |
| EfficientNet | 1.0 | 0.79 | 0.81 | 1.0 | 0.84 | 1.0 | 0.78 | 1.0 | 0.97 | 1.0 |
| FCOS | 1.0 | 0.94 | 1.0 | 1.0 | 1.0 | 1.0 | 1.0 | 1.0 | 1.0 | 1.0 |
| FoveaBox | 1.0 | 1.0 | 0.49 | 0.96 | 0.87 | 0.99 | 0.67 | 0.87 | 0.96 | 0.92 |
| FreeAnchor | 1.0 | 0.82 | 1.0 | 0.98 | 0.9 | 1.0 | 0.99 | 0.98 | 0.88 | 1.0 |
| FSAF | 1.0 | 1.0 | 0.64 | 1.0 | 0.82 | 1.0 | 0.99 | 1.0 | 1.0 | 1.0 |
| GFL | 1.0 | 0.97 | 0.45 | 0.95 | 0.93 | 0.96 | 0.94 | 0.98 | 0.73 | 1.0 |
| LD | 1.0 | 0.99 | 0.56 | 1.0 | 1.0 | 0.93 | 0.98 | 0.88 | 0.96 | 1.0 |
| NAS-FPN | 1.0 | 0.92 | 0.52 | 0.86 | 1.0 | 0.79 | 0.72 | 0.94 | 0.65 | 0.92 |
| PAA | 1.0 | 1.0 | 0.97 | 0.99 | 0.94 | 1.0 | 0.97 | 1.0 | 1.0 | 1.0 |
| RetinaNet | 1.0 | 1.0 | 1.0 | 1.0 | 1.0 | 1.0 | 1.0 | 1.0 | 1.0 | 1.0 |
| RTMDet | 1.0 | 1.0 | 0.99 | 0.98 | 1.0 | 1.0 | 1.0 | 1.0 | 1.0 | 0.88 |
| TOOD | 0.83 | 0.85 | 0.52 | 0.87 | 0.76 | 0.84 | 0.8 | 0.97 | 0.96 | 0.8 |
| VarifocalNet | 1.0 | 0.77 | 0.48 | 0.55 | 0.86 | 0.83 | 0.64 | 0.62 | 0.61 | 0.65 |
| YOLOv5 | 1.0 | 1.0 | 0.74 | 1.0 | 1.0 | 1.0 | 0.92 | 1.0 | 1.0 | 1.0 |
| YOLOv6 | 1.0 | 1.0 | 0.73 | 1.0 | 1.0 | 1.0 | 1.0 | 1.0 | 1.0 | 1.0 |
| YOLOv7 | 0.98 | 0.76 | 0.78 | 0.72 | 0.91 | 1.0 | 0.78 | 1.0 | 0.75 | 0.79 |
| YOLOv8 | 1.0 | 0.92 | 0.64 | 0.82 | 1.0 | 0.82 | 0.64 | 1.0 | 0.83 | 0.97 |
| YOLOX | 1.0 | 0.9 | 0.62 | 0.92 | 0.95 | 0.99 | 0.88 | 1.0 | 0.97 | 0.98 |
| Faster R-CNN | 0.85 | 0.49 | 0.46 | 0.52 | 0.66 | 0.73 | 0.53 | 0.63 | 0.55 | 0.49 |
| Cascade R-CNN | 0.75 | 0.64 | 0.47 | 0.54 | 0.67 | 0.67 | 0.62 | 0.65 | 0.55 | 0.6 |
| Cascade RPN | 1.0 | 0.92 | 0.51 | 0.79 | 0.88 | 1.0 | 0.96 | 1.0 | 0.85 | 0.82 |
| Double Heads | 0.76 | 0.72 | 0.46 | 0.57 | 0.68 | 0.71 | 0.63 | 0.87 | 0.6 | 0.62 |
| FPG | 1.0 | 0.89 | 0.5 | 0.93 | 0.97 | 0.9 | 0.72 | 0.91 | 0.96 | 0.53 |
| Grid R-CNN | 0.91 | 0.68 | 0.48 | 0.68 | 0.71 | 0.77 | 0.63 | 0.85 | 0.6 | 0.56 |
| Guided Anchoring | 1.0 | 1.0 | 0.87 | 0.99 | 1.0 | 1.0 | 0.8 | 1.0 | 1.0 | 0.9 |
| HRNet | 0.96 | 0.78 | 0.55 | 0.59 | 0.73 | 0.83 | 0.62 | 0.81 | 0.62 | 0.62 |
| Libra R-CNN | 1.0 | 1.0 | 0.67 | 1.0 | 1.0 | 1.0 | 1.0 | 1.0 | 1.0 | 1.0 |
| PAFPN | 0.74 | 0.63 | 0.4 | 0.51 | 0.63 | 0.71 | 0.62 | 0.66 | 0.57 | 0.44 |
| RepPoints | 1.0 | 1.0 | 0.63 | 1.0 | 0.94 | 0.77 | 1.0 | 1.0 | 1.0 | 0.9 |
| Res2Net | 0.9 | 0.64 | 0.57 | 0.55 | 0.73 | 0.65 | 0.66 | 0.77 | 0.66 | 0.72 |
| ResNeSt | 0.97 | 0.61 | 0.5 | 0.73 | 0.72 | 0.96 | 0.61 | 0.93 | 0.61 | 0.58 |
| SABL | 0.77 | 0.53 | 0.45 | 0.54 | 0.66 | 0.66 | 0.61 | 0.66 | 0.56 | 0.47 |
| Sparse R-CNN | 1.0 | 0.96 | 0.63 | 0.91 | 1.0 | 0.72 | 0.8 | 0.78 | 0.87 | 0.85 |
| DETR | 0.77 | 0.56 | 0.39 | 0.45 | 0.71 | 0.64 | 0.51 | 0.55 | 0.55 | 0.57 |
| Conditional DETR | 1.0 | 0.88 | 0.67 | 0.84 | 1.0 | 0.8 | 0.86 | 1.0 | 0.84 | 0.95 |
| DDQ | 1.0 | 1.0 | 1.0 | 1.0 | 1.0 | 1.0 | 1.0 | 1.0 | 1.0 | 1.0 |
| DAB-DETR | 1.0 | 1.0 | 1.0 | 1.0 | 1.0 | 1.0 | 1.0 | 0.99 | 1.0 | 0.98 |
| Deformable DETR | 1.0 | 1.0 | 0.94 | 0.99 | 1.0 | 1.0 | 1.0 | 1.0 | 1.0 | 1.0 |
| DINO | 1.0 | 1.0 | 0.93 | 1.0 | 1.0 | 1.0 | 1.0 | 1.0 | 1.0 | 1.0 |
| PVT | 1.0 | 1.0 | 0.95 | 1.0 | 1.0 | 1.0 | 1.0 | 1.0 | 1.0 | 0.96 |
| PVTv2 | 1.0 | 0.93 | 0.64 | 0.94 | 0.87 | 1.0 | 0.76 | 1.0 | 0.76 | 0.6 |

Table 18: **Ablation** experimental results (**sphere**) of **vehicle** detection in the metric of **mAR50(%)**.

| | Clean | Random | ACTIVE | DTA | FCA | APPA | POOPatch | 3D2Fool | CAMOU | RPAU |
|---|---|---|---|---|---|---|---|---|---|---|
| ATSS | 0.97 | 0.71 | 0.45 | 0.79 | 0.84 | 0.85 | 0.68 | 0.98 | 0.71 | 0.73 |
| AutoAssign | 0.98 | 0.9 | 0.74 | 0.92 | 0.94 | 1.0 | 0.83 | 0.98 | 0.91 | 0.99 |
| CenterNet | 0.98 | 0.91 | 0.58 | 0.91 | 0.92 | 0.89 | 0.93 | 0.9 | 0.88 | 0.94 |
| CentripetalNet | 0.98 | 0.78 | 0.66 | 0.79 | 0.9 | 0.86 | 0.7 | 0.85 | 0.71 | 0.86 |
| CornerNet | 0.96 | 0.75 | 0.61 | 0.87 | 0.93 | 0.86 | 0.7 | 0.94 | 0.76 | 0.82 |
| DDOD | 0.99 | 0.96 | 0.74 | 0.99 | 0.86 | 1.0 | 0.82 | 1.0 | 0.97 | 0.94 |
| DyHead | 1.0 | 0.79 | 0.53 | 0.79 | 0.92 | 0.8 | 0.79 | 0.73 | 0.77 | 0.99 |
| EfficientNet | 1.0 | 0.94 | 0.86 | 0.99 | 0.98 | 1.0 | 0.98 | 0.98 | 0.96 | 0.93 |
| FCOS | 1.0 | 0.99 | 0.99 | 1.0 | 1.0 | 1.0 | 1.0 | 1.0 | 1.0 | 1.0 |
| FoveaBox | 0.94 | 0.84 | 0.5 | 0.76 | 0.87 | 0.81 | 0.72 | 0.8 | 0.76 | 0.91 |
| FreeAnchor | 0.97 | 0.73 | 0.79 | 0.88 | 0.81 | 0.98 | 0.74 | 0.85 | 0.84 | 0.88 |
| FSAF | 0.93 | 0.75 | 0.49 | 0.8 | 0.72 | 0.81 | 0.67 | 0.89 | 0.74 | 0.84 |
| GFL | 1.0 | 0.8 | 0.48 | 0.9 | 0.95 | 0.95 | 0.89 | 0.92 | 0.81 | 0.9 |
| LD | 0.98 | 0.86 | 0.52 | 0.9 | 0.92 | 0.87 | 0.75 | 0.94 | 0.85 | 0.93 |
| NAS-FPN | 1.0 | 0.92 | 0.66 | 0.96 | 0.98 | 0.97 | 0.73 | 0.82 | 0.75 | 0.88 |
| PAA | 0.99 | 0.96 | 0.9 | 0.97 | 0.93 | 0.99 | 0.86 | 0.99 | 1.0 | 0.95 |
| RetinaNet | 1.0 | 0.81 | 0.67 | 0.85 | 0.89 | 0.86 | 0.81 | 0.84 | 0.83 | 0.8 |
| RTMDet | 1.0 | 1.0 | 0.93 | 0.99 | 1.0 | 0.98 | 0.99 | 1.0 | 1.0 | 0.98 |
| TOOD | 0.9 | 0.72 | 0.48 | 0.9 | 0.73 | 0.79 | 0.72 | 0.84 | 0.74 | 0.86 |
| VarifocalNet | 0.99 | 0.75 | 0.45 | 0.66 | 0.87 | 0.83 | 0.67 | 0.8 | 0.67 | 0.79 |
| YOLOv5 | 1.0 | 1.0 | 0.98 | 1.0 | 1.0 | 1.0 | 1.0 | 1.0 | 0.97 | 0.99 |
| YOLOv6 | 1.0 | 1.0 | 0.97 | 1.0 | 1.0 | 1.0 | 1.0 | 1.0 | 0.99 | 1.0 |
| YOLOv7 | 1.0 | 0.94 | 0.97 | 0.94 | 0.98 | 1.0 | 0.96 | 1.0 | 0.98 | 0.99 |
| YOLOv8 | 1.0 | 0.95 | 0.87 | 0.93 | 1.0 | 0.98 | 0.89 | 0.99 | 0.84 | 0.99 |
| YOLOX | 1.0 | 0.97 | 0.77 | 0.97 | 0.99 | 1.0 | 0.98 | 0.92 | 0.94 | 0.98 |
| Faster R-CNN | 0.85 | 0.45 | 0.35 | 0.47 | 0.54 | 0.59 | 0.49 | 0.61 | 0.44 | 0.49 |
| Cascade R-CNN | 0.82 | 0.51 | 0.4 | 0.52 | 0.58 | 0.64 | 0.53 | 0.62 | 0.48 | 0.51 |
| Cascade RPN | 1.0 | 0.91 | 0.62 | 0.93 | 0.96 | 0.99 | 0.96 | 1.0 | 0.86 | 0.96 |
| Double Heads | 0.83 | 0.61 | 0.44 | 0.58 | 0.58 | 0.72 | 0.54 | 0.8 | 0.57 | 0.57 |
| FPG | 1.0 | 0.78 | 0.5 | 0.93 | 0.82 | 0.99 | 0.7 | 0.94 | 0.9 | 0.71 |
| Grid R-CNN | 0.89 | 0.51 | 0.37 | 0.6 | 0.58 | 0.73 | 0.55 | 0.76 | 0.47 | 0.55 |
| Guided Anchoring | 1.0 | 0.97 | 0.98 | 0.99 | 0.99 | 1.0 | 0.89 | 1.0 | 0.97 | 0.97 |
| HRNet | 0.89 | 0.6 | 0.52 | 0.59 | 0.54 | 0.73 | 0.51 | 0.84 | 0.56 | 0.6 |
| Libra R-CNN | 0.94 | 0.76 | 0.49 | 0.8 | 0.9 | 0.74 | 0.85 | 0.81 | 0.72 | 0.93 |
| PAFPN | 0.84 | 0.55 | 0.37 | 0.52 | 0.59 | 0.65 | 0.58 | 0.71 | 0.5 | 0.54 |
| RepPoints | 0.97 | 0.89 | 0.52 | 0.84 | 0.83 | 0.84 | 0.8 | 0.95 | 0.94 | 0.84 |
| Res2Net | 0.88 | 0.59 | 0.52 | 0.66 | 0.77 | 0.83 | 0.54 | 0.79 | 0.55 | 0.73 |
| ResNeSt | 0.91 | 0.63 | 0.43 | 0.8 | 0.78 | 0.91 | 0.68 | 0.92 | 0.64 | 0.62 |
| SABL | 0.82 | 0.49 | 0.4 | 0.52 | 0.56 | 0.57 | 0.53 | 0.71 | 0.46 | 0.49 |
| Sparse R-CNN | 0.99 | 0.89 | 0.55 | 0.79 | 0.95 | 0.81 | 0.83 | 0.9 | 0.82 | 0.86 |
| DETR | 0.71 | 0.44 | 0.3 | 0.32 | 0.57 | 0.52 | 0.63 | 0.43 | 0.38 | 0.5 |
| Conditional DETR | 1.0 | 0.9 | 0.71 | 0.94 | 1.0 | 0.92 | 0.8 | 0.98 | 0.9 | 0.94 |
| DDQ | 1.0 | 1.0 | 1.0 | 1.0 | 1.0 | 1.0 | 1.0 | 1.0 | 1.0 | 1.0 |
| DAB-DETR | 1.0 | 0.94 | 1.0 | 0.99 | 0.99 | 1.0 | 0.98 | 0.96 | 1.0 | 0.99 |
| Deformable DETR | 1.0 | 0.9 | 0.74 | 0.86 | 0.87 | 0.93 | 0.95 | 0.96 | 0.86 | 0.91 |
| DINO | 1.0 | 0.92 | 0.95 | 0.99 | 0.96 | 1.0 | 1.0 | 1.0 | 0.92 | 1.0 |
| PVT | 0.98 | 0.95 | 0.78 | 0.87 | 0.96 | 0.97 | 0.91 | 0.94 | 0.87 | 0.8 |
| PVTv2 | 1.0 | 0.99 | 0.77 | 0.97 | 1.0 | 1.0 | 0.78 | 0.99 | 0.98 | 0.8 |

Table 19: **Ablation** experimental results (**distance**) of **Traffic sign** detection in the metric of **mAR50(%)**.

| | Clean | AdvCam | RP$_2$ | ShapeShifter |
|---|---|---|---|---|
| ATSS | 0.929 | 0.93 | 0.892 | 0.919 |
| AutoAssign | 0.921 | 0.943 | 0.896 | 0.915 |
| CenterNet | 0.903 | 0.91 | 0.875 | 0.914 |
| CentripetalNet | 0.951 | 0.946 | 0.924 | 0.951 |
| CornerNet | 0.942 | 0.951 | 0.929 | 0.947 |
| DDOD | 0.916 | 0.926 | 0.9 | 0.915 |
| DyHead | 0.927 | 0.933 | 0.866 | 0.921 |
| EfficientNet | 0.921 | 0.913 | 0.887 | 0.919 |
| FCOS | 0.939 | 0.932 | 0.913 | 0.929 |
| FoveaBox | 0.915 | 0.917 | 0.871 | 0.913 |
| FreeAnchor | 0.921 | 0.919 | 0.877 | 0.912 |
| FSAF | 0.901 | 0.907 | 0.864 | 0.899 |
| GFL | 0.927 | 0.942 | 0.887 | 0.908 |
| LD | 0.933 | 0.931 | 0.88 | 0.92 |
| NAS-FPN | 0.93 | 0.942 | 0.883 | 0.925 |
| PAA | 0.928 | 0.921 | 0.886 | 0.91 |
| RetinaNet | 0.921 | 0.912 | 0.863 | 0.899 |
| RTMDet | 0.929 | 0.942 | 0.862 | 0.927 |
| TOOD | 0.922 | 0.93 | 0.895 | 0.921 |
| VarifocalNet | 0.929 | 0.932 | 0.902 | 0.921 |
| YOLOv5 | 0.941 | 0.943 | 0.882 | 0.945 |
| YOLOv6 | 0.94 | 0.954 | 0.901 | 0.936 |
| YOLOv7 | 0.945 | 0.942 | 0.89 | 0.942 |
| YOLOv8 | 0.942 | 0.943 | 0.868 | 0.944 |
| YOLOX | 0.923 | 0.922 | 0.866 | 0.906 |
| Faster R-CNN | 0.891 | 0.897 | 0.863 | 0.861 |
| Cascade R-CNN | 0.929 | 0.924 | 0.891 | 0.895 |
| Cascade RPN | 0.927 | 0.93 | 0.887 | 0.901 |
| Double Heads | 0.887 | 0.895 | 0.847 | 0.88 |
| FPG | 0.921 | 0.935 | 0.859 | 0.897 |
| Grid R-CNN | 0.913 | 0.911 | 0.865 | 0.91 |
| Guided Anchoring | 0.928 | 0.921 | 0.899 | 0.925 |
| HRNet | 0.923 | 0.915 | 0.91 | 0.904 |
| Libra R-CNN | 0.921 | 0.922 | 0.885 | 0.905 |
| PAFPN | 0.901 | 0.89 | 0.85 | 0.882 |
| RepPoints | 0.915 | 0.908 | 0.865 | 0.887 |
| Res2Net | 0.91 | 0.897 | 0.861 | 0.899 |
| ResNeSt | 0.929 | 0.901 | 0.872 | 0.889 |
| SABL | 0.92 | 0.913 | 0.88 | 0.898 |
| Sparse R-CNN | 0.931 | 0.925 | 0.9 | 0.927 |
| DETR | 0.908 | 0.904 | 0.878 | 0.933 |
| Conditional DETR | 0.93 | 0.931 | 0.907 | 0.924 |
| DDQ | 0.949 | 0.95 | 0.901 | 0.932 |
| DAB-DETR | 0.931 | 0.94 | 0.902 | 0.919 |
| Deformable DETR | 0.943 | 0.955 | 0.893 | 0.933 |
| DINO | 0.94 | 0.944 | 0.885 | 0.936 |
| PVT | 0.906 | 0.886 | 0.868 | 0.861 |
| PVTv2 | 0.915 | 0.899 | 0.877 | 0.905 |

Table 20: Ablation study on training dataset.

| Physical attacks | Training datasets | Median ASR |
|---|---|---|
| AdvCam | ImageNet | 0 |
| AdvCaT | 376 self-collected images | 0 |
| MTD | - | 2 |
| LAP | INRIA | 2 |
| AdvPattern | Market1501 | 2 |
| AdvTshirt | 40 self-collected videos | 3 |
| DAP | INRIA | 5 |
| NaTPatch | INRIA | 5 |
| InvisCloak | COCO | 5 |
| AdvTexture | INRIA | 7 |

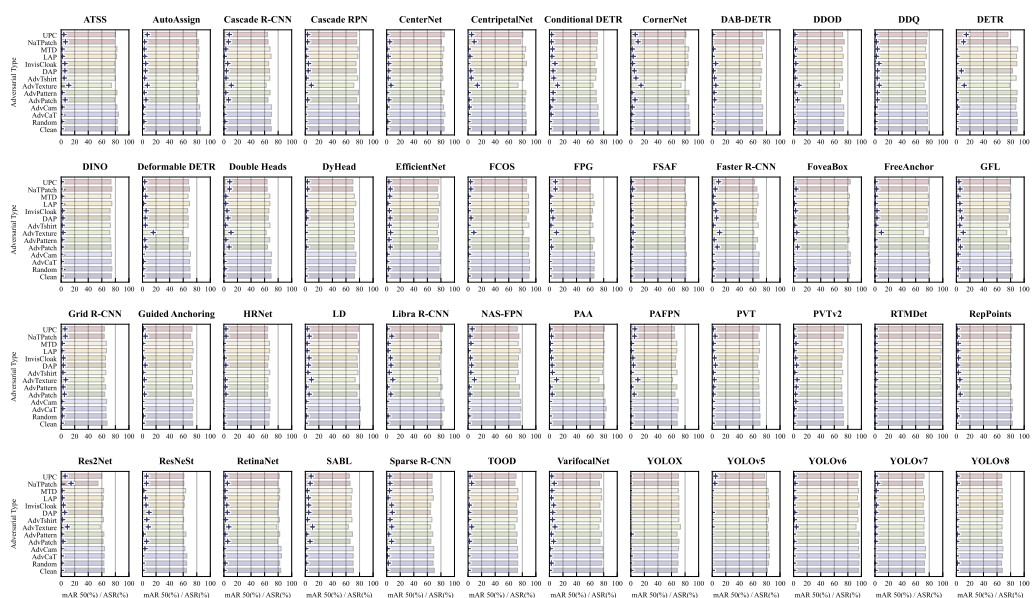

Figure 17: **Overall** experimental results of **person** detection in the metric of **mAR50(%)**, please zoom in for better view.

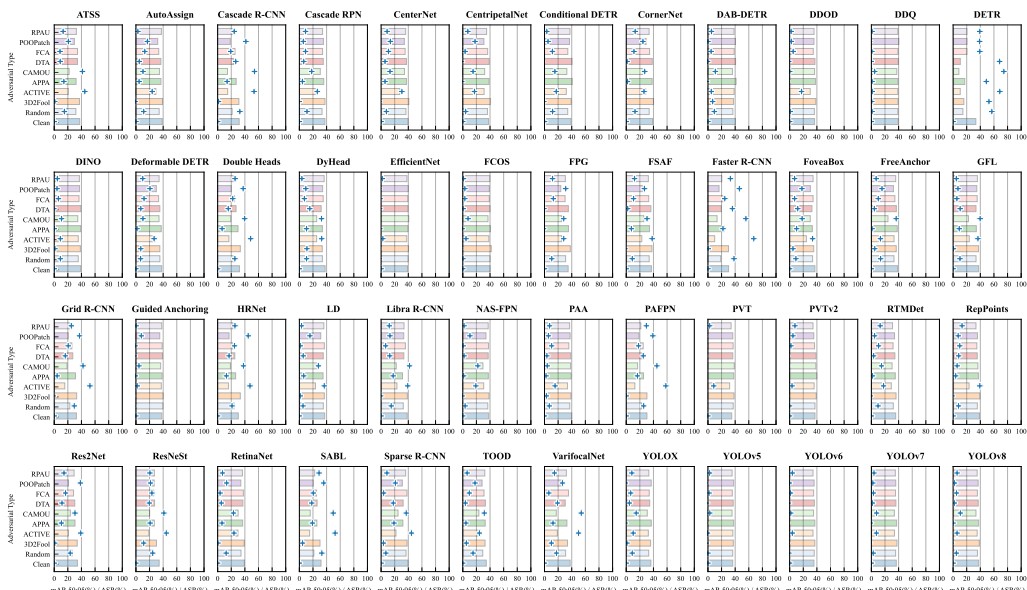

Figure 18: **Overall** experimental results of **vehicle** detection in the metric of mAR50:95(%).

Table 21: Ablation study on 2D and 3D perturbations.

| Perturbations | Entire surface | CornerNet | VarifocalNet |
|---|---|---|---|
| Clean | - | 87 | 80 |
| Random | - | 87 | 77 |
| AdvTexture | ✓ | 74 | 73 |
| AdvTexture | ✗ | 81(7) | 77(4) |
| AdvPatch | ✓ | 82 | 75 |
| AdvPatch | ✗ | 85(3) | 79(4) |
| NatPatch | ✓ | 78 | 74 |
| NatPatch | ✗ | 83(5) | 77(3) |

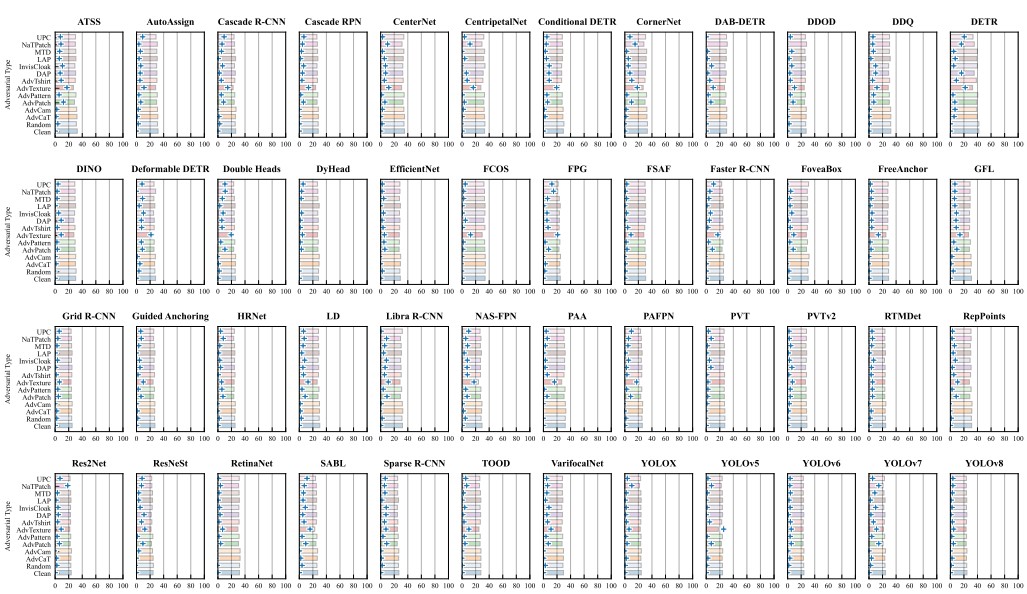

Figure 19: **Overall** experimental results of **person** detection in the metric of mAR50:95(%).

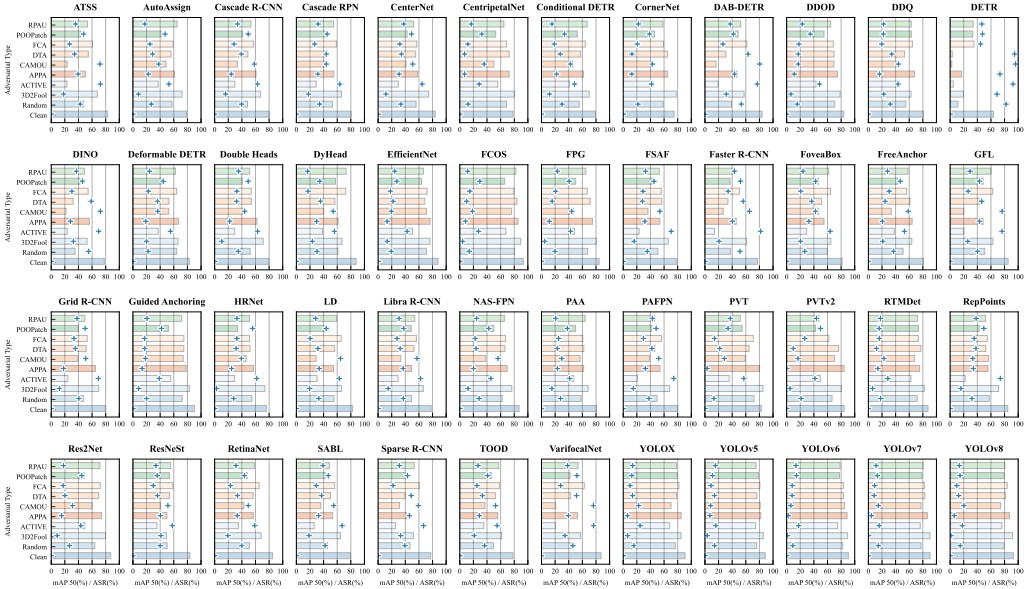

Figure 20: **Overall** experimental results of **vehicle** detection in the metric of mAP50(%).

Table 22: Comparison of reported and reproduced results.

|  |  | Clean | Random | CAMOU | DTA | ACTIVE |
|---|---|---|---|---|---|---|
| YOLOv3 | Reported | 86 | 67 | 60 | 32 | 23 |
|  | Reproduced | 86 | 66 | 62 | 33 | 23 |
| YOLOv7 | Reported | 93 | 86 | 83 | 59 | 42 |
|  | Reproduced | 93 | 85 | 83 | 60 | 41 |
| PVT | Reported | 89 | 78 | 69 | 56 | 52 |
|  | Reproduced | 89 | 78 | 69 | 56 | 51 |

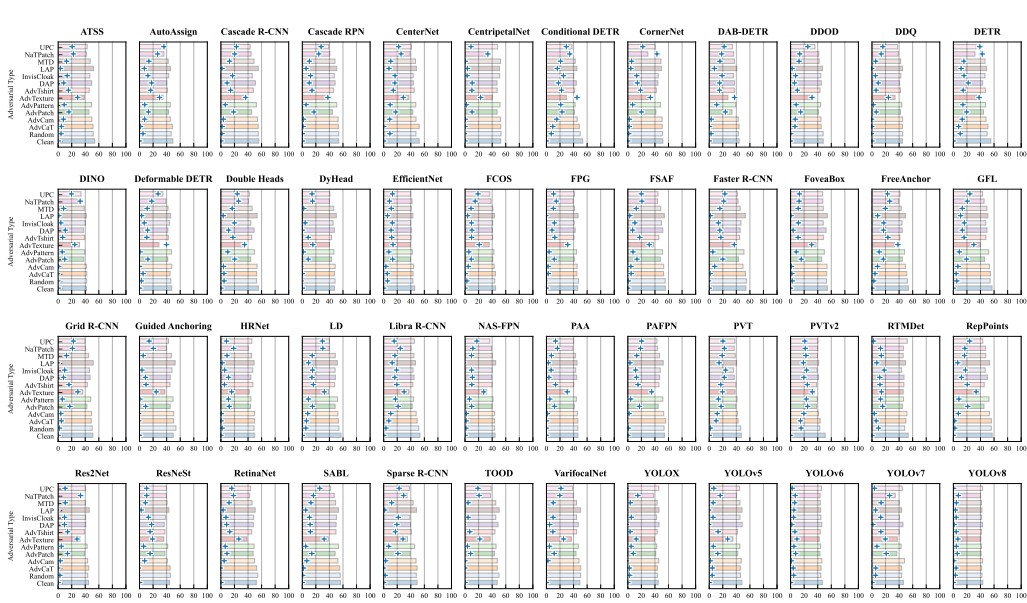

Figure 21: **Overall** experimental results of **person** detection in the metric of mAP50(%).

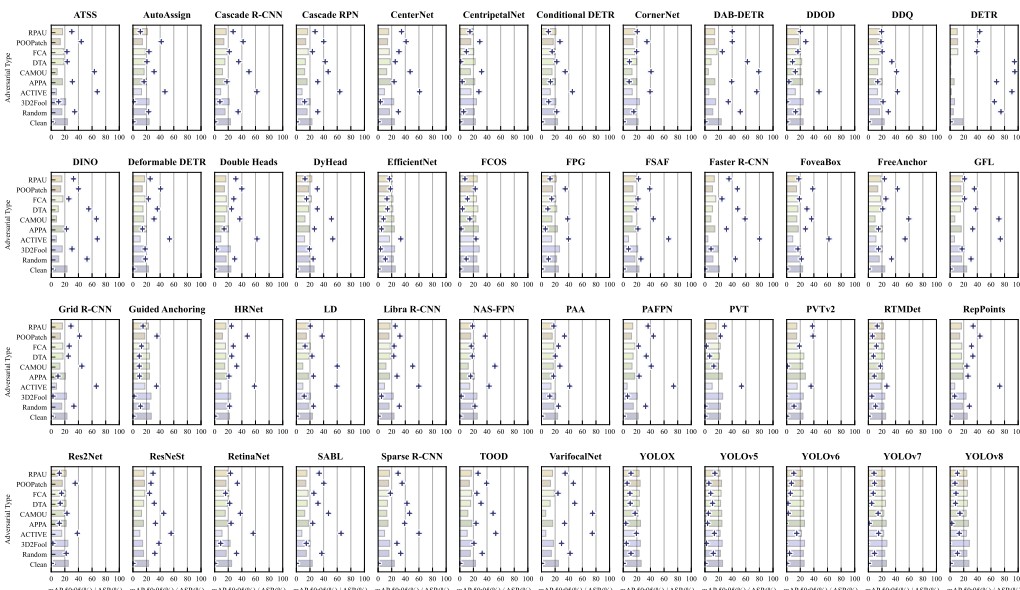

Figure 22: **Overall** experimental results of **vehicle** detection in the metric of mAP50:95(%).

Table 23: User feedback survey.

| Number | Questions |
|---|---|
| Q1 | How easy was it to follow the Docker installation guide for CARLA? (Rating 1-5) |
| Q2 | How helpful was the tutorial on customizing adversarial objects in the documentation? (Rating 1-5) |
| Q3 | Were you able to successfully deploy CARLA using the provided resources? (Yes or No) |
| Q4 | Were you able to successfully customize adversarial objects using the provided resources? (Yes or No) |
| Q5 | Overall, how satisfied are you with the ease of CARLA deployment and customizing adversarial objects? (Rating 1-5) |

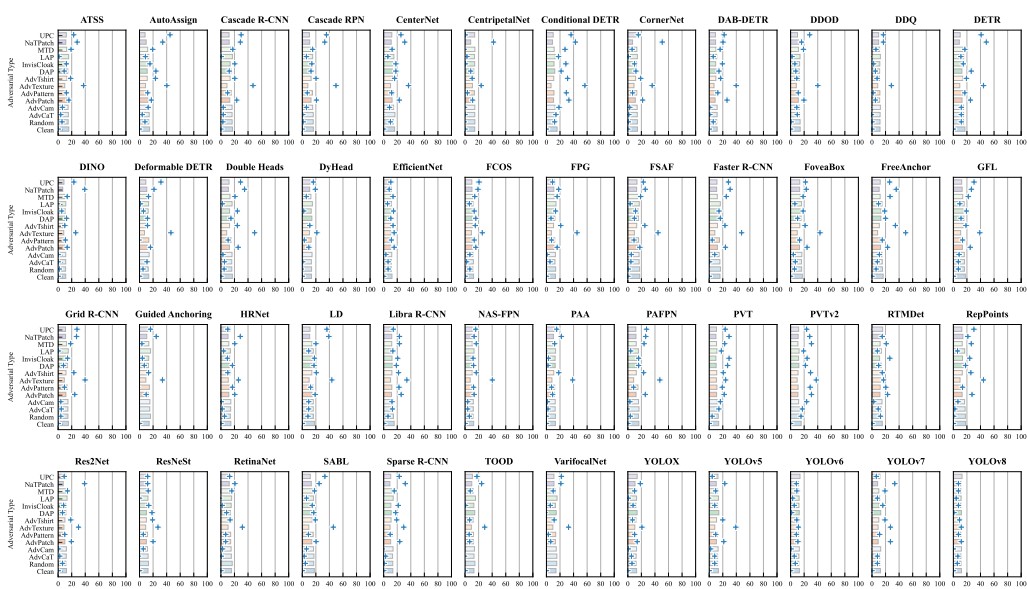

Figure 23: **Overall** experimental results of **person** detection in the metric of mAP50:95(%).

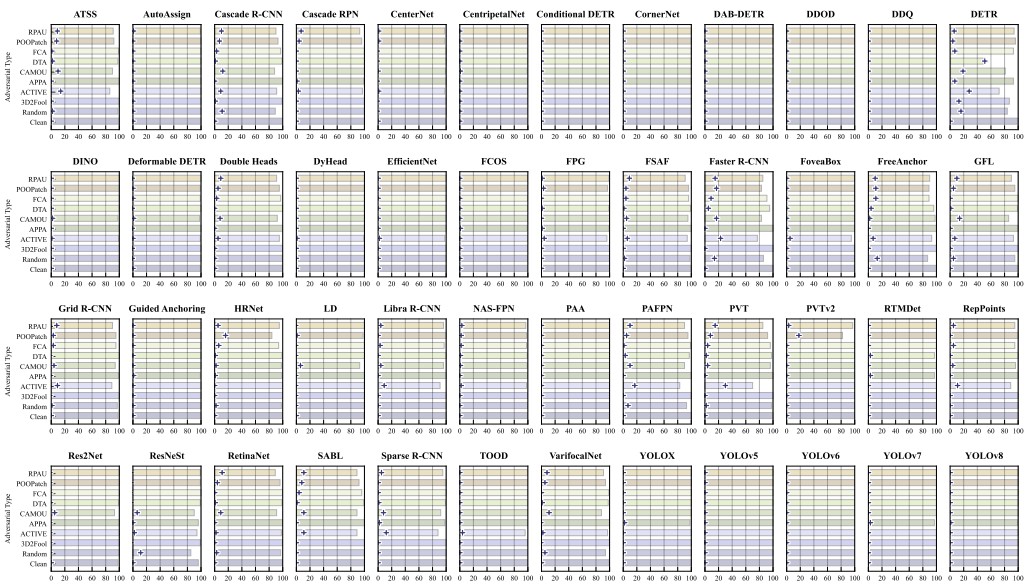

Figure 24: The **ablation** experimental results of **vehicle** detection on **Azimuth angle** ($\phi$) in the metric of mAR50(%).

Table 24: User feedback survey.

| Questions | User1 | User2 | User3 | User4 | User5 |
|-----------|-------|-------|-------|-------|-------|
| Q1 | 4 | 5 | 5 | 4 | 5 |
| Q2 | 5 | 5 | 5 | 5 | 5 |
| Q3 | Yes | Yes | Yes | Yes | Yes |
| Q4 | Yes | Yes | Yes | Yes | Yes |
| Q5 | 4 | 5 | 5 | 4.5 | 5 |

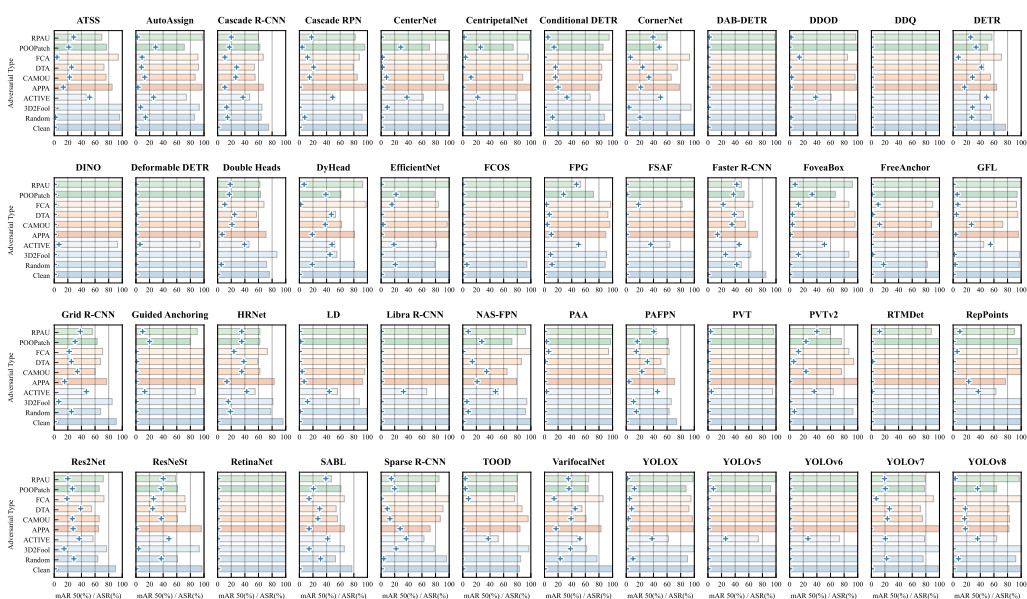

Figure 25: The **ablation** experimental results of **vehicle** detection on **Altitude angle** ($\theta$) in the metric of mAR50(%).

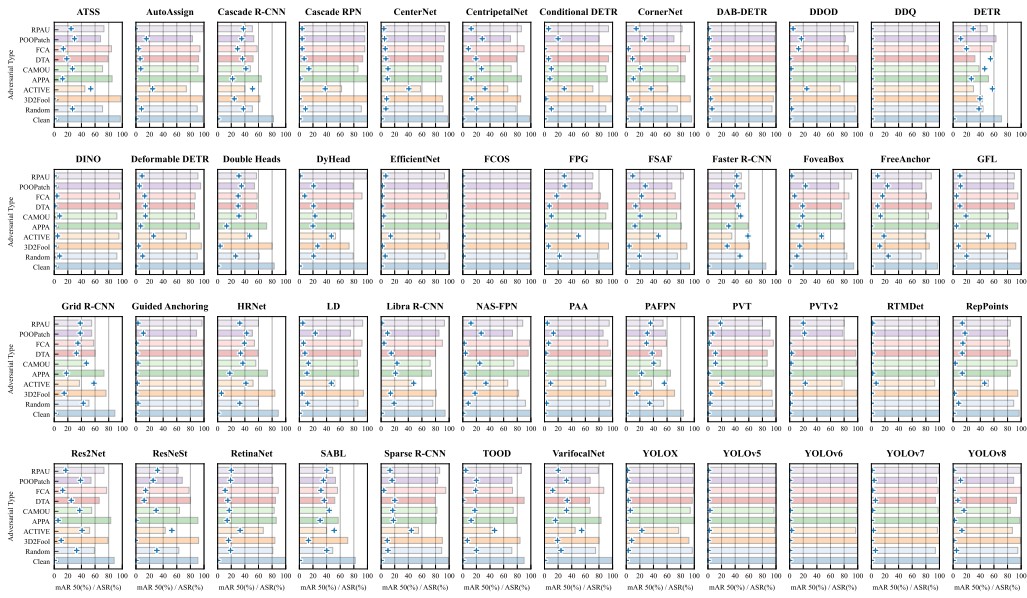

Figure 26: The **ablation** experimental results of **vehicle** detection on **Ball-space** in the metric of mAR50(%).

