# OpenReview forum: "PADetBench: Towards Benchmarking Physical Attacks against Object Detection"
_ICLR.cc/2025/Conference — ICLR 2025 Conference Withdrawn Submission_

### Official Review · Reviewer_sHDw · 2024-10-26

**Soundness:** 3
**Presentation:** 3
**Contribution:** 3
**Rating:** 6
**Confidence:** 4

**Summary:**

The paper proposes a highly flexible and scalable benchmark for physical adversarial attacks against detection models and evaluates physical adversarial attacks under various physical dynamics by real-world simulators. It has a complete end-to-end pipeline, including data generation, detection, evaluation, and analysis. Further, it generates comprehensive evaluations and analyses to highlight the limitations of existing algorithms and provide considerable insights.

**Strengths:**

1. The paper shows great efforts to organize such a large benchmark and provide detailed analysis.
2. Overall, the paper is well-written and almost clear to me.
3. Analysis tools in the benchmark are helpful to many researchers, and the user feedbacks reflect its ease of use.
4. The discussion about "where are we" and "where to go" is interesting and inspiring.

**Weaknesses:**

1. The presentation can be improved. Although comprehensive experimental results are necessary, some crowd the layout and cause a terrible experience. For instance, Figure 2 provides extensive results while each sub-figure holds only a small subset of space. It makes readers try their best to broaden the figure and keep their eyes fixed on it. Selectively displaying results may be a better choice.
2. The comparison of other benchmarks in object detection should be available. Since the benchmark is proposed to illustrate the data, detectors, and adversarial attacks in object detection, the comparison of previous works or demonstration of the applied methods is important. In fact, the corresponding information is stated in Table 1 (target objects) and Table 2 (detectors). However, a clear comparison of the previous benchmarks is missing, which can be introduced in a list like the tables above.
3. The applied objects and physical simulator might be limited. Various detectors and adversarial attacks are utilized in the paper, while the physical simulator and the applied objects are limited. For example, more objects like border trees besides vehicles, persons, and traffic signs can be considered. Furthermore, the fixed simulator is likely to retain a special pattern in the generated data, which may lead to less reliable results on the benchmark.
4. The transferability of adversarial attacks should be considered in the analysis. In practice, the detectors are unknown to attackers, usually named as a "black-box" setting in adversarial attacks, and attackers are likely to create adversarial examples based on a substitute model by the transferability of adversarial examples. The analysis of the transferability of adversarial examples in object detection seems absent, though the evaluation in white-box settings is available (e.g., Figure 3 and Figure 4).

**Questions:**

1. Could authors publicize the code and data when the paper is under review? Since the paper is proposed as a benchmark, I suggest the authors release their code and data when it is under review. In my opinion, their extensive efforts can be better validated if these materials are accessible. However, the authors claim the abstract by saying, "The code and datasets will be publicly available." Indeed, I respect the authors' choice and promise that their choice would not reduce my ranking.
2. The evaluation in real scenarios can be taken into account. The paper introduces a benchmark on physical attacks against object detection but only contains examples synthesized by simulators. The overhead cost of time and financial resources is somewhat troublesome in producing plenty of experiments. However, small experiments can be done as in previous works. The paper is named with "physical attacks" and no special qualifier, isn't it?
3. The paper ignores the adversarial defense methods or common corruption in objection detection. Actually, adversarial defense has rapidly developed in recent years, along with the development of adversarial attacks. Nowadays, many classifiers and detectors are protected with adversarial defenses against adversarial examples. Besides, common corruption like blur or compression is present in detector inputs under real-world scenarios. Both adversarial defense and common corruption play significant roles in evaluating physical attacks. However, corresponding results suggest that the evaluation may be insufficient in the real world.
4. The primary metric worth rethinking. The primary metric used in the benchmark is mAR (mean Average Recall), explained as the ratio of TP and GT (GT = TP + FN). Nevertheless, the mAP (mean Average Precision), the ratio of TP and all predictions (predictions = TP + FP) may be more representative in evaluating physical attacks. Physical attacks can generally mislead detectors to wrong results with varying TP and FP. The mAR can only examine the influence of physical attacks on TP, while mAP shows effectiveness in evaluating TP and FP. It seems that mAP can be a better choice, as it is also frequently present in object detection.

---

### Official Review · Reviewer_ZYTh · 2024-11-01

**Soundness:** 3
**Presentation:** 1
**Contribution:** 3
**Rating:** 5
**Confidence:** 3

**Summary:**

This work proposes a general benchmark for assessing the performance of physical adversarial attacks. Additionally, it involves over 8,000 evaluations to strengthen the findings.

**Strengths:**

This work provides extensive experiments to support the proposed idea.

**Weaknesses:**

I appreciate that the authors present extensive experimental results to assess the performance of physical adversarial attacks and address the shortcomings of current benchmarks, including their time-consuming and costly nature, challenges in aligning physical dynamics, cross-domain loss, and difficulties in comparison (lines 46-53). However, readers unfamiliar with physical adversarial attacks may struggle to understand the significance of these issues and how the proposed benchmark effectively addresses them. As it currently stands, the paper resembles a shopping list, making it difficult for readers to grasp how the problem is solved amidst the multitude of experimental results.

Specifically, the authors do not provide a quantitative metric to demonstrate that the previous benchmark is time-consuming or how the proposed benchmark mitigates this issue. The same critique applies to the other three points. Additionally, adversarial attacks on object detection encompass at least five objectives: appearing attack, hiding attacks, mis-classifying attacks, mis-locating attacks, and latency attacks [1]. While not all objectives have been implemented in physical attacks, the authors should clarify the scope of the proposed metrics (Equations 1 and 2).

Furthermore, the corollaries presented (lines 423-460) do not appear to be novel; similar ideas have been explored in existing literature. The authors should provide stronger empirical evidence to demonstrate that the proposed benchmark is superior to existing works. Overall, I believe this work is a significant milestone for assessing the performance of physical adversarial attacks, but the writing style requires refinement, and the evidence supporting its superiority should be emphasized.

[1] Overload: Latency attacks on object detection for edge devices.

**Questions:**

The authors should pay more attention to how the proposed benchmark solves the problem addressed.

---

### Official Review · Reviewer_9B3c · 2024-11-01

**Soundness:** 2
**Presentation:** 1
**Contribution:** 2
**Rating:** 3
**Confidence:** 5

**Summary:**

Evaluating and comparing physical attacks in real-world conditions is a complex challenge. Most research on physical attacks assesses the effectiveness of their proposed methods using digital experiments on standard benchmarks like, e.g., COCO, followed by controlled or semi-controlled real-world tests to gauge their impact in real-world conditions. This paper aims to develop a standard benchmark to compare physical attacks in real-world conditions fairly. To do so, the authors generate real-world scenarios for numerous parameters (attacked object type, weather conditions, …) through simulation. This allows shared simulated scenes to compare physical attacks. Using these simulations, they evaluate many physical attacks proposed in the literature against a large ensemble of object detectors.

**Strengths:**

-	Moving toward better evaluations of physical attacks in real-world conditions is important and interesting.
-	A large set of physical attacks and object detectors are evaluated.

**Weaknesses:**

Despite creating a standard benchmark to compare what may be the performance of physical attacks in real-world scenarios, this paper falls short of creating such a benchmark. Too many details are missing, and the presentation could be significantly improved. Please find bellow the different weaknesses that I find necessary to address:

- A lack of comparison with other benchmarks of the literature. For example, Zhang et al. (2023b) generated the DCI dataset using the CARLA simulator. What is the difference between the author's work and the work of Zhang et al. (2023b)? Is it just the number of physical attacks and object detectors included in the evaluation? For physical attacks against traffic signs, what makes the author's work valuable over Hingun et al. (2023) work? Hingun et al. (2023) propose to model real-world conditions to better project and apply the patch in the image. Which of your or their benchmark is better suited to compare physical attacks against traffic sign recognition? Which of these benchmarks most accurately represents real-world conditions?

- A lack of details about how the datasets are generated, how the different physical attacks are projected into the scene. Did you use the physical attacks available in the GitHub that may be associated with the attack, or did you re-implement and design the attack yourself? On which object detectors are the different physical attacks optimized? How are the different ground truths boxes generated?

- The main weakness for me is the following. The authors express the need to model better cross-domain transformations $T_{P2D}(T_{D2P} (δ))$. I agree that this is an interesting and important research direction that may benefits the physical attack community. However, this paper does not propose a contribution to advance in this direction. What are the discrepancies between the simulations you used and real-world scenarios, considering that the used simulations may not accurately represent real-world conditions? Can I expect the ranking of physical attacks to remain consistent in real-world environments? Evaluations using the COCO dataset or simulations rely on digital experiments, and it is still unclear which serves as the best proxy for real-world conditions. To close the gap between numerical experiment and physical experiment, Hingun et al. (2023) propose to model the brightness of real-world scenes to better project the patch in the image. It would be a valuable contribution if the authors could provide such an experiment.

**Questions:**

In addition to the questions in the weakness section, please find additional questions below.

- Why is the clean performance of DETR that bad? Same question for Faster R-CNN. Did you use recent versions of these detectors?
- What is the meaning of the following sentence: “This phenomenon is caused by the victim models of the
attack method lagging behind the development of the detection method, which also motivates us to fill this gap. »?

---

### Official Review · Reviewer_u7ZA · 2024-11-03

**Soundness:** 3
**Presentation:** 3
**Contribution:** 3
**Rating:** 5
**Confidence:** 5

**Summary:**

This paper provides an end-to-end pipeline to evaluate physical adversarial examples with different parameters, including environments, vehicle and pedestrian models, weather patterns, and camera placements. The authors benchmarked 23 physical attacks with different target object detectors, physical dynamics, and evaluation metrics.

**Strengths:**

1.	The goal of this paper is significant to this area.
2.	The authors evaluated attacks under multiple metrics and discussed the strength of some metrics.
3.	The authors conducted comprehensive evaluations.

**Weaknesses:**

1.	Some figures are hard to read, e.g. Fig 1. Some of the text is too small and blurred.
2.	Some settings seem problematic. Please see the questions.
3.	The paper lacks conclusions and inspirations drawn from benchmarking multiple attacks. For example, which technique is essential in improving adversarial effectiveness?

**Questions:**

1.	What does the sphere object/the Sphere text mean in Figures 1 and 6?
2.	Did the authors benchmark the attacks under the white-box setting or black-box setting? It is essential to evaluate these settings separately for equity.
3.	How much is the gap between the simulated environment and the real world? Are the rankings of the attacks consistent with the real world?
4.	According to previous work related to adversarial clothes, the human body and clothes are non-rigid, which makes 3D simulation and generalization to the real world very difficult. How did the authors address this problem?

I understand that some problems are difficult to address, and this paper's overall goal is worth advocating. I’m happy to raise the score if the authors can provide some inspiration to this area.

---

### Note · Authors · 2024-11-22

I have read and agree with the venue's withdrawal policy on behalf of myself and my co-authors.